# Scan and Snap: Understanding Training Dynamics and Token Composition in 1-layer Transformer

Yuandong Tian[1]    Yiping Wang[2,4]    Beidi Chen[1,3]    Simon Du[2]

[1]Meta AI (FAIR)    [2]University of Washington    [3]Carnegie Mellon University    [4]Zhejiang University
{yuandong,beidic}@meta.com,    {ypwang61,ssdu}@cs.washington.edu

## Abstract

Transformer architectures have shown impressive performance in multiple research domains and have become the backbone of many neural network models. However, there is limited understanding on how Transformer works. In particular, with a simple predictive loss, how the representation emerges from the gradient *training dynamics* remains a mystery. In this paper, we analyze the SGD training dynamics for 1-layer transformer with one self-attention plus one decoder layer, for the task of next token prediction in a mathematically rigorous manner. We open the black box of the dynamic process of how the self-attention layer combines input tokens, and reveal the nature of underlying inductive bias. More specifically, with the assumption (a) no positional encoding, (b) long input sequence, and (c) the decoder layer learns faster than the self-attention layer, we prove that self-attention acts as a *discriminative scanning algorithm*: starting from uniform attention, it gradually attends more to key tokens that are distinct for a specific next token to be predicted, and pays less attention to common key tokens that occur across different next tokens. Among distinct tokens, it progressively drops attention weights, following the order of low to high co-occurrence between the key and the query token in the training set. Interestingly, this procedure does not lead to winner-takes-all, but decelerates due to a *phase transition* that is controllable by the learning rates of the two layers, leaving (almost) fixed token combination. We verify this **scan and snap** dynamics on synthetic and real-world data (WikiText).

## 1 Introduction

The Transformer architecture [1] has demonstrated wide applications in multiple research domains, including natural language processing [2, 3, 4], computer vision [5, 6, 7], speech [8, 9], multimodality [10, 11], etc. Recently, large language models (LLMs) based on decoder-only Transformer architecture also demonstrate impressive performance [4, 12, 13], after fine-tuned with instruction data [14] or reward models [15]. Why a pre-trained model, often supervised by simple tasks such as predicting the next word [4, 3, 13] or filling in the blanks [2, 16, 17], can learn highly valuable representations for downstream tasks, remains a mystery.

To understand how Transformer works, many previous works exist. For example, it has been shown that Transformer is a universal approximator [18], can approximate Turing machines [19, 20], and can perform a diverse set of tasks, e.g., hierarchical parsing of context-free grammar [21], if its weights are set properly. However, it is unclear whether the weights designed to achieve specific tasks are at a critical point, or can be learned by SoTA optimizers (e.g., SGD, Adam [22], AdaFactor [23], AdamW [24]). In fact, many existing ML models, such as $k$-NN, Kernel SVM, or MLP, are also universal approximators, while their empirical performance is often way below Transformer.

To demystify such a behavior, it is important to understand the *training dynamics* of Transformer, i.e., how the learnable parameters change over time during training. In this paper, as a first step, we formally characterize the SGD training dynamics of 1-layer position-encoding-free Transformer for

next token prediction, a popular training paradigm used in GPT series [3, 4], in a mathematically rigorous manner. The 1-layer Transformer contains one softmax self-attention layer followed by one decoder layer which predicts the next token. Under the assumption that the sequence is long, and the decoder learns faster than the self-attention layer, we prove the following interesting dynamic behaviors of self-attention during training. **Frequency Bias**: it progressively pays more attention to key tokens that co-occur a lot with the query token, and loses attention to tokens that co-occur less. **Discriminative Bias**: it pays attention to distinct tokens that appear uniquely given the next token to be predicted, while loses interest to common tokens that appear across multiple next tokens. These two properties suggest that self-attention implicitly runs an algorithm of *discriminative scanning*, and has an inductive bias to favor unique key tokens that frequently co-occur with the query ones.

Furthermore, while self-attention layer tends to become more sparse during training, as suggested by Frequency Bias, we discover that it will not collapse to one-hot, due to a *phase transition* in the training dynamics. In the end, the learning does not converge to any stationary points with zero gradient, but ventures into a region where the attention changes slowly (i.e., logarithmically over time), and appears frozen and learned. We further show that the onset of the phase transition are controlled by the learning rates: large learning rate gives sparse attention patterns, and given fixed self-attention learning rate, large decoder learning rate leads to faster phase transition and denser attention patterns. Finally, the SGD dynamics we characterize in this work, named **scan and snap**, is verified in both synthetic and simple real-world experiments on WikiText [25].

**Concurrent works on Transformer dynamics**. Compared to [26] that uses $\ell_2$ loss, our analysis focuses on cross-entropy, which is more realistic, imposes no prior knowledge on possible attention patterns inaccessible to training, and allows tokens to be shared across topics. Compared to [27] that analyzes "positional attention" that is independent of input data with symmetric initialization, our analysis focuses on attention on input data without symmetric assumptions. [28, 29, 30] give similar conclusions that self-attention attends to relevant tokens. In comparison, our work analyzes richer phenomena in 1-layer transformers related to frequency and discriminative bias, which has not been brought up by these works. For example, sparse attention patterns are connected with co-occurrence frequency of contextual token and query, characterization of such connection over training with softmax, including two-stage behaviors of attention logits, etc. We also leverage analytical solutions to certain nonlinear continuous dynamics systems that greatly simplifies the analysis. Detailed comparison can be found in Appendix B.

## 2 Related Works

**Expressiveness of Attention-based Models**. A line of work studies the expressive power of attention-based models. One direction focuses on the universal approximation power [18, 31, 32, 33, 20]. More recent works present fine-grained characterizations of the expressive power for certain functions in different settings, sometimes with statistical analyses [34, 35, 36, 37, 21, 38, 39, 40]. In particular, there is growing interest in explaining the capability of in-context learning [41] of Transformer, by mapping the gradient descent steps of learning classification/regression into feedforward steps of Transformer layers [42, 43, 44, 45, 37, 46]. Different from our work, the results in these papers are existential and do not take training dynamics into consideration.

**Training Dynamics of Neural Networks**. Previous works analyze the training dynamics in multi-layer linear neural networks [47, 48], in the student-teacher setting [49, 50, 51, 52, 53, 54, 55, 56, 57], and infinite-width limit [58, 59, 60, 61, 62, 63, 64, 65, 66, 67, 68, 69, 70, 71], including extentions to attention-based models [72, 73]. For self-supervised learning, works exist to analyze linear networks [74] and understand the role played by nonlinearity [75]. Focusing on attention-based models, Zhang et al. [76] study adaptive optimization methods in attention models. Jelassi et al. [27] propose an idealized setting and show the vision transformer [5] trained by gradient descent can learn spatial structure. Li et al. [26] show that the 1-layer Transformer can learn a constrained topic model, in which any word belongs to one topic, with $\ell_2$ loss, BERT [2]-like architecture and additional assumptions on learned attention patterns. Snell et al. [77] study the dynamics of a single-head attention head to approximate the learning of a Seq2Seq architecture. While these papers also study the optimization dynamics of attention-based models, they focus on different settings and do not explain the phenomena presented in our paper.

# 3 Problem Setting

**Notation.** Let $\{\boldsymbol{u}_k\}_{k=1}^M$ be $d$-dimensional embeddings, and $\{x_t\}$ be discrete tokens. For each token, $x_t$ takes discrete values from 1 to $M$, denoted as $x_t \in [M]$, and $\boldsymbol{x}_t := \boldsymbol{e}_{x_t} \in \mathbb{R}^M$ is the corresponding one-hot vector, i.e., the $x_t$-th entry of $\boldsymbol{x}_t$ is 1 while others are zero. $\boldsymbol{u}_{x_t}$ is the token embedding at location $t$ in a sequence.

Let $U = [\boldsymbol{u}_1, \ldots, \boldsymbol{u}_M]^\top \in \mathbb{R}^{M \times d}$ be the embedding matrix, in which the $k$-th row of $U$ is the embedding vector of token $k$. $X = [\boldsymbol{x}_1, \ldots, \boldsymbol{x}_{T-1}]^\top \in \mathbb{R}^{(T-1) \times M}$ is the data matrix encoding the sequence of length $T - 1$. $XU \in \mathbb{R}^{(T-1) \times d}$ is the sequence of embeddings for a given sequence $\{x_1, \ldots, x_{T-1}\}$. It is clear that $X\boldsymbol{1}_M = \boldsymbol{1}_{T-1}$.

We use $X[i]$ to denote $i$-th sample in the sequence dataset. Similarly, $x_t[i]$ is the token located at $t$ in $i$-th sample. Let $\mathcal{D}$ be the dataset used for training.

**1-Layer Transformer Architecture**. Given a sequence $\{x_1, \ldots, x_T, x_{T+1}\}$, the embedding after 1-layer self attention is:

$$\hat{\boldsymbol{u}}_T = \sum_{t=1}^{T-1} b_{tT} \boldsymbol{u}_{x_t}, \qquad b_{tT} := \frac{\exp(\boldsymbol{u}_{x_T}^\top W_Q W_K^\top \boldsymbol{u}_{x_t}/\sqrt{d})}{\sum_{t=1}^{T-1} \exp(\boldsymbol{u}_{x_T}^\top W_Q W_K^\top \boldsymbol{u}_{x_t}/\sqrt{d})} \qquad (1)$$

Here $b_{tT}$ is the normalized self-attention weights ($\sum_{t=1}^{T-1} b_{tT} = 1$). One important detail is that we mask the weight that the query token attends to itself, which is also being used in previous works (e.g., QK-shared architecture [78]). See Sec. 7 for discussions about residual connection. Let $\boldsymbol{b}_T := [b_{1T}, \ldots, b_{T-1,T}]^\top \in \mathbb{R}^{T-1}$ be an attention vector, then $\boldsymbol{b}_T^\top \boldsymbol{1} = 1$ and $\hat{\boldsymbol{u}}_T = U^\top X^\top \boldsymbol{b}_T$.

$\ell_2$**-Normalization**. We consider adding a normalization to the output of the self-attention layer: $\tilde{\boldsymbol{u}}_T = U^\top \mathrm{LN}(X^\top \boldsymbol{b}_T)$, where $\mathrm{LN}(\boldsymbol{x}) := \boldsymbol{x}/\|\boldsymbol{x}\|_2$. NormFormer and RMSNorm [79, 80] also leverages this setting (up to a global constant). Our analysis can also be extended to standard LayerNorm [81], which also subtracts the mean of $\boldsymbol{x}$, while [80] shows that mean subtraction may not affect the empirical results much. LLaMA [82] also uses RMSNorm. Empirically $\hat{\boldsymbol{u}}_T$ (or $W_V \hat{\boldsymbol{u}}_T$) is normalized (instead of $X^\top \boldsymbol{b}_T$) and here we use an approximation to facilitate analysis: when the token embedding $\{\boldsymbol{u}_m\}$ are approximately orthogonal to each other, then $\|U^\top \boldsymbol{x}\|_2 \approx \|\boldsymbol{x}\|_2$ and thus $\tilde{\boldsymbol{u}}_T \approx \mathrm{LN}(\hat{\boldsymbol{u}}_T)$.

**Objective**. We maximize the likelihood of predicted $(T + 1)$-th token using cross entropy loss:

$$\max_{W_K, W_Q, W_V, U} J := \mathbb{E}_{\mathcal{D}} \left[ \boldsymbol{u}_{x_{T+1}}^\top W_V \tilde{\boldsymbol{u}}_T - \log \sum_l \exp(\boldsymbol{u}_l^\top W_V \tilde{\boldsymbol{u}}_T) \right] \qquad (2)$$

For simplicity, we consider single-head attention setting, and multiple-head attention can be regarded as single-head setting with simultaneous different initializations (see Sec. 4). We call $x_T = m$ as the **query token** of the sequence, and $x_{T+1} = n$ as the **next token** to be predicted. Other tokens $x_t$ ($1 \le t \le T - 1$) that are encoded in $X$ are called **contextual tokens**. Both the contextual and query tokens can take values from 1 to $M$ (i.e., $m \in [M]$) and next token takes the value from 1 to $K$ (i.e., $n \in [K]$) where $K \le M$. Fig. 1(a) shows the overall setting. For an overview of the notation used in the paper, please check Tbl. 1 in the Appendix.

## 3.1 Reparameterization

Instead of studying the dynamics with respect to the parameters of token embedding $U$, key, value and query projection matrices $W_K, W_Q$ and $W_V$, we study the dynamics of two *pairwise token relation matrices* $Y := UW_V^\top U^\top \in \mathbb{R}^{M \times M}$ and $Z := UW_Q W_K^\top U^\top/\sqrt{d} \in \mathbb{R}^{M \times M}$. Intuitively, entries of $Y$ and $Z$ store the "logits" of pairs of tokens. We regard the empirical parameterization using $U, W_K, W_Q$ and $W_V$ as a specific way of parametrization of $Y$ and $Z$, in order to reduce the number of parameters to be estimated. Previous work also leverage similar parameterization for self-attention layers [27, 46].

For real-world applications, the number of tokens $M$ can be huge (e.g., the vocabulary size $M = 50,272$ in OPT-175B [83] and $M = 32,000$ in LLaMA [82]) and directly optimizing $Y$ and $Z$ would be prohibitive. However, as we will show in this work, from the theoretical perspective, treating $Y$ and $Z$ as independent variables has some unique advantages and leads to useful insights.

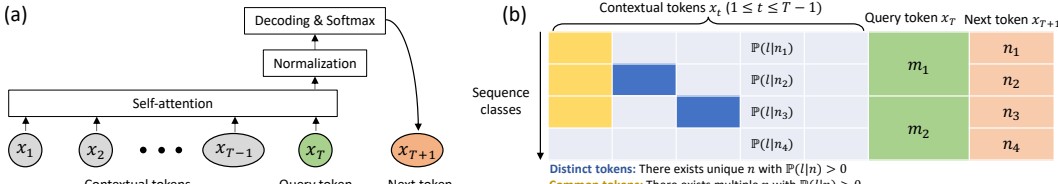

Figure 1: Overall of our setting. **(a)** A sequence with contextual tokens $\{x_1, \ldots, x_{T-1}\}$ and query token $x_T$ is fed into 1-layer transformer (self-attention, normalization and decoding) to predict the next token $x_{T+1}$. **(b)** The definition of sequence classes (Sec. 3.2). A sequence class specifies the conditional probability $\mathbb{P}(l|m, n)$ of the contextual tokens, given the query token $x_T = m$ and the next token $x_{T+1} = n$. For simplicity, we consider the case that the query token is determined by the next token: $x_T = \psi(x_{T+1})$ (and thus $\mathbb{P}(l|m, n) = \mathbb{P}(l|n)$), while the same query token $m$ may correspond to multiple next tokens (i.e., $\psi^{-1}(m)$ is not unique). We study two kinds of tokens: **common tokens** (CT) with $\mathbb{P}(l|n) > 0$ for multiple sequence class $n$, and **distinct tokens** (DT) with $\mathbb{P}(l|n) > 0$ for a single sequence class $n$ only.

**Lemma 1** (Dynamics of 1-layer Transformer). *The gradient dynamics of Eqn. 2 with batchsize 1 is:*

$$\dot{Y} = \eta_Y \mathrm{LN}(X^\top \boldsymbol{b}_T)(\boldsymbol{x}_{T+1} - \boldsymbol{\alpha})^\top, \quad \dot{Z} = \eta_Z \boldsymbol{x}_T (\boldsymbol{x}_{T+1} - \boldsymbol{\alpha})^\top Y^\top \frac{P^\perp_{X^\top \boldsymbol{b}_T}}{\|X^\top \boldsymbol{b}_T\|_2} X^\top \mathrm{diag}(\boldsymbol{b}_T) X \quad (3)$$

*Here $P^\perp_{\boldsymbol{v}} := I - \boldsymbol{v}\boldsymbol{v}^\top/\|\boldsymbol{v}\|_2^2$ projects a vector into $\boldsymbol{v}$'s orthogonal complementary space, $\eta_Y$ and $\eta_Z$ are the learning rates for the decoder layer $Y$ and self-attention layer $Z$, $\boldsymbol{\alpha} := [\alpha_1, \ldots, \alpha_M]^\top \in \mathbb{R}^M$ and $\alpha_m := \exp(Y^\top \mathrm{LN}(X^\top \boldsymbol{b}_T))/\boldsymbol{1}^\top \exp(Y^\top \mathrm{LN}(X^\top \boldsymbol{b}_T))$.*

Please check Appendix C for the proof. We consider $Y(0) = Z(0) = 0$ as initial condition. This is reasonable since empirically $Y$ and $Z$ are initialized by inner product of $d$-dimensional vectors whose components are independently drawn by i.i.d Gaussian. This initial condition is also more realistic than [27] that assumes dominant initialization in diagonal elements. Since $(\boldsymbol{x}_{T+1} - \boldsymbol{\alpha})^\top \boldsymbol{1} = 0$ and $P^\perp_{X^\top \boldsymbol{b}_T} X^\top \mathrm{diag}(\boldsymbol{b}_T)X\boldsymbol{1} = 0$, we have $\dot{Y}\boldsymbol{1} = \dot{Z}\boldsymbol{1} = 0$ and summation of rows of $Z(t)$ and $Y(t)$ remains zero. Since $\boldsymbol{x}_T$ is a one-hot column vector, the update of $Z = [\boldsymbol{z}_1, \boldsymbol{z}_2, \ldots, \boldsymbol{z}_M]^\top$ is done per row:

$$\dot{\boldsymbol{z}}_m = \eta_Z X^\top[i]\mathrm{diag}(\boldsymbol{b}_T[i])X[i]\frac{P^\perp_{X^\top[i]\boldsymbol{b}_T[i]}}{\|X^\top[i]\boldsymbol{b}_T[i]\|_2}Y(\boldsymbol{x}_{T+1}[i] - \boldsymbol{\alpha}[i]) \quad (4)$$

where $m = x_T[i]$ is the query token for sample $i$, $\boldsymbol{z}_m$ is the $m$-th row of $Z$ and $\dot{\boldsymbol{z}}_{m'} = 0$ for row $m' \neq m = x_T[i]$. Note that if $x_T[i] = m$, then $b_T[i]$ is a function of $\boldsymbol{z}_m$ only (but not a function of any other $\boldsymbol{z}_{m'}$). Here we explicitly write down the current sample index $i$, since batchsize is 1.

## 3.2 Data Generation

Next we specify a data generation model (Fig. 1(b)), named *sequence class*, for our analysis.

**Sequence Class.** We regard the input data as a mixture of multiple *sequence classes*. Each sequence class is characterized by a triple $s_{m,n} := (\mathbb{P}(l|m, n), m, n)$. To generate a sequence instance from the class, we first set $x_T = m$ and $x_{T+1} = n$, and then generate the contextual tokens with conditional probability $\mathbb{P}(l|m, n)$. Let $\mathrm{supp}(m, n)$ be the subset of token $l$ with $\mathbb{P}(l|m, n) > 0$.

In this work, we consider the case that given a next token $x_{T+1} = n$, the corresponding sequence always ends with a specific query token $x_T = m =: \psi(n)$. This means that we could index sequence class with next token $x_{T+1} = n$ alone: $s_n := (\mathbb{P}(l|\psi(n), n), \psi(n), n)$, $\mathbb{P}(l|m, n) = \mathbb{P}(l|n)$ and $\mathrm{supp}(n) := \mathrm{supp}(\psi(n), n)$.

Note that $|\psi^{-1}(m)| = 1$ means that the occurrence of token $m$ alone decides next token $n$ to be predicted, regardless of other tokens in the sequence, which is a trivial case. When $|\psi^{-1}(m)| \geq 2$, the same query token $m$, combined with other token $l$ in the sequence with non-zero probability $\mathbb{P}(l|m, n) > 0$, determine the next token.

**Overlapping sequence class**. Two sequence classes $s_n$ and $s_{n'}$ *overlap* if $\mathrm{supp}(n) \cap \mathrm{supp}(n') \neq \emptyset$.

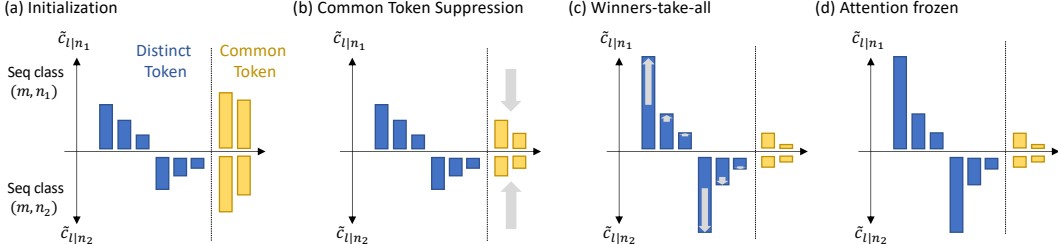

Figure 2: Overview of the training dynamics of self-attention map. Here $\tilde{c}_{l|m,n} := \mathbb{P}(l|m,n)\exp(z_{ml})$ is the un-normalized attention score (Eqn. 5). **(a)** Initialization stage. $z_{ml}(0) = 0$ and $\tilde{c}_{l|m,n} = \mathbb{P}(l|m,n)$. Distinct tokens (Sec. 3.2) shown in blue, common tokens in yellow. **(b)** Common tokens (CT) are suppressed ($\dot{z}_{ml} < 0$, Theorem 2). **(c)** Winners-take-all stage. Distinct tokens (DT) with large initial value $\tilde{c}_{l|m,n}(0)$ start to dominate the attention map (Sec. 5, Theorem 3). **(d)** Once passing the phase transition, i.e., $t \geq t_0 = O(K \ln M / \eta_Y)$, attention appears (almost) frozen (Sec. 6) and token composition is fixed in the self-attention layer.

**(Global) distinct and common tokens**. Let $\Omega(l) := \{n : \mathbb{P}(l|n) > 0\}$ be the subset of next tokens that co-occur with contextual token $l$. We now can identify two kinds of tokens: the *distinct* token $l$ which has $|\Omega(l)| = 1$ and the *common* token $l$ with $|\Omega(l)| > 1$. Intuitively, this means that there exists one common token $l$ so that both $\mathbb{P}(l|n)$ and $\mathbb{P}(l|n')$ are strictly positive, e.g., common words like 'the', 'this', 'which' that appear in many sequence classes. In Sec. 5, we will see how these two type of contextual tokens behave very differently when self-attention layer is involved in the training: distinct tokens tend to be paid attention while common tokens tend to be ignored.

### 3.3 Assumptions

To make our analysis easier, we make the following assumptions:

**Assumption 1.** *We consider **(a)** no positional encoding, **(b)** The input sequence is long ($T \to +\infty$) and **(c)** The decoder layer learns much faster than the self-attention layer (i.e., $\eta_Y \gg \eta_Z$).*

Assumption 1(a) suggests that the model is (almost) permutation-invariant. Given the next token to predict $x_{T+1} = n$ and the query token $x_T = m$ acted as query, the remaining tokens in the sequence may shuffle. Assumption 1(b) indicates that the frequency of a token $l$ in the sequence approaches its conditional probability $\mathbb{P}(l|m,n) := \mathbb{P}(l|x_T = m, x_{T+1} = n)$.

Note that the assumptions are comparable with or even weaker than previous works, e.g., [27] analyzes positional attention with symmetric initialization, without considering input data and [28] models the data distribution as discriminative/non-discriminative patterns, similar to our common/distinct tokens. Empirically, NoPE [84] shows that decoder-only Transformer models without positional encoding still works decently, justifying that Assumption 1(a) is reasonable.

Given the event $\{x_T = m, x_{T+1} = n\}$, suppose for token $l$, the conditional probability that it appears in the sequence is $\mathbb{P}(l|m,n)$. Then for very long sequence $T \to +\infty$, in expectation the number of token $l$ appears in a sequence of length $T$ approaches $T\mathbb{P}(l|m,n)$. Therefore the *per-token* self-attention weight $c_{l|m,n}$ is computed as:

$$c_{l|m,n} := \frac{T\mathbb{P}(l|m,n)\exp(z_{ml})}{\sum_{l'} T\mathbb{P}(l'|m,n)\exp(z_{ml'})} = \frac{\mathbb{P}(l|m,n)\exp(z_{ml})}{\sum_{l'} \mathbb{P}(l'|m,n)\exp(z_{ml'})} =: \frac{\tilde{c}_{l|m,n}}{\sum_{l'} \tilde{c}_{l'|m,n}} \quad (5)$$

Here $z_{ml}$ is $\boldsymbol{z}_m$'s $l$-th entry and $\tilde{c}_{l|m,n} := \mathbb{P}(l|m,n)\exp(z_{ml})$ is un-normalized attention score.

**Lemma 2.** *Given the event $\{x_T = m, x_{T+1} = n\}$, when $T \to +\infty$, we have*

$$X^\top \boldsymbol{b}_T \to \boldsymbol{c}_{m,n}, \qquad X^\top \mathrm{diag}(\boldsymbol{b}_T)X \to \mathrm{diag}(\boldsymbol{c}_{m,n}) \quad (6)$$

*where $\boldsymbol{c}_{m,n} = [c_{1|m,n}, c_{2|m,n}, \ldots, c_{M|m,n}]^\top \in \mathbb{R}^M$. Note that $\boldsymbol{c}_{m,n}^\top \mathbf{1} = 1$.*

By the data generation process (Sec. 3.2), given the next token $x_{T+1} = n$, the query token $x_T = m$ is uniquely determined. In the following, we just use $\boldsymbol{c}_n$ to represent $\boldsymbol{c}_{m,n}$ (and similar for $\tilde{\boldsymbol{c}}_n$).

## 4 Dynamics of $Y$

We first study the dynamics of $Y$. From Assumption 1(c), $Y$ learns much faster and we can treat the lower layer output (i.e., $X^\top \boldsymbol{b}_T$) as constant. From Lemma 2, when the sequence is long, we know

given the next token $x_{T+1} = n$, $X^\top \boldsymbol{b}_T$ becomes fixed. Therefore, the dynamics of $Y$ becomes:

$$\dot{Y} = \eta_Y \boldsymbol{f}_n (\boldsymbol{e}_n - \boldsymbol{\alpha}_n)^\top, \quad \boldsymbol{\alpha}_n = \frac{\exp(Y^\top \boldsymbol{f}_n)}{\mathbf{1}^\top \exp(Y^\top \boldsymbol{f}_n)} \tag{7}$$

Here $\boldsymbol{f}_n := \frac{X^\top \boldsymbol{b}_T}{\|X^\top \boldsymbol{b}_T\|_2} \rightarrow \frac{\boldsymbol{c}_n}{\|\boldsymbol{c}_n\|_2} \in \mathbb{R}^M$. Obviously $\|\boldsymbol{f}_n\|_2 = 1$ and $\boldsymbol{f}_n \geq 0$. Define $F = [\boldsymbol{f}_1, \ldots, \boldsymbol{f}_K]$. Since the vocabulary size $M$ typically is a huge number, and different sequence classes can cover diverse subset of vocabulary, we study the weak correlation case:

**Assumption 2** (Weak Correlations). *We assume $M \gg K^2$ and $\{\boldsymbol{f}_n\}_{n=1}^K$ satisfies $F^\top F = I + E$, where the eigenvalues of $E \in \mathbb{R}^{K \times K}$ satisfies $|\lambda_1| < \frac{1}{K}$ and $|\lambda_i(E)| \geq \frac{6}{\sqrt{M}}, \forall i \in [K]$.*

Assumption 2 means that $\boldsymbol{f}_n$ share some weak correlations and it immediately leads to the fact that $F^\top F$ is invertible and $F$ is column full-rank. Note that the critical point $Y^*$ of Eqn. 7 should satisfy that for any given $x_{T+1} = n$, we need $\boldsymbol{\alpha} = \boldsymbol{e}_n$. But such $Y^*$ must contain infinity entries due to the property of the exponential function in $\boldsymbol{\alpha}$ and we can not achieve $Y^*$ in finite steps. To analyze Eqn. 7, we leverage a *reparameterized* version of the dynamics, by setting $W = [\boldsymbol{w}_1, \ldots, \boldsymbol{w}_K]^\top := F^\top Y \in \mathbb{R}^{K \times M}$ and compute gradient update on top of $W$ instead of $Y$:

**Lemma 3.** *Given $x_{T+1} = n$, the dynamics of $W$ is (here $\boldsymbol{\alpha}_j = \exp(\boldsymbol{w}_j)/\mathbf{1}^\top \exp(\boldsymbol{w}_j)$):*

$$\dot{\boldsymbol{w}}_j = \eta_Y \mathbb{I}(j = n)(\boldsymbol{e}_n - \boldsymbol{\alpha}_n) \tag{8}$$

*While we cannot run gradient update on $W$ directly, it can be achieved by modifying the gradient of $Y$ to be $\dot{Y} = \eta_Y (\boldsymbol{f}_n - FE'\boldsymbol{e}_n)(\boldsymbol{e}_n - \boldsymbol{\alpha}_n)^\top$. If $\lambda_1$ is small, the modification is small as well.*

Please check Appendix D for the proof. Lemma 3 shows that for every fixed $n$, only the corresponding row of $W$ is updated, which makes the analysis much easier. We now can calculate the backpropagated gradient used in Eqn. 3.

**Theorem 1.** *If Assumption 2 holds, the initial condition $Y(0) = 0$, $M \gg 100$, $\eta_Y$ satisfies $M^{-0.99} \ll \eta_Y < 1$, and each sequence class appears uniformly during training, then after $t \gg K^2$ steps of batch size 1 update, given event $x_{T+1}[i] = n$, the backpropagated gradient $\boldsymbol{g}[i] := Y(\boldsymbol{x}_{T+1}[i] - \boldsymbol{\alpha}[i])$ takes the following form:*

$$\boldsymbol{g}[i] = \gamma \left( \iota_n \boldsymbol{f}_n - \sum_{n' \neq n} \beta_{nn'} \boldsymbol{f}_{n'} \right) \tag{9}$$

*Here the coefficients $\iota_n(t)$, $\beta_{nn'}(t)$ and $\gamma(t)$ are defined in Appendix with the following properties:*

- *(a) $\xi_n(t) := \gamma(t) \sum_{n \neq n'} \beta_{nn'}(t) \boldsymbol{f}_n^\top(t) \boldsymbol{f}_{n'}(t) > 0$ for any $n \in [K]$ and any $t$;*

- *(b) The speed control coefficient $\gamma(t) > 0$ satisfies $\gamma(t) = O(\eta_Y t/K)$ when $t \leq \frac{\ln(M) \cdot K}{\eta_Y}$ and $\gamma(t) = O\left(\frac{K \ln(\eta_Y t/K)}{\eta_Y t}\right)$ when $t \geq \frac{2(1+\delta') \ln(M) \cdot K}{\eta_Y}$ with $\delta' = \Theta(\frac{\ln \ln M}{\ln M})$.*

In the remark of Lemma 5 in Appendix, we analyze the original dynamics (Eqn. 7) with identical off-diagonal elements of $E$, and Theorem 1 still holds with a smaller effective learning rate.

## 5 The dynamics of Self-attention

Now we analyze the dynamics of self-attention logits $Z$, given the dynamics of upper layer $Y$.

**Lemma 4** (Self-attention dynamics). *With Assumption 1(b) (i.e., $T \rightarrow +\infty$), Eqn. 4 becomes:*

$$\dot{\boldsymbol{z}}_m = \eta_Z \gamma \sum_{n \in \psi^{-1}(m)} \text{diag}(\boldsymbol{f}_n) \sum_{n' \neq n} \beta_{nn'} (\boldsymbol{f}_n \boldsymbol{f}_n^\top - I) \boldsymbol{f}_{n'}, \tag{10}$$

Please check Appendix E for the proof. Now we study the dynamics of two types of contextual tokens (Sec. 3.2), namely *distinct tokens* (DT) which appear only for a single next token (i.e., $|\Omega(l)| = 1$ with $\Omega(l) := \{n : \mathbb{P}(l|n) > 0\}$), and *common tokens* (CT) that appear across multiple next tokens ($|\Omega(l)| > 1$). We show their fates are very different: over training, ***distinct tokens gain attention but common ones lose it***.

**Theorem 2** (Fates of contextual tokens). *Let $G_{CT}$ be the set of common tokens (CT), and $G_{DT}(n)$ be the set of distinct tokens (DT) that belong to next token $n$. Then if Assumption 2 holds, under the self-attention dynamics (Eqn. 10), we have:*

- *(a) for any distinct token $l \in G_{DT}(n)$, $\dot{z}_{ml} > 0$ where $m = \psi(n)$;*

- *(b) if $|G_{CT}| = 1$ and at least one next token $n \in \psi^{-1}(m)$ has at least one distinct token, then for the single common token $l \in G_{CT}$, $\dot{z}_{ml} < 0$.*

Now we know DTs grow and a single CT will shrink. For multiple CTs to shrink, the condition can be a bit involved (see Corollary 2 in Appendix E). The following theorem further shows that the growth rates of DTs critically depend on their initial conditions:

**Theorem 3** (Growth of distinct tokens). *For a next token $n$ and its two distinct tokens $l$ and $l'$, the dynamics of the **relative gain** $r_{l/l'|n}(t) := f_{nl}^2(t)/f_{nl'}^2(t) - 1 = \tilde{c}_{l|n}^2(t)/\tilde{c}_{l'|n}^2(t) - 1$ has the following analytic form (here the query token $m = \psi(n)$ and is uniquely determined by distinct token $l$):*

$$r_{l/l'|n}(t) = r_{l/l'|n}(0)e^{2(z_{ml}(t)-z_{ml}(0))} =: r_{l/l'|n}(0)\chi_l(t) \tag{11}$$

*where $\chi_l(t) := e^{2(z_{ml}(t)-z_{ml}(0))}$ is the **growth factor** of distinct token $l$. If there exist a dominant token $l_0$ such that the initial condition satisfies $r_{l_0/l|n}(0) > 0$ for all its distinct token $l \neq l_0$, and all of its common tokens $l$ satisfy $\dot{z}_{ml} < 0$. Then both $z_{ml_0}(t)$ and $f_{nl_0}(t)$ are monotonously increasing over $t$, and*

$$e^{2f_{nl_0}^2(0)B_n(t)} \leq \chi_{l_0}(t) \leq e^{2B_n(t)} \tag{12}$$

*here $B_n(t) := \eta_Z \int_0^t \xi_n(t')\mathrm{d}t'$. Intuitively, larger $B_n$ gives larger $r_{l_0/l|n}$ and sparser attention map.*

**Self-attention as an algorithm of token scanning**. From Eqn. 11, we could see that self-attention performs *token scanning*. To see that, consider the simplest initialization that $\boldsymbol{z}(0) = 0$, which means that $r_{l_0/l|n}(0) = \left(\frac{\mathbb{P}(l_0|m,n)}{\mathbb{P}(l|m,n)}\right)^2 - 1$. Therefore, distinct token $l$ with low conditional probability $\mathbb{P}(l|m,n)$ will have $r_{l_0/l|n}(0) \gg 0$, According Eqn. 11, this leads to quickly growing ratio $r_{l_0/l|n}(t)$, which means that the corresponding component $f_{nl}$ will be quickly dwarfed by the dominating component $f_{nl_0}$. On the other hand, token with high conditional probability $\mathbb{P}(l|m,n)$ will have smaller $r_{l_0/l|n}(0)$, and the ratio $r_{l_0/l|n}(t)$ grows slower, costing longer time for $l_0$ to dominate $l$.

**Initial value as prior information**. From the theorems, it is clear that the initial value $r_{l/l'|n}(0) := \left(\frac{\mathbb{P}(l|m,n)\exp(z_{ml}(0))}{\mathbb{P}(l'|m,n)\exp(z_{ml'}(0))}\right)^2 - 1$ critically determines the fate of the dynamics. Two tokens $l$ and $l'$ with comparable conditional probability $\mathbb{P}(l|m,n)$ and $\mathbb{P}(l'|m,n)$ can be suppressed in either way, depending on their initial logits $z_{ml}(0)$ and $z_{ml'}(0)$. In the empirical implementation, the initial value of the logits are determined by the inner products of independently initialized high-dimensional vectors, which fluctuate around zero.

The concept of "initial value as prior" can explain empirical design choices such as *multi-head self-attention* [1]. From this perspective, each head $h$ has its own $Z_h$ and is initialized independently, which could enable more diverse token combination (e.g., a combination of 1st, 3rd, 5th tokens, rather than a combination of 1st, 2nd, 3rd tokens).

## 6 The Moment of Snapping: When Token Combination is fixed

Theorem 3 suggests two possible fates of the self-attention weights: if $\xi_n(t)$ decays slowly (e.g., $\xi_n(t) \geq 1/t$), then $B_n(t) \to +\infty$ and all contextual tokens except for the dominant one will drop (i.e., $f_{nl} \to 0$) following the ranking order of their conditional probability $\mathbb{P}(l|m,n)$. Eventually, winner-takes-all happens. Conversely, if $\xi_n(t)$ drops so fast that $B_n(t)$ grows very slowly, or even has an upper limit, then the self-attention patterns are "snapped" and token combination is learned and fixed.

The conclusion is not obvious, since $\xi_n(t)$ depends on the decay rate of $\gamma(t)$ and $\beta_{nn'}(t)$, which in turns depends on the inner product $\boldsymbol{f}_n^\top(t)\boldsymbol{f}_{n'}(t)$, which is related to the logits of the common tokens that also decays over time.

Here we perform a qualitative estimation when there is only a single common token $l$ and every next token shares a single token $m$ (i.e., for any next token $n$, $\psi(n) = m$). We assume all normalization

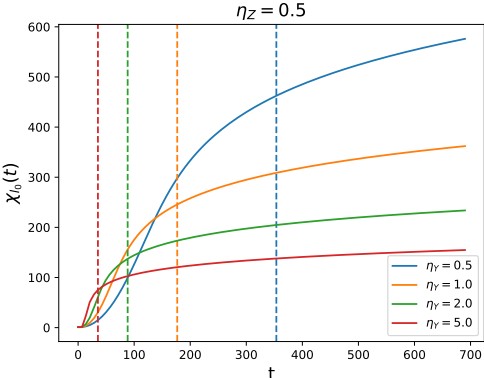

Figure 3: Growth factor $\chi_l(t)$ (Theorem 3) over time with fixed $\eta_Z = 0.5$ and changing $\eta_Y$. Each solid line is $\chi_l(t)$ and the dotted line with the same color corresponds to the transition time $t_0$ for a given $\eta_Y$.

terms in $\boldsymbol{f}_n$ are approximately constant, denoted as $\rho_0$, which means that $\boldsymbol{f}_n^\top \boldsymbol{f}_{n'} \approx \exp(2z_{ml})/\rho_0^2$ and $\beta_{nn'} \approx E'_{nn'} \approx \boldsymbol{f}_n^\top \boldsymbol{f}_{n'} \approx \exp(2z_{ml})/\rho_0^2$ as well, and $1 - \boldsymbol{f}_n^\top \boldsymbol{f}_{n'} \approx 1$ due to the fact that common token components are small, and will continue to shrink during training.

Under these approximations, its dynamics (Eqn. 10) can be written as follows (here $C_0 := \rho_0^4/K$):

$$\dot{z}_{ml} = \eta_Z \gamma \sum_{n \in \psi^{-1}(m)} f_{nl} \sum_{n' \neq n} \beta_{nn'}(f_{nl}^2 - 1)f_{nl'} \approx -C_0^{-1}\eta_Z\gamma e^{4z_{ml}}, \quad \xi_n(t) \approx C_0^{-1}\gamma e^{4z_{ml}} \quad (13)$$

Surprisingly, we now find a *phase transition* by combining the rate change of $\gamma(t)$ in Theorem 1:

**Theorem 4** (Phase Transition in Training). *If the dynamics of the single common token $z_{ml}$ satisfies $\dot{z}_{ml} = -C_0^{-1}\eta_Z\gamma(t)e^{4z_{ml}}$ and $\xi_n(t) = C_0^{-1}\gamma(t)e^{4z_{ml}}$, then we have:*

$$B_n(t) = \begin{cases} \frac{1}{4}\ln\left(C_0 + \frac{2(M-1)^2}{KM^2}\eta_Y\eta_Z t^2\right) & t < t'_0 := \frac{K\ln M}{\eta_Y} \\ \frac{1}{4}\ln\left(C_0 + \frac{2K(M-1)^2}{M^2}\frac{\eta_Z}{\eta_Y}\ln^2(M\eta_Y t/K)\right) & t \geq t_0 := \frac{2(1+o(1))K\ln M}{\eta_Y} \end{cases} \quad (14)$$

*As a result, there exists a* phase transition *during training:*

- ***Attention scanning**. At the beginning of the training, $\gamma(t) = O(\eta_Y t/K)$ and $B_n(t) \approx \frac{1}{4}\ln K^{-1}(\rho_0^4 + 2\eta_Y\eta_Z t^2) = O(\ln t)$. This means that the growth factor for dominant token $l_0$ is (sub-)linear: $\chi_{l_0}(t) \geq e^{2f_{nl_0}^2(0)B_n(t)} \approx [K^{-1}(\rho_0^4 + 2\eta_Y\eta_Z t^2)]^{0.5f_{nl_0}^2(0)}$, and the attention on less co-occurred token drops gradually.*

- ***Attention snapping**. When $t \geq t_0 := 2(1+\delta')K\ln M/\eta_Y$ with $\delta' = \Theta(\frac{\ln\ln M}{\ln M})$, $\gamma(t) = O\left(\frac{K\ln(\eta_Y t/K)}{\eta_Y t}\right)$ and $B_n(t) = O(\ln\ln t)$. Therefore, while $B_n(t)$ still grows to infinite, the growth factor $\chi_{l_0}(t) = O(\ln t)$ grows at a much slower logarithmic rate.*

See proof in Appendix F. This gives a few insights about the training process: **(a)** larger learning rate $\eta_Y$ of the decoder $Y$ leads to shorter phase transition time $t_0 \approx 2K\ln M/\eta_Y$, **(b)** scaling up both learning rate ($\eta_Y$ and $\eta_Z$) leads to larger $B_n(t)$ when $t \to +\infty$, and thus sparser attention maps, and **(c)** given fixed $\eta_Z$, small learning rate $\eta_Y$ leads to larger $B_n(t)$ when $t \geq t_0$, and thus sparser attention map. Fig. 3 shows numerical simulation results of the growth rate $\chi_l(t)$. Here we set $K = 10$ and $M = 1000$, and we find smaller $\eta_Y$ given fixed $\eta_Z$ indeed leads to later transition and larger $B_n(t)$ (and $\chi_l(t)$).

## 7 Discussion and Limitations

**Positional encoding**. While our main analysis does not touch positional encoding, it can be added easily following the relative encoding schemes that adds a linear bias when computing self attention (E.g., T5 [17], ALiBi [85], MusicTransformer [86]). More specifically, the added linear bias $\exp(z_{ml} + z_0) = \exp(z_{ml})\exp(z_0)$ corresponds to a prior of the contextual token to be learned in the self-attention layer.

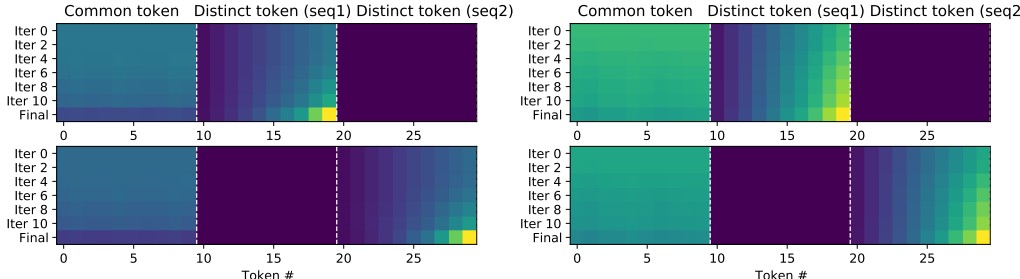

Figure 4: Visualization of $c_n$ ($n = 1, 2$) in the training dynamics of 1-layer Transformer using SGD on `Syn-Small` setting. Top row for query token $n = 1$ and bottom row for query token $n = 2$. **Left:** SGD training with $\eta_Y = \eta_Z = 1$. Attention pattern $c_n$ becomes sparse and concentrated on highest $\mathbb{P}(l|n)$ (rightmost) for each sequence class (Theorem 3). **Right:** SGD training with $\eta_Y = 10$ and $\eta_Z = 1$. With larger $\eta_Y$, convergence becomes faster but the final attention maps are less sparse (Sec. 6).

**Residue connection**. Residue connection can be added in the formulation, i.e., $\bar{u}_T = \text{LN}(\text{LN}(\tilde{u}_T) + u_{x_T})$, where $\tilde{u}_T$ is defined in Eqn. 1, and $\bar{u}_T$ is used instead in the objective (Eqn. 2). In this case, the $\beta_{nn'}$ in Theorem 1 now is approximately $\beta_{nn'} \sim f_n^\top f_{n'} + \mathbb{I}(\psi(n) = \psi(n'))$, which is much larger for sequence classes $n$ and $n'$ that share the same query token $x_T$ than otherwise. In this case, Theorem 1 now gives $g[i] = \gamma \left( \iota_n f_n - \sum_{n \neq n' \in \psi^{-1}(\psi(n))} \beta_{nn'} f_{n'} \right)$ for $x_{T+1}[i] = n$. Due to the additional constraint $n' \in \psi^{-1}(\psi(n))$ (i.e., $n$ and $n'$ shares the same query token), we can define *local* distinct and common tokens to be *within* the sequence class subset $\psi^{-1}(m)$ and Theorem 2 now applies within each subset. Empirically this makes more sense, since the query token $x_T = m_1$ or $m_2$ alone can already separate different subsets $\psi^{-1}(m_1)$ and $\psi^{-1}(m_2)$ and there should not be any interactions across the subsets. Here we just present the most straightforward analysis and leave this extension for future work.

**Possible future extension to multi-layer cases**. For multilayer training, a lasting puzzle is to explain how the input tokens get combined together to form high-level concepts. The analysis above shows that the training leads to sparse attention even among relevant tokens, and demonstrates that there is a priority in token combinations for 1-layer attention based on their co-occurrence: even if there are 10 relevant contextual tokens to the query, the self-attention may only pick 1-2 tokens to combine first due to attention sparsity. This can be regarded as a starting point to study how tokens are composed hierarchically. In comparison, [28, 29, 30] show that attention attends to all relevant tokens, which may not suggest a hierarchical / multi-layer architecture.

## 8 Experiments

We conduct experiments on both synthetic and real-world dataset to verify our theoretical findings.

**Syn-Small**. Following Sec. 3.2, we construct $K = 2$ sequence classes with vocabulary size $M = 30$. The first 10 tokens (0-9) are shared between classes, while the second and third 10 tokens (10-19 and 20-29) are distinct for class 1 and class 2, respectively. The conditional probability $\mathbb{P}(l|n)$ for tokens 10-19 is increasing monotonously (the same for 20-29). The 1-layer Transformer is parameterized with $Y$ and $Z$ (Sec. 3.1), is trained with initial condition $Y(0) = Z(0) = 0$ and SGD (with momentum 0.9) using a batchsize of 128 and sequence length $T = 128$ until convergence.

Fig. 4 shows the simulation results. The attention indeed becomes sparse during training, and increasing $\eta_Y$ with fixed $\eta_Z$ leads to faster convergence but less sparse attention. Both are consistent with our theoretical predictions (Theorem 3 and Sec. 6). Interestingly, if we use Adam optimizer instead, self-attention with different learning rate $\eta_Y = \eta_Z$ picks different subsets of distinct tokens to focus on, showing tune-able inductive bias (Fig. 5). We leave analysis on Adam for future work.

**Syn-Medium**. To further verify our theoretical finding, we now scale up $K$ to create `Syn-Medium` and compute how attention sparsity for distinct tokens (in terms of entropy) changes with the learning rates (Fig. 6). We can see indeed the entropy goes down (i.e., attention becomes sparser) with larger $\eta_Z$, and goes up (i.e., attention becomes less sparse) by fixing $\eta_Z$ and increasing $\eta_Y$ passing the threshold $\eta_Y/\eta_Z \approx 2$, consistent with Sec. 6. Note that the threshold is due to the fact that our theory is built on Assumption 1(c), which requires $\eta_Y$ to be reasonably larger than $\eta_Z$.

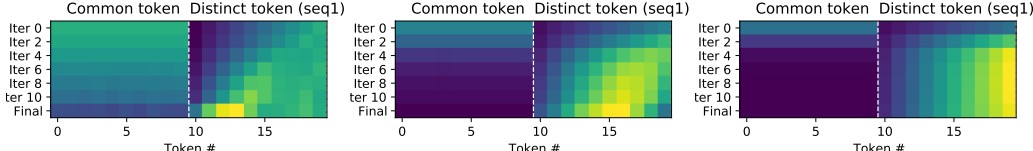

Figure 5: Visualization of (part of) $c_n$ for sequence class $n = 1$ in the training dynamics using Adam [22] on `Syn-Small` setting. **From left to right**: $\eta_V = \eta_Z = 0.1, 0.5, 1$. With different learning rate Adam seems to steer self-attention towards different subset of distinct tokens, showing tune-able inductive bias.

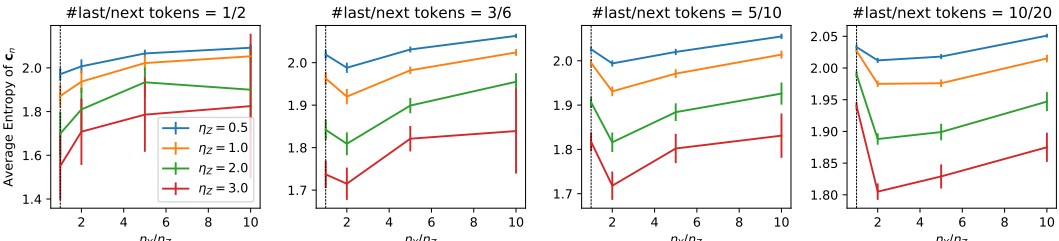

Figure 6: Average entropy of $c_n$ (Eqn. 5) on distinct tokens versus learning rate ratio $\eta_Y / \eta_Z$ with more query tokens $M$/next tokens $K$. We report mean values over 10 seeds and standard derivation of the mean.

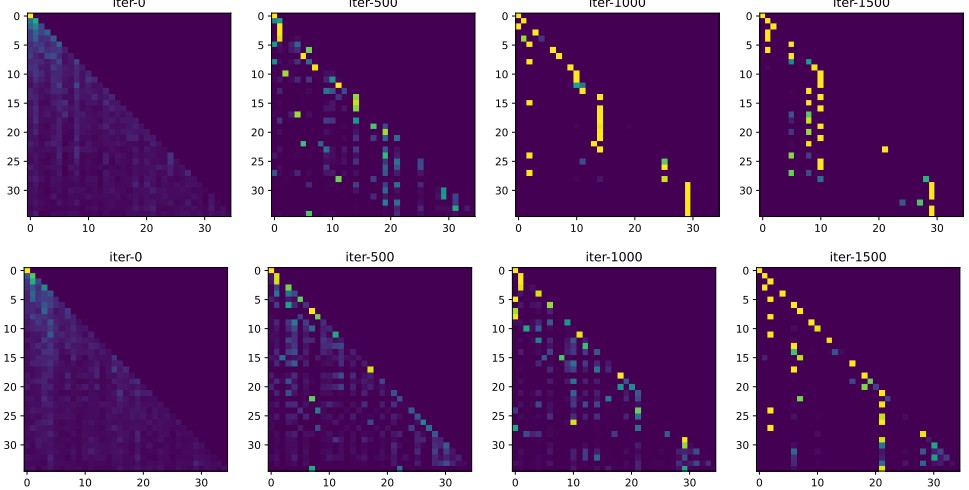

Figure 7: Attention patterns in the lowest self-attention layer for 1-layer (top) and 3-layer (bottom) Transformer trained on WikiText2 using SGD (learning rate is 5). Attention becomes sparse over training.

**Real-world Dataset**. We also test our finding on WikiText [25] using both 1-layer and multi-layer Transformers with regular parameterization that computes $Y$ and $Z$ with embedding $U$. In both cases, attentions of the first layer freeze (and become sparse) at some point (Fig. 7), even if the learning rate remains the same throughout training. More results are in Appendix G.

## 9    Conclusion and Future Work

In this work, we formally characterize SGD training dynamics of 1-layer Transformer, and find that the dynamics corresponds to a *scan and snap* procedure that progressively pays more attention to key tokens that are distinct and frequently co-occur with the query token in the training set. To our best knowledge, we are the first to analyze the attention dynamics and reveal its inductive bias on data input, and potentially open a new door to understand how Transformer works.

Many future works follow. According to our theory, large dataset suppresses spurious tokens that are perceived as distinct in a small dataset but are actual common ones. Our finding may help suppress such tokens (and spurious correlations) with prior knowledge, without a large amount of data.

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

| Basic Notations | |
|---|---|
| $M$ | Vocabulary size |
| $T$ | Sequence length |
| $\boldsymbol{e}_k$ | One-hot vector (1 at component $k$) |
| $X \in \mathbb{R}^{T-1 \times M}$ | Input sequence (of length $T-1$) |
| $\boldsymbol{b}_T \in \mathbb{R}^{T-1}$ | Vector of self-attention weights to predict the token at time $T$. |
| $\boldsymbol{x}_t \in \mathbb{R}^M$ | contextual token ($0 \le t \le T-2$) (one-hot) |
| $\boldsymbol{x}_{T-1} \in \mathbb{R}^M$ | Last/query token (one-hot) |
| $\boldsymbol{x}_T \in \mathbb{R}^M$ | Next token (class label) to be predicted (one-hot) |
| $\boldsymbol{x}_t[i] \in \mathbb{R}^M$ | $i$-th training sample of token at location $t$ in the sequence |
| $K$ | Number of possible choices the next token could take. |
| $\boldsymbol{\alpha}(t)$ | Softmax score of the output layer. |
| Learnable Parameters | |
| $Y \in \mathbb{R}^{M \times M}$ | decoder layer parameters |
| $Z \in \mathbb{R}^{M \times M}$ | self-attention logits |
| $\boldsymbol{z}_m$ | $m$-th row of $Z$ (i.e., attention logits for a query/query token $m$) |
| Hyperparameters | |
| $\eta_Y$ | Learning rate of the decoder layer |
| $\eta_Z$ | Learning rate of the self-attention layer |
| Token Types and Distribution | |
| $\psi(n)$ | Mapping from next token $x_T = n$ to its unique last/query token |
| $\psi^{-1}(m)$ | The subset of next tokens for last/query token $x_{T-1} = m$ |
| $\mathbb{P}(l\|m,n)$ | Conditional probability of contextual token $l$ |
| | given query token is $m$ and next token to be predicted as $n$ |
| $G_{\mathrm{CT}}$ | Subset of common tokens |
| $G_{\mathrm{DT}}(n)$ | Subset of distinct tokens for $x_T = n$ |
| Attention Score | |
| $\tilde{\boldsymbol{c}}_n \in \mathbb{R}^M$ | Unnormalized attention score given next token $x_T = n$ |
| $\boldsymbol{c}_n \in \mathbb{R}^M$ | $\ell_1$-normalized attention score given next token $x_T = n$ |
| $\boldsymbol{f}_n \in \mathbb{R}^M$ | $\ell_2$-normalized attention score given next token $x_T = n$ |
| $\boldsymbol{g} \in \mathbb{R}^M$ | Back-propagated gradient for $\boldsymbol{f}_n$ |
| $F$ | Input matrix of the decoder layer. Each column of $F$ is $\boldsymbol{f}_n$ |
| Self-attention dynamics | |
| $r_{l'/l\|n}(t)$ | Relative gain between distinct token $l$ and $l'$ for next token $n$ |
| $B_n(t)$ | Growth factor bound of the relative gain |
| $\gamma(t)$ | Speed control coefficient |

Table 1: Overall notation table of the main symbols in the paper.

# A    Notation Table

Tbl. 1 gives the notation of the main quantities in the paper.

# B    Detailed comparison with the concurrent works

## B.1    Comparison with [28]

**Setting, Assumptions and Conclusions**. [28] analyzes the SGD convergence of 1-layer ViT model (1 layer self-attention + 2 layer FFN with ReLU, with the top layer of FFN fixed as random, token embedding fixed). Under a specific binary data model in which the data label is determined by counting the number of tokens that belong to positive/negative pattern, [28] gives a generalization bound when the number of hidden nodes in FFN is large, and at the same time, shows that the self-attention attends to relevant tokens and becomes sparse (if number of relevant tokens are small).

In comparison, our work focuses on language models, assume broader data distribution (e.g., multiple classes, arbitrary conditional probability of token given class label) and incorporate LayerNorm

naturally. We propose more detailed quantitative properties, e.g., attention sparsity even among relevant tokens, two-stage evolution of attention scores, with a much simpler analysis.

**Techniques**. The techniques used in [28] are based on feature learning techniques applied to MLP (e.g., [87]). It identifies lucky neurons if the number of hidden neurons is large enough. In comparison, our framework and analysis is much simpler by leveraging that certain nonlinear continuous dynamics systems can be integrated out analytically to yield clean solutions (e.g., Theorem 3 (Eqn. 11) and Theorem 4 (Eqn. 128)), avoiding complicated bounds in [28]. This allows us to characterize the converging behavior of self-attentions when $t \to +\infty$.

### B.2 Comparison with [29]

[29] focuses on 1-layer attention-based prompt-tuning, in which some parameters of the models are fixed ($W_p$, $W_q$). The analysis focuses on the initial (3x one-step) SGD trajectory, and constructs the dataset model containing specific context-relevant/context-irrelevant data, and the context-vector indicates the token relevance. As a result, [29] shows the attention becomes sparse (i.e., attending to context-relevant tokens) over time, which is consistent with ours, and shows that prompt-attention can find the relevant tokens and achieve high accuracy while self-attention/linear-attention can't.

In comparison, our work goes beyond the 2-classes model and further points out that the attention weight will be relevant to the conditional probability of the contextual tokens, which is more detailed than the sparse attention result in [29] that relies on the sparsity assumption of contextual tokens itself. We also focus on the pre-training stage (training from scratch, predicting the next token), characterize the entire trajectory under SGD for the self-attention layer, in particular its converging behavior.

### B.3 Comparison with [30]

Compared to [29], [30] also analyzes the dynamics of the query-key matrix and the embedding of a single tunable token (often `[CLS]` token). It makes connection between the binary classification problem with 1-layer transformer and max-margin SVM formulation, when the tokens are linearly separable. The dynamics is characterized completely, which is nice. Note here is not an attention since its norm can be shown to go to infinity over training.

In comparison, our work does not learn the embedding of an individual token, but focuses on the dynamics of (all-pair) attention scores during training. We also work on multiple-class setup and do not explicitly assume the linear separability among classes.

## C   Proof of Section 3

**Lemma 1** (Dynamics of 1-layer Transformer). *The gradient dynamics of Eqn. 2 with batchsize 1 is:*

$$\dot{Y} = \eta_Y \mathrm{LN}(X^\top \boldsymbol{b}_T)(\boldsymbol{x}_{T+1} - \boldsymbol{\alpha})^\top, \quad \dot{Z} = \eta_Z \boldsymbol{x}_T (\boldsymbol{x}_{T+1} - \boldsymbol{\alpha})^\top Y^\top \frac{P^\perp_{X^\top \boldsymbol{b}_T}}{\|X^\top \boldsymbol{b}_T\|_2} X^\top \mathrm{diag}(\boldsymbol{b}_T) X \quad (3)$$

*Here $P^\perp_{\boldsymbol{v}} := I - \boldsymbol{v}\boldsymbol{v}^\top / \|\boldsymbol{v}\|_2^2$ projects a vector into $\boldsymbol{v}$'s orthogonal complementary space, $\eta_Y$ and $\eta_Z$ are the learning rates for the decoder layer $Y$ and self-attention layer $Z$, $\boldsymbol{\alpha} := [\alpha_1, \ldots, \alpha_M]^\top \in \mathbb{R}^M$ and $\alpha_m := \exp(Y^\top \mathrm{LN}(X^\top \boldsymbol{b}_T))/\mathbf{1}^\top \exp(Y^\top \mathrm{LN}(X^\top \boldsymbol{b}_T))$.*

*Proof.* With the reparameterization of $Y$ and $Z$, the loss function is the following:

$$J(Y, Z) = \mathbb{E}_{\mathcal{D}} \left[ \boldsymbol{x}_{T+1}^\top Y^\top \mathrm{LN}(X^\top \boldsymbol{b}_T) - \log(\mathbf{1}^\top \exp(Y^\top \mathrm{LN}(X^\top \boldsymbol{b}_T))) \right] \quad (15)$$

and

$$\alpha_m = \frac{\exp(\boldsymbol{e}_m^\top Y^\top \mathrm{LN}(X^\top \boldsymbol{b}_T))}{\mathbf{1}^\top \exp(Y^\top \mathrm{LN}(X^\top \boldsymbol{b}_T))} \quad (16)$$

Therefore, taking matrix differentials, we have:

$$\mathrm{d}J = (\boldsymbol{x}_{T+1} - \boldsymbol{\alpha})^\top \mathrm{d}(Y^\top \mathrm{LN}(X^\top \boldsymbol{b})) = (\boldsymbol{x}_{T+1} - \boldsymbol{\alpha})^\top \left( \mathrm{d}Y^\top \mathrm{LN}(X^\top \boldsymbol{b}) + Y^\top \frac{P^\perp_{X^\top \boldsymbol{b}}}{\|X^\top \boldsymbol{b}\|} X^\top \mathrm{d}\boldsymbol{b} \right)$$
$$(17)$$

since in general we have $\mathrm{d}(\exp(\boldsymbol{a})/\mathbf{1}^\top \exp(\boldsymbol{a})) = L\mathrm{d}\boldsymbol{a}$ with $L := \mathrm{diag}(\boldsymbol{b}) - \boldsymbol{b}\boldsymbol{b}^\top$, let $\boldsymbol{a} := XZ^\top \boldsymbol{x}_T$ and we have:

$$\mathrm{d}J \;=\; (\boldsymbol{x}_{T+1} - \boldsymbol{\alpha})^\top \left( \mathrm{d}Y^\top \mathrm{LN}(X^\top \boldsymbol{b}) + Y^\top \frac{P_{X^\top \boldsymbol{b}}^\perp}{\|X^\top \boldsymbol{b}\|} X^\top L\mathrm{d}(XZ^\top \boldsymbol{x}_T) \right) \tag{18}$$

$$\;=\; (\boldsymbol{x}_{T+1} - \boldsymbol{\alpha})^\top \left( \mathrm{d}Y^\top \mathrm{LN}(X^\top \boldsymbol{b}) + Y^\top \frac{P_{X^\top \boldsymbol{b}}^\perp}{\|X^\top \boldsymbol{b}\|} X^\top LX\mathrm{d}Z^\top \boldsymbol{x}_T \right) \tag{19}$$

Finally notice that $P_{X^\top \boldsymbol{b}}^\perp X^\top L = P_{X^\top \boldsymbol{b}}^\perp X^\top \mathrm{diag}(\boldsymbol{b})$ due to the fact that $P_{\boldsymbol{v}}^\perp \boldsymbol{v} = 0$ and the conclusion follows. $\qquad\square$

**Lemma 2.** *Given the event $\{x_T = m, x_{T+1} = n\}$, when $T \to +\infty$, we have*

$$X^\top \boldsymbol{b}_T \to \boldsymbol{c}_{m,n}, \qquad\qquad X^\top \mathrm{diag}(\boldsymbol{b}_T)X \to \mathrm{diag}(\boldsymbol{c}_{m,n}) \tag{6}$$

*where $\boldsymbol{c}_{m,n} = [c_{1|m,n}, c_{2|m,n}, \ldots, c_{M|m,n}]^\top \in \mathbb{R}^M$. Note that $\boldsymbol{c}_{m,n}^\top \mathbf{1} = 1$.*

*Proof.* Let $\boldsymbol{p} = [\exp(z_{m1}), \ldots, \exp(z_{mM})]^\top \in \mathbb{R}^M$, $p_{x_t} := \exp(z_{mx_t})$, and $\boldsymbol{p}_X := [\exp(z_{mx_1}), \ldots, \exp(z_{mx_{T-1}})]^\top$, then for any $T$ we have

$$X^\top \boldsymbol{b}_T = \sum_{t=1}^{T-1} b_{tT} \boldsymbol{x}_t = \sum_{t=1}^{T-1} \frac{p_{x_t} \boldsymbol{x}_t}{\sum_{t'} p_{x_{t'}}} = \frac{X^\top \boldsymbol{p}_X}{\mathbf{1}^\top X^\top \boldsymbol{p}_X} \tag{20}$$

Combining Lemma 18 and the definition of $c_{l|m,n}$ (Eqn. 5), we have that when $T \to +\infty$,

$$X^\top \boldsymbol{b}_T \to \sum_{l=1}^{M} \frac{\mathbb{P}(l|m,n)\exp(z_{ml})\boldsymbol{e}_l}{\sum_{l'}\mathbb{P}(l'|m,n)\exp(z_{ml'})} = \boldsymbol{c}_{m,n} \tag{21}$$

Similarly:

$$X^\top \mathrm{diag}(\boldsymbol{b}_T)X = \frac{X^\top \mathrm{diag}(\boldsymbol{p}_X)X}{\mathbf{1}^\top X^\top \boldsymbol{p}_X} \tag{22}$$

Let $T \to +\infty$, then we also get

$$X^\top \mathrm{diag}(\boldsymbol{b}_T)X \to \mathrm{diag}(\boldsymbol{c}_{m,n}) \tag{23}$$

$\qquad\square$

# D    Proof of Section 4

## D.1    Notation

For convenience, we introduce the following notations for this section:

- Denote $E' := (I + E)^{-1} - I$.

- Apply orthogonal diagonalization on $E$ and obtain $E = U^\top DU$ where $U := [\boldsymbol{u}_1, ..., \boldsymbol{u}_K] \in O_{K \times K}$, $D = \mathrm{diag}(\lambda_1, ..., \lambda_K)$ and $|\lambda_1| \geq ... \geq |\lambda_K| \geq 0$.

- Denote $F' := [F, F^\circ] \in \mathbb{R}^{M \times M}$ where $F^\circ \in \mathbb{R}^{M \times (M-K)}$ is some matrix such that $\mathrm{rank}(F') = M$. This is possible since $\{\boldsymbol{f}_i\}_{i \in [K]}$ are linear-independent.

- Denote $W' := (F')^\top Y = [F, F^\circ]^\top Y = [W^\top, Y^\top F^\circ]^\top = [\boldsymbol{w}_1, \ldots, \boldsymbol{w}_K, \boldsymbol{w}_{K+1}, \ldots, \boldsymbol{w}_M]^\top \in \mathbb{R}^{M \times M}$.

- Denote $\boldsymbol{\zeta}_n := \frac{M}{M-1}(\boldsymbol{e}_n - \frac{1}{M}\mathbf{1}) \in \mathbb{R}^M$.

- Denote $q_1 := \boldsymbol{\zeta}_i^\top \boldsymbol{\zeta}_i = 1 + \frac{1}{M-1}$, $q_0 := \boldsymbol{\zeta}_j^\top \boldsymbol{\zeta}_i = -\frac{M}{(M-1)^2}$ where $i, j \in [M], i \neq j$.

- Denote $h$ to be a continuous function that satisfies $h(0) = 0$ and $\dot{h} = \eta_Y \cdot (M - 1 + \exp(Mh))^{-1}$. Details in Lemma 6.

- Denote $\omega_1$ to be the constant defined in Lemma 8 that satisfies $\omega_1 = \Theta(\frac{\ln\ln(M)}{\ln(M)})$.

- Denote $N_n := \sum_{i=1}^N \mathbb{I}[x_{T+1} = n]$ to be the number of times the event $x_{T+1} = n$ happens.

- Denote $\bar{N} := \lceil N/K \rceil$ to be the average value of $N_n$ when $\mathbb{P}(n) \equiv 1/K$ and $\Delta := \lceil \sqrt{N \ln(\frac{1}{\delta})} \rceil$ to be the radius of confidence interval centered on $\bar{N}$ with confidence $1 - \delta$. Here $\Delta/\bar{N} \asymp \frac{K}{\sqrt{N}}\sqrt{\ln(\frac{1}{\delta})} \ll 1$ since $N \gg K^2$. Details in Lemma 10 and Remark 4.

- Denote $\bar{W}'(N) := [\bar{\boldsymbol{w}}_1(N), ..., \bar{\boldsymbol{w}}_K(N), \boldsymbol{0}, ..., \boldsymbol{0}]^\top \in \mathbb{R}^{M \times M}$, where $\bar{\boldsymbol{w}}_n(N) := (M - 1)h(\bar{N})\boldsymbol{\zeta}_n, \ \forall n \in [K]$.

## D.2 Proof of Lemma 3

We assume $\cup_{m \in [M]}\psi^{-1}(m) = [K]$ for convenience, but we claim that our proof can be easily generalized into the case where $\Omega \neq [K]$ by reordering the subscript of the vectors. First, we prove the dynamics equation of the reparameterized dynamics of $Y$.

**Lemma 3.** *Given $x_{T+1} = n$, the dynamics of $W$ is (here $\boldsymbol{\alpha}_j = \exp(\boldsymbol{w}_j)/\boldsymbol{1}^\top \exp(\boldsymbol{w}_j)$):*

$$\dot{\boldsymbol{w}}_j = \eta_Y \mathbb{I}(j = n)(\boldsymbol{e}_n - \boldsymbol{\alpha}_n) \tag{8}$$

*While we cannot run gradient update on $W$ directly, it can be achieved by modifying the gradient of $Y$ to be $\dot{Y} = \eta_Y(\boldsymbol{f}_n - FE'\boldsymbol{e}_n)(\boldsymbol{e}_n - \boldsymbol{\alpha}_n)^\top$. If $\lambda_1$ is small, the modification is small as well.*

*Proof.* We let $F' := [F, F^\circ] \in \mathbb{R}^{M \times M}$ where $\mathrm{rank}(F') = M$, this is possible since $\{\boldsymbol{f}_n\}_{n \in [K]}$ are linear-independent. And we further define $W' := (F')^\top Y = [F, F^\circ]^\top Y = [W^\top, Y^\top F^\circ]^\top = [\boldsymbol{w}_1, \ldots, \boldsymbol{w}_K, \boldsymbol{w}_{K+1}, \ldots, \boldsymbol{w}_M]^\top \in \mathbb{R}^{M \times M}$. When given $x_{T+1} = n$, the first term of the differential of loss function $J$ is:

$$\mathrm{tr}\left(\mathrm{d}Y^\top \frac{X^\top \boldsymbol{b}_T}{\|X^\top \boldsymbol{b}_T\|_2}(\boldsymbol{x}_{T+1} - \boldsymbol{\alpha})^\top\right) = \mathrm{tr}(\mathrm{d}Y^\top F'(F')^{-1}\boldsymbol{f}_n(\boldsymbol{x}_{T+1} - \boldsymbol{\alpha})^\top)$$
$$= \mathrm{tr}(\mathrm{d}(W')^\top \boldsymbol{e}_n(\boldsymbol{x}_{T+1} - \boldsymbol{\alpha})^\top) \tag{24}$$

So $\dot{W}' = \boldsymbol{e}_n(\boldsymbol{x}_{T+1} - \boldsymbol{\alpha})^\top$. This nice property will limit $W$ to independently update its $n$-th row for any $x_{T+1} = n \in [K]$, and the last $M - K$ rows of $W'$ are not updated. Similarly for $\boldsymbol{\alpha}$ we have

$$\boldsymbol{\alpha} = \frac{\exp(UW_V\tilde{\boldsymbol{u}}_T)}{\boldsymbol{1}^\top \exp(UW_V\tilde{\boldsymbol{u}}_T)} = \frac{\exp(Y^\top \boldsymbol{f}_n)}{\boldsymbol{1}^\top \exp(Y^\top \boldsymbol{f}_n)} = \frac{\exp(Y^\top F'(F')^{-1}\boldsymbol{f}_n)}{\boldsymbol{1}^\top \exp(Y^\top F'(F')^{-1}\boldsymbol{f}_n)} = \frac{\exp(\boldsymbol{w}_n)}{\boldsymbol{1}^\top \exp(\boldsymbol{w}_n)} \tag{25}$$

We get Eqn. 8 by combining the above results.

If we don't run gradient update on $W$ directly, we can run a modified gradient update on $Y$:

$$\dot{Y} = \eta_Y(\boldsymbol{f}_n - FE'\boldsymbol{e}_n)(\boldsymbol{e}_n - \boldsymbol{\alpha}_n)^\top \tag{26}$$

This will lead to (note that $F$ does not change over time due to Assumption 1 (c)):

$$\dot{W} = F^\top \dot{Y} = \eta_Y F^\top(\boldsymbol{f}_n - FE'\boldsymbol{e}_n)(\boldsymbol{e}_n - \boldsymbol{\alpha}_n)^\top \tag{27}$$
$$= \eta_Y\left[F^\top \boldsymbol{f}_n - F^\top F(I - (I + E)^{-1})\boldsymbol{e}_n\right](\boldsymbol{e}_n - \boldsymbol{\alpha}_n)^\top \tag{28}$$
$$= \eta_Y\left(F^\top \boldsymbol{f}_n - F^\top F\boldsymbol{e}_n + \boldsymbol{e}_n\right)(\boldsymbol{e}_n - \boldsymbol{\alpha}_n)^\top \tag{29}$$
$$= \eta_Y \boldsymbol{e}_n(\boldsymbol{e}_n - \boldsymbol{\alpha}_n)^\top \tag{30}$$

By Lemma 17, we know that if $\lambda_1$ is small, so does $\max_{i \in [K]}|\lambda_i(E')|$ and thus the modification is small as well. In Lemma 5 Remark 1, we will show that the additional term $-FE'\boldsymbol{e}_n$ effectively reduces the learning rate, if all off-diagonal elements of $E$ are the same. $\qquad \square$

Lemma 3 shows that we can transfer the problem into solving $K$ independent and similar non-linear ODE. And we then show that such a problem can be well solved by following Lemma. Recall that $\boldsymbol{\zeta}_n := \frac{M}{M-1}(\boldsymbol{e}_n - \frac{1}{M}\boldsymbol{1}) \in \mathbb{R}^M$, we have:

**Lemma 5.** *Assume $Y$ is initialized to be a zero matrix, $Z$ is fixed, and the learning rate of $Y$ is $\eta_Y$. Then if event $x_{T+1} = n$ always holds at $s$ step ($s \geq 1$) we have*

$$\boldsymbol{w}_n(s) = (M-1)h^*(s)\boldsymbol{\zeta}_n \tag{31}$$

$$\alpha_{nj}(s) = \begin{cases} \dfrac{\exp(Mh^*(s-1))}{(M-1)+\exp(Mh^*(s-1))} & , \quad j = n \\ \dfrac{1}{(M-1)+\exp(Mh^*(s-1))} & , \quad j \neq n \end{cases} \tag{32}$$

*And thus $\boldsymbol{e}_n - \boldsymbol{\alpha}_n(s) = \frac{M-1}{M-1+\exp(Mh^*(s-1))}\boldsymbol{\zeta}_n$. Here $h^*(s)$ satisfies:*

$$h^*(s) = \begin{cases} h^*(s-1) + \dfrac{\eta_Y}{(M-1)+\exp(Mh^*(s-1))} & , \quad s \geq 1 \\ 0 & , \quad s = 0 \end{cases} \tag{33}$$

*Proof.* We prove this Lemma by induction.

**Step 1**: Note that $Y$ is initialized to be a zero matrix, then $\boldsymbol{w}_i(0) = 0, \forall i \in [K]$. So we have

$$\alpha_n(1) = \frac{1}{M}, \quad \forall j \in [K] \tag{34}$$

$$\dot{w}_{nj}(1) = \begin{cases} 1 - \dfrac{1}{M}, & j = n \\ -\dfrac{1}{M}, & j \neq n \end{cases} \tag{35}$$

$$w_{nj}(1) = \begin{cases} \eta_Y(1 - \dfrac{1}{M}), & j = n \\ -\dfrac{\eta_Y}{M}, & j \neq n \end{cases} \tag{36}$$

It's easy to check that these equations match that of Lemma 5.

**Step $s$**: Assume the equations of Lemma 5 hold for step $s-1$. Then at the $s$ step, we have

$$\alpha_{nj}(s) = \begin{cases} \dfrac{\exp((M-1)h^*(s-1))}{\exp((M-1)h^*(s-1)) + (M-1)\exp(-h^*(s-1))} = \dfrac{\exp(Mh^*(s-1))}{\exp(Mh^*(s-1)) + (M-1)}, & j = n \\ \dfrac{\exp(-h^*(s-1))}{\exp((M-1)h^*(s-1)) + (M-1)\exp(-h^*(s-1))} = \dfrac{1}{\exp(Mh^*(s-1)) + (M-1)}, & j \neq n \end{cases} \tag{37}$$

$$\dot{w}_{nj}(s) = \begin{cases} \dfrac{M-1}{\exp(Mh^*(s-1)) + (M-1)}, & j = n \\ -\dfrac{1}{\exp(Mh^*(s-1)) + (M-1)}, & j \neq n \end{cases} \tag{38}$$

$$w_{nj}(s) = \begin{cases} (M-1)\cdot(\dfrac{\eta_Y}{\exp(Mh^*(s-1)) + (M-1)} + h^*(s-1)) \quad =(M-1)h^*(s), & j = n \\ -(\dfrac{\eta_Y}{\exp(Mh^*(s-1)) + (M-1)} + h^*(s-1)) \qquad\quad = -h^*(s), & j \neq n \end{cases} \tag{39}$$

And the equations of Lemma 5 also hold for step $s$. So we finish the proof. $\square$

**Remark 1.** *If we following the original dynamics (Eqn. 7), then it corresponds to the $W$ dynamics as follows:*

$$\dot{W} = \eta_Y(\boldsymbol{e}_n + (I+E)E'\boldsymbol{e}_n)(\boldsymbol{e}_n - \boldsymbol{\alpha}_n)^\top = \eta_Y F^\top \boldsymbol{f}_n(\boldsymbol{e}_n - \boldsymbol{\alpha}_n)^\top \tag{40}$$

*When all off-diagonal elements of $E$ are identical, i.e., $\boldsymbol{f}_n^\top \boldsymbol{f}_{n'} = \rho$ for $n \neq n'$, then $0 \leq \rho \leq 1$ and we have*

$$\dot{w}_n = \eta_Y(\boldsymbol{e}_n - \boldsymbol{\alpha}_n)^\top \tag{41}$$

$$\dot{w}_j = \eta_Y\rho(\boldsymbol{e}_n - \boldsymbol{\alpha}_n)^\top, \quad j \neq n \tag{42}$$

*So if different sequence classes are sampled uniformly, then by similar induction argument, we will have*

$$\boldsymbol{w}_n(N) = (M-1)h^*(N/K) \left[ \boldsymbol{\zeta}_n + \rho \sum_{n' \neq n} \boldsymbol{\zeta}_{n'} \right] = (1-\rho)(M-1)h^*(N/K)\boldsymbol{\zeta}_n \quad (43)$$

*where the last equation is due to the fact that $\sum_n \boldsymbol{\zeta}_n = \frac{M}{M-1} \sum_n \left( \boldsymbol{e}_n - \frac{1}{M}\boldsymbol{1} \right) = \frac{M}{M-1}(\boldsymbol{1}-\boldsymbol{1}) = 0$. This means that $\sum_{n' \neq n} \boldsymbol{\zeta}_{n'} = -\boldsymbol{\zeta}_n$. Therefore, the effective learning rate is $\eta'_Y := (1-\rho)\eta_Y \leq \eta_Y$.*

### D.3  Property of $h^*(s)$ and its continuous counterpart.

Before further investigation on $Y$, we need to get some basic properties of $h^*$, in particular, how fast it grows over time. First, if we consider the continuous version of $h^*$, namely $h$, then we can directly obtain the equation that $h$ needs to satisfy by integrating the corresponding differential equation.

**Lemma 6.** *If we consider the continuous version of $h^*(s)$, namely $h$, as the following ODE:*

$$\frac{dh}{dt} = \frac{\eta_Y}{(M-1)+\exp(Mh)} \quad (44)$$

*and assume $h(0) = 0$, then we have*

$$\exp(Mh(t)) + (M-1)Mh(t) = M\eta_Y t + 1 \quad (45)$$

$\square$

Then we will show that the $h$ is actually almost the same as the original step function $h^*$.

**Lemma 7.** *For $h$ and $h^*$ we have:*

- *(a) For any $s \in \mathbb{N}, 0 \leq h^*(s) - h(s) \leq \frac{2\eta_Y}{M}$. Then there exists some constant $c = \Theta(1)$ such that for any $s \leq \ln(M)/\eta_Y$, $h(s+c) \geq h^*(s) \geq h(s)$.*

- *(b) $h^*(s) - h(s) \to 0$ when $s \to +\infty$.*

*Proof.* **(a)** First we show that $h^*(s) \geq h(s)$ for all $s \in \mathbb{N}$, and the convex packet function of $h^*$ can almost control the upper bound of $h$. Define $h^\circ : \mathbb{R}^+ \to \mathbb{R}^+$ as follows:

$$h^\circ(t) := (t - \lfloor t \rfloor) \cdot [h^*(\lceil t \rceil) - h^*(\lfloor t \rfloor)] + h^*(\lfloor t \rfloor), \ \forall t \in \mathbb{R}^+ \quad (46)$$

Here $\lceil \cdot \rceil$ and $\lfloor \cdot \rfloor$ mean ceil function and floor function, respectively. It's clear that $h^\circ$ is a strictly monotonically increasing function, and for any $s \in \mathbb{N}$, $h^\circ(s) = h^*(s)$, while for any $t \notin \mathbb{N}$, $(t, h^\circ(t))$ lies on the line connecting point $(\lfloor t \rfloor, h^*(\lfloor t \rfloor))$ and point $(\lceil t \rceil, h^*(\lceil t \rceil))$. To prevent ambiguity, we let $\dot{h}^\circ(t)$ to be the left limit of $h^\circ$, i.e., $\dot{h}^\circ(t) = \lim_{t' \to t-} \dot{h}^\circ(t')$.

We claim $h(t) \leq h^\circ(t), \ \forall t \in \mathbb{R}^+$. We prove it by induction. First when $t = 0$, we have $h^\circ(0) = h^*(0) = h(0) = 0$. Then we assume $h(t') \leq h^\circ(t')$ hold for time $t' \leq t \in \mathbb{N}$ and prove that $h(t') \leq h^\circ(t')$ hold for $t' \in (t, t+1]$. If this is not true, then from the continuity of $h^\circ$ and $h$, we know it must exist $t'' \in (t, t+1]$ such that $h(t'') \geq h^\circ(t'')$ and $\dot{h}(t'') > \dot{h}^\circ(t'')$. The later condition results that $\eta_Y[M - 1 + \exp(Mh(t''))]^{-1} > \eta_Y[M - 1 + \exp(Mh^*(\lfloor t'' \rfloor))]^{-1}$. So

$$h(t'') < h^*(\lfloor t'' \rfloor) = h^\circ(\lfloor t'' \rfloor) \leq h^\circ(t'') \quad (47)$$

This contradicts the hypothesis $h(t'') \geq h^\circ(t'')$. So $h(t') \leq h^\circ(t')$ hold for $t' \in (t, t+1]$ and thus for all $t \in \mathbb{R}^+$. Hence for any $s \in \mathbb{N}$, we have $h(s) \leq h^\circ(s) = h^*(s)$. Actually, we can use the similar method to prove that $h(s) < h^*(s)$ for any $s \in \mathbb{N}^+$.

Then we show $h^*(s) - h(s) \leq 2\eta_Y/M$ by proving that for any $s \in \mathbb{N}^+$, $h(s)$ must meet at least one of the following two conditions:

**(i)** $h(s) \in [h^*(s-1), h^*(s)]$.

**(ii)** $h^*(s) - h(s) < h^*(s-1) - h(s-1)$.

If (i) doesn't hold, then we have for any $t \in [s-1, s), h(t) \leq h(s) < h^*(s-1) = h^\circ(s-1)$, which results that $\dot{h}(t) > \dot{h}^\circ(t)$ for all $t \in [s-1, s)$. Therefore, $h^*(s) - h^*(s-1) = h^\circ(s) - h^\circ(s-1) < h(s) - h(s-1)$ and thus $h(s)$ meets condition (ii). It's clear that $h(0)$ and $h(1)$ meet (i).

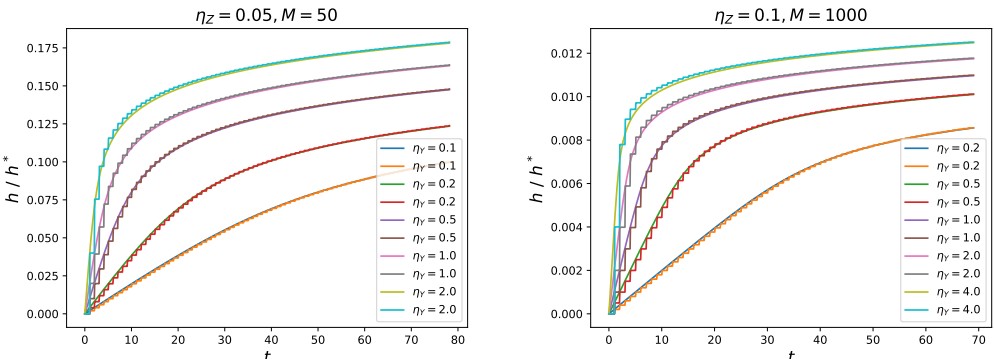

Figure 8: Numerical simulation of $h^*$ and $h$ with changing $\eta_Y$. The stepped folded line represents $h^*$ and the smooth curve represents $h$. The gap between $h^*$ and $h$ is bounded and goes to zero when time grows.

These two conditions mean that the gap between $h^*$ and $h$ will not grow if $h(s)$ is smaller than $h^*(s-1)$. Then for all $h(s)$ that meet (i), we have $h^*(s) - h(s) \leq h^*(s) - h^*(s-1) \leq h^*(1) - h^*(0) = \eta_Y/M$ from Eqn. 33. And for any $s \geq 2$, every time $h(s)$ transfer from (i) to (ii) exactly at $s$, which means that $h(s-1)$ meets (i) and thus no smaller than $h^*(s-2)$, we get $h^*(s) - h(s) \leq h^*(s) - h(s-1) \leq h^*(s) - h^*(s-2) \leq h^*(2) - h^*(0) \leq 2\eta_Y/M$.

Finally from Eqn. 53 in Lemma 9, when $s \leq \frac{\ln M}{\eta_Y}$, we get $h(s) = \Theta(\eta_Y t/M)$ and thus there exist some constant $c = \Theta(1)$ such that $h(s+c) \geq h(s) + 2\eta_Y/M \geq h^*(s) \geq h(s)$.

**(b)** Assume that there exist $\epsilon \in (0, 2\eta_Y/M]$ such that $h^*(s) - h(s) \geq \epsilon$ for all $s \in \mathbb{N}$. Since $h$ is unbounded, then $\dot{h}(t) \to 0$ when $t \to \infty$ from Eqn. 33, so there exist some $s_0' \in \mathbb{N}$ such that when $s \geq s_0', h(s+1) - h(s) \leq \epsilon + \ln(1/2)/M$. Also, from Lemma 9 we know that exists $s_0'' = \frac{(3+\delta)\ln(M)}{\eta_Y}$ where $\delta > 0, \delta = \Theta(1)$ such that when $s \geq s_0'', \exp(Mh(s)) > 2(M-1)$. Since $s \to \infty$, we just consider the case that $s = \lfloor t \rfloor \geq s_0 := \max(s_0', s_0'')$. Then denote $\Delta_1 := \frac{2(M-1)}{\exp(Mh(s))} < 1$, we have:

$$
\begin{aligned}
\dot{h}^\circ(t) - \dot{h}(t) &= \frac{\eta_Y}{M-1+\exp(Mh^*(s))} - \frac{\eta_Y}{M-1+\exp(Mh(t))} \\
&\leq \frac{\eta_Y}{M-1+\exp(M(h(s)+\epsilon))} - \frac{\eta_Y}{M-1+\exp(Mh(s+1))} \\
&= -\frac{\eta_Y \exp(Mh(s)) \cdot [\exp(M\epsilon) - \exp(Mh(s+1) - Mh(s))]}{[M-1+\exp(M(h(s)+\epsilon))] \cdot [M-1+\exp(Mh(s+1))]} \\
&\leq -\frac{\eta_Y \exp(Mh(s)) \cdot \exp(M\epsilon)}{2[M-1+\exp(M(h(s)+\epsilon))] \cdot [M-1+\frac{1}{2}\exp(M(h(s)+\epsilon))]} \\
&\leq -\frac{\eta_Y \exp(M\epsilon)}{(1+\Delta_1)^2 \exp(Mh(s))\exp(4\eta_Y)}, \quad (s \geq s_0 = \max(s_0', s_0'')) \\
&\leq -\frac{\exp(M\epsilon)}{4\exp(4\eta_Y)M} \cdot \frac{1}{t} =: -\frac{C}{t}
\end{aligned}
\tag{48}
$$

Here $C = \frac{\exp(M\epsilon)}{4\exp(4\eta_Y)M} > 0$ and for the last inequality, we use the fact that $t \geq s_0' > \frac{3\ln M}{\eta_Y}$ and thus $h(s) \leq h(t) = O(\frac{\ln(M\eta_Y t)}{M})$ from Lemma 9. So we get

$$
[h^\circ(t) - h(t)] - [h^\circ(s_0) - h(s_0)] \leq -\int_{t'=s_0}^{\infty} \frac{Cdt}{t} \to -\infty
\tag{49}
$$

This contradicts $h^\circ(t) - h(t) \geq 0$! So the original assumption doesn't hold, which means that $h^*(s) - h(s) \to 0$ when $s \to \infty$. $\qquad\square$

**Remark 2.** *By some qualitative estimation, we claim that if $\eta_Y = O(1)$, then there exists some constant $c = O(\ln M)$ such that $h(s) \leq h^*(s) \leq h(s+c)$ for all $s > s_1 := \frac{2\ln(1+\omega_1)}{\eta_Y}$ where $\omega_1 = \Theta(\ln\ln M/\ln M)$ is defined in Lemma 8. Denote $\delta h(t) := h^\circ(t) - h(t)$, when $\delta h(t) \ll h(t)$,*

we have $\dot{\delta h}(t) = \dot{h}^\circ(t) - \dot{h}(t) \asymp -\eta_Y M \cdot \delta h(t) \cdot \exp(-Mh(t)) \asymp -\delta h(t)/t$ by computing the second-order derivative of $\delta h$, and thus $h^\circ(t) - h(t) \asymp 2\eta_Y s_0/(Mt) = O(\ln M/(Mt))$. Combining this with the fact that $h(t) = \Theta(\ln(M\eta_Y t)/M)$ when $t > s_1$, we prove our claim. The results of Lemma 7 and Remark 2 are also confirmed by the numerical simulation results as Fig. 8.

So from Lemma 7 and Remark 2, we just assume $\eta_Y < 1$ and replace $h^*$ with $h$ in the latter parts for convenience. Then we further investigate the properties of Eqn. 45.

**Lemma 8.** *There exists $\omega_i, 0 < \omega_i \ll 1, i = 2, 3$, such that for $h \in \mathbb{J}_1 := [\frac{1}{M^{2-\omega_0}}, \frac{(1+\omega_1)\ln(M)}{M}]$, we have $\exp(Mh(t)) \le (M-1)Mh(t)$. And for $h \notin \mathbb{J}_1$, we have $\exp(Mh(t)) > (M-1)Mh(t)$. Here $\omega_1 = \Theta(\frac{\ln\ln(M)}{\ln(M)})$, and if $M \gg 100$, we have $\omega_0 \lesssim (\frac{1}{M^{0.99}\ln M}) \ll 0.01$.*

*Proof.* It's obvious that $\exp(Mh(t)) - (M-1)Mh(t)$ has two zero points in $\mathbb{R}^+$. Let $h(t) = M^{-(2-\omega_0)}$, we get

$$\omega_0 = \frac{1}{\ln M}(\ln(\frac{M}{M-1}) + \frac{1}{M^{1-\omega_0}}) = O(\frac{1}{M^{0.99}\ln(M)}) \tag{50}$$

For another zero point, let $\omega_1 \in (0, 1)$ to be some constant such that $h(t) = \frac{(1+\omega_1)\ln(M)}{M}$ satisfies $\exp(Mh) = (M-1)Mh$, then we get

$$M^{\omega_1} = (1+\omega_1)\ln(M)\frac{(M-1)}{M} = c' \cdot \ln(M)\frac{(M-1)}{M}$$

$$\Rightarrow \quad \omega_1 = \Theta(\frac{\ln\ln(M)}{\ln(M)}) \tag{51}$$

where $c' \in (0.5, 2)$ is some universal constant. $\qquad\square$

**Remark 3.** *From Lemma 8, if we assume $M \gg 100$, then $\omega_0 \ll 0.01$, and if we assume $\eta_Y \gg \frac{1}{M^{1-\omega_0}} > \frac{1}{M^{0.99}}$, then $h(1) \gtrsim \frac{\eta_Y}{M} \gg \frac{1}{M^{2-\omega_0}}$ and function $\exp(Mh(t)) - (M-1)Mh(t)$ has only one zero point $\frac{(1+\omega_1)\ln M}{M}$ in $[1, \infty)$. For convenience, we just assume $M \gg 100$ and $1 > \eta_Y \gg \frac{1}{M^{0.99}}$ and thus focus on the unique zero point $\frac{(1+\omega_1)\ln M}{M}$ of $h$ in the latter parts.*

We can then show the properties of speed control coefficient $\gamma(t) := \frac{(M-1)^2 h(t/K)}{(M-1)+\exp(Mh(t/K))}$ as below.

**Lemma 9.** *We have two stage for $h$ and $\gamma$:*

- *When $t \le \frac{K\ln(M)}{\eta_Y}$, we have $\exp(Mh(t/K)) \le \min(M-1, (M-1)Mh(t/K))$, $h = O(\eta_Y t/(MK))$ and $\gamma(t) = O(\eta_Y t/K)$.*

- *When $t \ge \frac{2(1+\omega_1)K\ln(M)}{\eta_Y}$ where $\omega_1 = \Theta(\frac{\ln\ln M}{\ln M})$ is defined in Lemma 8, we have $\exp(Mh(t/K)) \ge \max(M-1, (M-1)Mh(t/K))$, $h = O(\frac{1}{M}\ln(M\eta_Y t/K))$ and $\gamma(t) = O(\frac{K\ln(M\eta_Y t/K)}{\eta_Y t})$.*

*Proof.* For convenience, we just let $K = 1$. And the proof for $K \ne 1$ is similar. We denote $\Delta_1(h) := \frac{\exp(Mh)}{M-1}$ and $\Delta_2(h) := \frac{\exp(Mh)}{(M-1)Mh}$.

**Step 1**: $t \le \frac{\ln(M)}{\eta_Y}$. If $h \ge \frac{\ln(M-1)}{M}$, from Eqn. 45 we have:

$$t \ge \frac{M - 2 + (M-1)\ln(M-1)}{M\eta_Y} > \frac{\ln(M)}{\eta_Y} \tag{52}$$

So when $t \le \frac{\ln(M)}{\eta_Y}$ we have $h < \frac{\ln(M-1)}{M}$, and thus $\exp(Mh(t)) \le \min(M-1, (M-1)Mh(t))$, i.e., $\Delta_1, \Delta_2 \le 1$. Then from Eqn. 45 we get

$$h = \frac{M\eta_Y t + 1}{(1+\Delta_2)M(M-1)} = O\left(\frac{1}{M}\eta_Y t\right) \tag{53}$$

$$\gamma = \frac{(M-1)h}{1+\Delta_1} = \frac{M\eta_Y t + 1}{(1+\Delta_1)(1+\Delta_2)M} = O(\eta_Y t) \tag{54}$$

**Step 2**: $t > \frac{2(1+\omega_1)\ln(M)}{\eta_Y}$ where $\omega_1 = \Theta(\frac{\ln\ln(M)}{\ln(M)})$. So now $h > \frac{\ln(M-1)}{M}$ and thus $\Delta_1 > 1$ from Eqn. 52. Then if $\exp(Mh) \le M(M-1)h$, i.e. $\Delta_2 \le 1$, from Lemma 8 we have $h = \frac{M\eta_Y t+1}{(1+\Delta_2)M(M-1)} \le \frac{(1+\omega_1)\ln(M)}{M}$. Therefore,

$$t \le \frac{1}{\eta_Y}\left((1+\omega_1)(1+\Delta_2)\frac{M-1}{M}\ln M - \frac{1}{M}\right) < \frac{2(1+\omega_1)\ln(M)}{\eta_Y}. \tag{55}$$

Contradiction! So when $t \ge \frac{2(1+\omega_1)\ln(M)}{\eta_Y}$, we have $\Delta_2 > 1$. Then from Eqn. 45 we get:

$$h = \frac{1}{M}\ln\left(\frac{M\eta_Y t+1}{1+\Delta_2^{-1}}\right) = O\left(\frac{1}{M}\ln(M\eta_Y t)\right) \tag{56}$$

$$\gamma = \frac{M-1}{M}\frac{(M-1)\ln(\frac{M\eta_Y t+1}{1+\Delta_2^{-1}})}{(1+\Delta_1^{-1})(\frac{M\eta_Y t+1}{1+\Delta_2^{-1}})} = O\left(\frac{\ln(M\eta_Y t)}{\eta_Y t}\right) \tag{57}$$

$\square$

### D.4  The dynamics under multiple uniformly sampled sequence classes

We then generalize our analysis of $W$ to the case where $x_{T+1}$ can be any value in $[K]$ rather than fixing $x_{T+1} = n$ with the key observation that the row vectors of $W'$ can be independently updated. Before formalizing this result, we first conduct the concentration inequality of the sampling number for each next-token case. Let $N_n := \sum_{i=1}^{N}\mathbb{I}[x_{T+1} = n]$ to be the number of times the event $x_{T+1} = n$ happens, then we have:

**Lemma 10.** *For $\delta \in (0, 1)$, with probability at least $1 - \delta$ we have*

$$|N_n - \lceil N\mathbb{P}(n)\rceil| \le \sqrt{\frac{N}{2}\ln(\frac{2}{\delta})} + 1 < \sqrt{N\ln(\frac{2}{\delta})} \tag{58}$$

*Proof.* From Hoeffding's inequality, we have

$$\mathbb{P}\left(\left|\frac{N_n}{N} - \mathbb{P}(n)\right| > t\right) \le 2\exp(-2Nt^2) \tag{59}$$

Let $t = \sqrt{\frac{1}{2N}\ln(\frac{2}{\delta})}$ and we can get the results by direct calculation. $\square$

**Remark 4.** *From Lemma 10, if we consider the uniform sampling case where $\mathbb{P}(n) \equiv \frac{1}{K}$, then $N\mathbb{P}(n) = N/K \gg \sqrt{N}$. So $N_n$ are all concentrated around $N\mathbb{P}(n)$. Recall the definition of $\bar{N} = \lceil N/K \rceil$ and $\Delta = \lceil\sqrt{N\ln(\frac{1}{\delta})}\rceil$, with probability at least $1 - \delta$ we have:*

$$|N_n - \bar{N}| \lesssim \Delta \ll \bar{N} \tag{60}$$

We then further investigate the concentration of $h(N_n)$:

**Lemma 11.** *For $\delta \in (0, 1)$, with probability at least $1 - \delta$ we have*

$$|h(N_n) - h(\bar{N})| \lesssim h(\bar{N}) \cdot \frac{\Delta}{\bar{N}} \tag{61}$$

$$\left|\frac{1}{M-1+\exp(Mh(N_n))} - \frac{1}{M-1+\exp(Mh(\bar{N}))}\right|$$
$$\lesssim \frac{1}{M-1+\exp(Mh(\bar{N}))} \cdot \sigma' \tag{62}$$

*where $\sigma' > 0$ is some constant such that $\sigma' \le \frac{1}{3}\eta_Y\Delta \ll \ln(M)$. And if $N \ge \frac{2K(1+\omega_1)\ln M}{\eta_Y}$ where $\omega_1$ is defined in Lemma 8, then $\sigma' \lesssim \frac{\Delta}{\bar{N}} \ll 1$.*

*Proof.* First, we note that $h$ has a decreasing gradient, so $h(x) \geq \dot{h}(x) \times x$ and $h(x_1 + x_2) - h(x_1) \leq \dot{h}(x_1) \times x_2$ for any $x_1, x_2 \geq 0$. So with probability at least $1 - \delta$, we have:

$$|h(N_n) - h(\bar{N})| \leq h(\bar{N}) - h(\bar{N} - \Delta) \leq \dot{h}(\bar{N} - \Delta) \times \Delta \leq \frac{h(\bar{N})\Delta}{\bar{N} - \Delta} \asymp h(\bar{N}) \cdot \frac{\Delta}{\bar{N}} \qquad (63)$$

For the second inequality, without loss of generality, we let $N_n > \bar{N}$. Denote $g(s) := (M - 1 + \exp(Mh(s)))^{-1}$ and note that:

$$
\begin{aligned}
\frac{\mathrm{d}g}{\mathrm{d}s} &= \frac{M \exp(Mh(s))}{(M - 1 + \exp(Mh(s)))^2} \cdot \frac{\mathrm{d}h}{\mathrm{d}s} \\
&= \frac{1}{M - 1 + \exp(Mh(s))} \cdot \frac{\eta_Y M \exp(Mh(s))}{(M - 1 + \exp(Mh(s)))^2} \\
&\leq \frac{1}{M - 1 + \exp(Mh(s))} \cdot \frac{M}{(M - 1)} \cdot \frac{\eta_Y}{4}
\end{aligned}
\qquad (64)
$$

the last equality holds only when $h(s) = \frac{\ln(M-1)}{M}$. So from $|g(\bar{N} + \Delta) - g(N_n)| \leq \max_{s \in [N_n, N_n + \Delta]} \dot{g}(s) \cdot \Delta$, we get:

$$\left| \frac{1}{M - 1 + \exp(Mh(\bar{N} + \Delta))} - \frac{1}{M - 1 + \exp(Mh(\bar{N}))} \right| \leq \frac{1}{M - 1 + \exp(Mh(\bar{N}))} \cdot \frac{1}{3}\eta_Y \Delta \qquad (65)$$

If $\bar{N} < \frac{2(1 + \omega_1)\ln(M)}{\eta_Y} + \Delta$ with $\omega_1 = \Theta(\frac{\ln \ln M}{\ln M})$ defined in Lemma 8, we have $\sigma' \leq \eta_Y \Delta / 3 \ll \eta_Y \bar{N} \lesssim \ln(M)$. If $\bar{N} \geq \frac{2(1 + \omega_1)\ln(M)}{\eta_Y} + \Delta$, we utilize the Eqn.45 and obtain:

$$
\begin{aligned}
&\left| \frac{1}{M - 1 + \exp(Mh(\bar{N} + \Delta))} - \frac{1}{M - 1 + \exp(Mh(\bar{N}))} \right| \\
&= \frac{1}{M - 1 + \exp(Mh(\bar{N}))} \cdot \frac{|\exp(Mh(\bar{N} + \Delta)) - \exp(Mh(\bar{N}))|}{M - 1 + \exp(Mh(\bar{N} + \Delta))} \\
&\leq \frac{1}{M - 1 + \exp(Mh(\bar{N}))} \cdot \frac{M \eta_Y \Delta}{M - 1 + \exp(Mh(\bar{N} + \Delta))}, \quad (Eqn. \ 45) \\
&\leq \frac{1}{M - 1 + \exp(Mh(\bar{N}))} \cdot \frac{M \eta_Y \Delta}{M + \frac{1}{2} \cdot M \eta_Y (\bar{N} + \Delta)}, \quad (\text{Lemma } 9, N_n \geq \frac{2(1 + \omega_1)\ln(M)}{\eta_Y} + \Delta) \\
&\overset{\sim}{<} \frac{1}{M - 1 + \exp(Mh(\bar{N}))} \cdot \frac{\Delta}{\bar{N}}
\end{aligned}
$$

So $\sigma' \leq \Delta / \bar{N}$. When $N_n < \bar{N}$, with probability at least $1 - \delta$ we have $N_n \gtrsim \bar{N} - \Delta$, and similar inequalities also hold for such cases, so we finish the proof. $\qquad \square$

Recall that $\boldsymbol{\zeta}_n \in \mathbb{R}^M$ is defined as $\boldsymbol{\zeta}_n = \frac{M}{M-1}(\boldsymbol{e}_n - \frac{1}{M}\mathbf{1})$. And we have $q_1 := \boldsymbol{\zeta}_i^\top \boldsymbol{\zeta}_i = 1 + \frac{1}{M-1}$, $q_0 := \boldsymbol{\zeta}_j^\top \boldsymbol{\zeta}_i = -\frac{M}{(M-1)^2}$ for all $i, j \in [M]$ where $i \neq j$. For convenience, we denote $\bar{W}'(N) := [\bar{\boldsymbol{w}}_1(N), ..., \bar{\boldsymbol{w}}_K(N), \mathbf{0}, ..., \mathbf{0}]^\top \in \mathbb{R}^{M \times M}$, where $\bar{\boldsymbol{w}}_n(N) := (M - 1)h(\lceil N/K \rceil)\boldsymbol{\zeta}_n = (M - 1)h(\bar{N})\boldsymbol{\zeta}_n$. So using these concentration inequalities, we get:

**Lemma 12.** *Assume the assumptions in Lemma 5 hold but we uniformly sample the training data. Then if the total number of epochs $N$ satisfies $N \gg K^2$, we have $Y = (F')^{-\top}(I + \Theta')\bar{W}'(N)$ where $\Theta' := diag(\theta_1, \ldots, \theta_K, 0, \ldots, 0) \in \mathbb{R}^{M \times M}$ and with probability at least $1 - \delta$ we have $|\theta_i| \lesssim \frac{K}{\sqrt{N}}\sqrt{\ln(\frac{K}{\delta})}, \forall i \in [K]$.*

*Proof.* From Lemma 5 and the first inequality of Lemma 11, we know that

$$
\begin{aligned}
\boldsymbol{w}_n(N) &= (M - 1)h(N_n)\boldsymbol{\zeta}_n & (66) \\
&= (M - 1)h(\bar{N})\boldsymbol{\zeta}_n + (M - 1)(h(N_n) - h(\bar{N}))\boldsymbol{\zeta}_n & (67) \\
&= (1 + \theta_n) \cdot (M - 1)h(\bar{N})\boldsymbol{\zeta}_n & (68) \\
&= (1 + \theta_n)\bar{\boldsymbol{w}}_n(N) & (69)
\end{aligned}
$$

where for any $\delta \in (0,1)$, with probability at least $1 - \delta$ we have $|\theta_i| \lesssim \frac{K}{\sqrt{N}} \sqrt{\ln(\frac{K}{\delta})}, \forall n \in [K]$. Therefore, $W'(N) = [\boldsymbol{w}_1(N), \ldots, \boldsymbol{w}_K(N), \boldsymbol{0}, \ldots, \boldsymbol{0}]^\top = (I + \Theta')\bar{W}'(N)$, then from $W' = (F')^\top Y$, we finish the proof. $\qquad \square$

Then, we can give out the exact solution of $Y$ by pointing out the properties of $F^\circ$ and $F'$ from the observation that each row of $Y$ should be the linear combination of vectors in $\{\boldsymbol{f}_n^\top\}_{n \in [K]}$:

**Theorem 5.** *If Assumption 2 holds and $Y(0) = 0$. Furthermore, we assume the training data is uniformly sampled and the total number of epochs $N$ satisfies $N \gg K^2$. Then the solution of Eqn. 26 will be:*

$$Y = (F^\dagger)^\top (I + \Theta)\bar{W}(N) = F(I - E')(I + \Theta)\bar{W}(N) \tag{70}$$

*Here $\Theta := diag(\theta_1, \ldots, \theta_K)$ and for any $\delta \in (0,1)$, with probability at least $1 - \delta$ we have $|\theta_i| \lesssim \frac{K}{\sqrt{N}} \sqrt{\ln(\frac{K}{\delta})}, \forall i \in [K]$.*

*Proof.* Let $\boldsymbol{q}_i, i \in [M]$ be the $i$-th row vector of $(F')^{-1}$, then we have $\boldsymbol{q}_j^\top \boldsymbol{f}_i = \mathbb{I}[i = j]$. From Lemma 12 we get $Y = (F')^{-\top}(I + \Theta')\bar{W}'(N)$. And from Eqn. 26, we know all the columns of $Y$ are the linear combination of $\boldsymbol{f}_n, n \in [K]$. Note that $\bar{W}(N)$ has only top $K$ rows to be non-zero, so we need to constrain that all the top $K$ columns of $(F')^{-\top}$, i.e., $\boldsymbol{q}_i, i \in [K]$, to be the linear combination of $\boldsymbol{f}_n, n \in [K]$, which means that $\boldsymbol{q}_1, \ldots, \boldsymbol{q}_K$ must be the basis of $\Xi := \text{span}(\boldsymbol{f}_j; j \in [K])$ and thus $\boldsymbol{q}_{K+1}, \ldots, \boldsymbol{q}_M$ are the basis of $\Xi' := \text{span}(\boldsymbol{f}_j; K \leq j \leq M)$. Therefore, we get $\Xi \perp \Xi'$, and thus $[\boldsymbol{q}_1, \ldots, \boldsymbol{q}_K]$ can only be $(F^\dagger)^\top$. So the proof is done.

$\qquad \square$

Actually, we see that the result of Theorem 5 matches the modified gradient update on $Y$ (Eqn. 26). And we show that using such reparameterization dynamics, we can still approach the critical point of Eqn. 7 in the rate of $\mathcal{O}(\frac{1}{N})$:

**Corollary 1.** *Assume assumptions in Theorem 5 hold, $M \gg 100$ and $\eta_Y$ satisfies $M^{-0.99} \ll \eta_Y < 1$. Then $\forall n \in [K]$, we have*

$$\begin{aligned}(\boldsymbol{x}_{T+1} - \boldsymbol{\alpha}_n) &= \frac{M - 1}{(M - 1) + \exp(Mh(N_n))} \boldsymbol{\zeta}_n \\ &= \frac{M - 1}{(M - 1) + \exp(Mh(\bar{N}))} \cdot (1 + \sigma) \cdot \boldsymbol{\zeta}_n\end{aligned} \tag{71}$$

*where $\sigma > -1$ and for any $\delta \in (0,1)$, with probability at least $1 - \delta$ we have $|\sigma| \lesssim \eta_Y \sqrt{N \ln(\frac{1}{\delta})}$, and when $N \gg K(\sqrt{N \ln(\frac{1}{\delta})} + \frac{2(1+\omega_1)\ln M}{\eta_Y})$ with $\omega_1$ defined in Lemma 8, $|\sigma| \lesssim \frac{K}{\sqrt{N}} \sqrt{\ln(\frac{1}{\delta})}$. Further, to let $\|\boldsymbol{x}_{T+1} - \boldsymbol{\alpha}_n\|_2 \leq \epsilon$ with probability at least $1 - \delta$ for any $n \in [K]$ and $\epsilon \ll 1$, we need the total number of training epochs to be at most $O(\frac{K}{\epsilon \eta_Y} \log(\frac{M}{\epsilon}))$.*

*Proof.* Note that $\boldsymbol{x}_{T+1} = \boldsymbol{e}_n$, then we just need to combine Lemma 5 and the second inequality of Lemma 11, to get Eqn. 71. Denote $S_n$ to be the number of training epochs that are needed to let $\|\boldsymbol{x}_{T+1} - \boldsymbol{\alpha}_n\|_2 \asymp \epsilon$, then we have

$$h(S_n) \asymp \frac{1}{M} \ln(\frac{M}{\epsilon}) \tag{72}$$

But note that $h(t + 1) - h(t) \geq \frac{\eta_Y}{M - 1 + \exp(Mh(S_n))} \asymp \frac{\eta_Y \epsilon}{M - 1}, \forall t \in [0, S - 1]$ from Eqn. 71, we have

$$S_n \lesssim \frac{h(S_n)}{\eta_Y \epsilon / (M - 1)} \asymp \frac{1}{\epsilon \eta_Y} \ln(\frac{M}{\epsilon}) \tag{73}$$

Note that $\epsilon \ll 1$ and we have $N \gg K^2$, then we have $S = \sum_n S_n \lesssim \frac{K}{\epsilon \eta_Y} \ln(\frac{M}{\epsilon})$. $\qquad \square$

## D.5 Proof of Theorem 1

Finally, we turn to prove Theorem 1. Obviously, all the diagonal elements of $E$ are zero and all the off-diagonal elements of $E$ are non-negative since $c_{l|m,n} \geq 0$. Note that $E$ is a real symmetric matrix, then it can be orthogonal diagonalization by $E = U^\top D U$ where $U := [u_1, ..., u_K] \in O_{K \times K}, D = \text{diag}(\lambda_1, ..., \lambda_K)$ and $|\lambda_1| \geq ... \geq |\lambda_K| \geq 0$. Then we can get the following properties of $E$ and $E'$:

**Lemma 13.** $\max_{i,j \in [K]}(|E_{ij}|) \leq |\lambda_1|$.

*Proof.* We have:

$$|E_{ij}| = u_i^\top D u_j \leq |\lambda_1| \cdot \|u_i\|_2 \|u_j\|_2, \quad \forall i, j \in [K] \tag{74}$$

$\square$

**Lemma 14.** *If $E \in \mathbb{R}^K$ satisfies $|\lambda_1| \leq \lambda < 1$, then $(I + E)$ is invertible and $(I + E)^{-1} = I - E'$ ,where $E'$ satisfies $E' = U^\top D' U$ and $D' = \text{diag}(\lambda_1', ..., \lambda_K')$ and $\lambda_i' = \frac{\lambda_i}{1+\lambda_i}, \forall i \in [K]$.*

*Proof.* Since $U$ is orthonormal and $|\lambda_i| \leq \lambda < 1$, we have $E^n = U^\top D^n U \to O$. Then from the property of the Neumann series, we get $I + E$ is invertible and

$$(I + E)^{-1} = I + \sum_{n=1}^{\infty} (-1)^n E^n \tag{75}$$

$$= I + U^\top (\sum_{n=1}^{\infty} (-D^n) U \tag{76}$$

$$= I - U^\top D' U =: I - E' \tag{77}$$

Here we define $D' = \text{diag}(\lambda_1', ..., \lambda_K')$ and use the fact that $\sum_{n=1}^{\infty} (-\lambda_i)^n = -\frac{\lambda_i}{1+\lambda_i}$ $\square$

**Lemma 15.** *If $|\lambda_1| \leq \lambda < 1$, then $\max_{i \in [K]} |\lambda_i(E')| \leq \frac{1}{1-\lambda}|\lambda_1| \leq \frac{\lambda}{1-\lambda}$.*

*Proof.* We have

$$\max_{i \in [K]} |\lambda_i(E')| = \max_{i \in [K]} |-\frac{\lambda_i}{1+\lambda_i}| \leq \frac{\max_{i \in [K]} |\lambda_i|}{1 - \max_{i \in [K]} |\lambda_i|} \leq \frac{1}{1-\lambda}|\lambda_1| \tag{78}$$

$\square$

**Lemma 16.** *Assume that Assumption 2 holds, then all the diagonal elements of $E'$ are non-positive,i.e., $E'_{ii} \leq 0, \forall i \in [K]$. Further, if there exist any $k \neq i \in [K]$ such that $E_{ki} > 0$, then $E'_{ii} < 0$.*

*Proof.* Note that $E_{ii} = \sum_{k=1}^{K} \lambda_k u_{ik}^2 = 0$ (here $u_{ik}$ is the $k$-th component of eigenvector $u_i$) and $|\lambda_k| < 1$, we have

$$E'_{ii} = \sum_{k=1}^{K} \frac{\lambda_k}{1+\lambda_k} u_{ik}^2 = \sum_{k=1}^{K} \lambda_k u_{ik}^2 - \sum_{k=1}^{K} \frac{\lambda_k^2}{1+\lambda_k} u_{ik}^2 = -\sum_{k=1}^{K} \frac{\lambda_k^2}{1+\lambda_k} u_{ik}^2 \leq 0 \tag{79}$$

When $E'_{ii} = 0$, then $\lambda := (\lambda_1, ..., \lambda_K)$ must don't have overlapping entries with respect to $u_i$, which results that $E_{ij} := \sum_{k=1}^{K} \lambda_k u_{ik} u_{jk} = 0$ holds for any $j \in [K]$. So we prove the results.

$\square$

**Lemma 17.** *If $\lambda_1 < 1$, then $|E'_{nn'} - E_{nn'}| \leq |\lambda_1|^2 (1 - |\lambda_1|)^{-1}$.*

*Proof.* From Lemma 14 we have:

$$
\begin{aligned}
|E'_{nn'} - E_{nn'}| &= |\sum_{k=1}^{K} \lambda_k u_{nk} u_{n'k} - \sum_{k=1}^{K} \frac{\lambda_k}{1 + \lambda_k} u_{nk} u_{n'k}| \\
&= |\sum_{k=1}^{K} \frac{\lambda_k^2}{1 + \lambda_k} u_{nk} u_{n'k}| \\
&\leq \frac{|\lambda_1|^2}{1 - |\lambda_1|} \sum_{k=1}^{K} |u_{nk}||u_{n'k}| \\
&\leq \frac{|\lambda_1|^2}{1 - |\lambda_1|} \sqrt{(\sum_{k=1}^{K} |u_{nk}|^2)(\sum_{k=1}^{K} |u_{n'k}|^2)} = \frac{|\lambda_1|^2}{1 - |\lambda_1|}
\end{aligned}
\tag{80}
$$

□

Finally we can prove our main theorem in Sec. 4.

**Theorem 1.** *If Assumption 2 holds, the initial condition $Y(0) = 0$, $M \gg 100$, $\eta_Y$ satisfies $M^{-0.99} \ll \eta_Y < 1$, and each sequence class appears uniformly during training, then after $t \gg K^2$ steps of batch size 1 update, given event $x_{T+1}[i] = n$, the backpropagated gradient $\boldsymbol{g}[i] := Y(\boldsymbol{x}_{T+1}[i] - \boldsymbol{\alpha}[i])$ takes the following form:*

$$
\boldsymbol{g}[i] = \gamma \left( \iota_n \boldsymbol{f}_n - \sum_{n' \neq n} \beta_{nn'} \boldsymbol{f}_{n'} \right)
\tag{9}
$$

*Here the coefficients $\iota_n(t)$, $\beta_{nn'}(t)$ and $\gamma(t)$ are defined in Appendix with the following properties:*

- *(a) $\xi_n(t) := \gamma(t) \sum_{n \neq n'} \beta_{nn'}(t) \boldsymbol{f}_n^\top(t) \boldsymbol{f}_{n'}(t) > 0$ for any $n \in [K]$ and any $t$;*

- *(b) The* speed control coefficient *$\gamma(t) > 0$ satisfies $\gamma(t) = O(\eta_Y t / K)$ when $t \leq \frac{\ln(M) \cdot K}{\eta_Y}$ and $\gamma(t) = O\left(\frac{K \ln(\eta_Y t / K)}{\eta_Y t}\right)$ when $t \geq \frac{2(1+\delta') \ln(M) \cdot K}{\eta_Y}$ with $\delta' = \Theta(\frac{\ln \ln M}{\ln M})$.*

*Proof.* Note that if Assumption 2 holds, then $F^\dagger = (I - E')F^\top$. Recall $q_1 := 1 + \frac{1}{M-1} \approx 1$ and $q_0 := -\frac{M}{(M-1)^2} \approx 0$. Then given $x_{T+1}[i] = n$, we get:

$$
\begin{aligned}
\boldsymbol{g}[i] \quad &:= \quad Y(\boldsymbol{x}_{T+1}[i] - \boldsymbol{\alpha}[i]) &\tag{81} \\
&= \quad F(I - E')(I + \Theta)\bar{W}(N)(\boldsymbol{x}_{T+1}[i] - \boldsymbol{\alpha}[i]), \quad \text{(Theorem 5)} &\tag{82} \\
&= \quad (1 + \sigma)\gamma * F(I - E')(I + \Theta)[q_0, \ldots, q_1, \ldots, q_0]^\top, \quad \text{(Lemma 5, Corollary 1)} &\tag{83} \\
&= \quad \gamma \left( \iota_n \boldsymbol{f}_n - \sum_{n' \neq n, n' \in [K]} \beta_{nn'} \boldsymbol{f}_{n'} \right) &\tag{84}
\end{aligned}
$$

where

$$
\begin{aligned}
\gamma(t) \quad &:= \quad \frac{(M-1)^2 h(\lceil t/K \rceil)}{(M-1) + \exp(Mh(\lceil t/K \rceil))} > 0 &\tag{85} \\
\iota_n \quad &:= \quad (1 + \sigma)[q_1 \cdot (1 + \theta_n)(1 - E'_{nn}) - q_0 \sum_{k \neq n, k \in [K]} (1 + \theta_k)E'_{kn}] &\tag{86} \\
&= \quad (1 + \sigma)[(1 - E'_{nn}) \cdot (1 + \delta_1) + \delta_2] &\tag{87} \\
\beta_{nn'} \quad &:= \quad (1 + \sigma)[q_1 \cdot (1 + \theta_n)E'_{nn'} + q_0((1 + \theta_{n'}) + \sum_{k \neq n, k \in [K]} (1 + \theta_k)E'_{kn'}))] &\tag{88} \\
&= \quad (1 + \sigma)[E'_{nn'} \cdot (1 + \delta_1) + \delta_3] &\tag{89}
\end{aligned}
$$

Here $\sigma$ is defined in Cor. 1 and satisfies $-1 < \sigma \ll \ln M$. $|\delta_1| \lesssim \frac{K}{\sqrt{N}}\sqrt{\ln(\frac{1}{\delta})} + \frac{1}{M} \ll 1$ and $|\delta_2|, |\delta_3| \leq \frac{M}{(M-1)^2} \times 2(1 + 3|\delta_1|) < \frac{3}{M}$. Here we use the fact that $|\theta|, |\theta_i| \lesssim \frac{K}{\sqrt{N}}\sqrt{\ln(\frac{1}{\delta})}$, $\sum_{k \in [K]} \lambda_k u_{jk} u_{jn'} = E_{kn'}$ and the fact from Lemma 15:

$$|E'_{kn}| \leq \max_{i \in [K]} |\lambda_i(E')| \leq \frac{1}{1 - 1/K}|\lambda_1| \leq \frac{1}{K-1} \tag{90}$$

**(a)** Now let's prove that $\xi_n(t) > 0$. First from $(I + E)(I - E') = I$ we have $E - E' - EE' = O$. Then use the symmetry of $E$ and $E'$, we get

$$(EE')_{nn} = \sum_{k=1} E_{nk} E'_{kn} = \sum_{k=1} E_{nk} E'_{nk} = \sum_{k=1} E_{nk} E'_{nk} = \sum_{k \neq n} E_{nk} E'_{nk} + E_{nn} E'_{nn} \tag{91}$$

Note that $F^\top F = I + E$, we have $E_{nn'} = \boldsymbol{f}_n^\top \boldsymbol{f}_{n'}, \forall n' \neq n$ and $E_{nn} = 0$. Then

$$(E - E' - EE')_{nn} = O_{nn} = 0 \Rightarrow \sum_{k \neq n} E_{nk} E'_{nk} = -E'_{nn} \tag{92}$$

Note that $|\lambda_i(E)| > 0, \forall i \in [K]$ in Assumption 2 implies that $E_{ki} > 0$ holds for some $k \neq i \in [K]$. Then from (1) of Lemma 16 we get $\sum_{k \neq n} E'_{nn'} \boldsymbol{f}_n^\top \boldsymbol{f}_{n'} > 0$.

From Theorem 1 we have $\beta_{nn'} = (1 + \sigma)[E'_{nn'} \cdot (1 + \delta_1) + \delta_3]$. Note that $0 < 1 + \sigma \ll \ln(M)$, we have:

$$
\begin{aligned}
\sum_{n' \neq n} \beta_{nn'} \boldsymbol{f}_n^\top \boldsymbol{f}_{n'} &= (1 + \sigma)[\sum_{n' \neq n}[E'_{nn'}(1 + \delta_1) + \delta_3]E_{nn'}] \\
&= (1 + \sigma)[-(1 + \delta_1)E'_{nn} + \delta_3 \sum_{n' \neq n} E_{nn'}] \\
&= (1 + \sigma)[(1 + \delta_1)\sum_{k=1}^{K} \frac{\lambda_k^2}{1 + \lambda_k} u_{nk}^2 + \delta_3 \sum_{n' \neq n} E_{nn'}] \quad \text{(Eqn. 79)} \\
&\geq (1 + \sigma)[\frac{1 + \delta_1}{1 - |\lambda_1|}(\min_i |\lambda_i(E)|^2) - \frac{3}{M} \cdot K|\lambda_1|], \quad (\text{Eqn. 90, } |\delta_3| < \frac{3}{M}) \\
&> (1 + \sigma)[\frac{1}{2}(\min_i |\lambda_i(E)|^2) - \frac{3}{M} \cdot K|\lambda_1|], \quad (|\delta_1| \ll 1, |\lambda_1| < \frac{1}{K} \ll 1) \\
&> 0, \quad \text{(Assumption 2)}
\end{aligned}
\tag{93}
$$

**(b)** We directly use Lemma 9, then we finish the proof. $\square$

# E  Proof of Section 5

**Lemma 4** (Self-attention dynamics). *With Assumption 1(b) (i.e., $T \to +\infty$), Eqn. 4 becomes:*

$$\dot{\boldsymbol{z}}_m = \eta_Z \gamma \sum_{n \in \psi^{-1}(m)} \text{diag}(\boldsymbol{f}_n) \sum_{n' \neq n} \beta_{nn'}(\boldsymbol{f}_n \boldsymbol{f}_n^\top - I)\boldsymbol{f}_{n'}, \tag{10}$$

*Proof.* Taking long sequence limit $(T \to +\infty)$, and summing over all possible choices of next token $x_{T+1} = n$, plugging in the backpropagated gradient (Eqn. 9) into the dynamics of $Z$ with query token $m$ (Eqn. 4), we arrive at the following:

$$
\begin{aligned}
\dot{\boldsymbol{z}}_m &= \eta_Z \sum_{n \in \psi^{-1}(m)} \text{diag}(\boldsymbol{c}_n)\frac{P_{\boldsymbol{f}_n}^{\perp}}{\|\boldsymbol{c}_n\|_2} Y(\boldsymbol{x}_{T+1}[i] - \boldsymbol{\alpha}[i]) \\
&= -\eta_Z \gamma \sum_{n \in \psi^{-1}(m)} \text{diag}(\boldsymbol{f}_n) P_{\boldsymbol{f}_n}^{\perp} \sum_{n' \neq n} \beta_{nn'} \boldsymbol{f}_{n'} \\
&= \eta_Z \gamma \sum_{n \in \psi^{-1}(m)} \text{diag}(\boldsymbol{f}_n)(\boldsymbol{f}_n \boldsymbol{f}_n^\top - I) \sum_{n' \neq n} \beta_{nn'} \boldsymbol{f}_{n'}
\end{aligned}
$$

$$
\tag{94}
$$
$$
\tag{95}
$$
$$
\tag{96}
$$

Note here we leverage the property that $P_{\boldsymbol{f}}^{\perp}\boldsymbol{f} = 0$ and $P_{\boldsymbol{c}_n}^{\perp} = P_{\boldsymbol{f}_n}^{\perp}$. $\qquad\square$

**Theorem 2** (Fates of contextual tokens). *Let $G_{CT}$ be the set of common tokens (CT), and $G_{DT}(n)$ be the set of distinct tokens (DT) that belong to next token $n$. Then if Assumption 2 holds, under the self-attention dynamics (Eqn. 10), we have:*

- *(a) for any distinct token $l \in G_{DT}(n)$, $\dot{z}_{ml} > 0$ where $m = \psi(n)$;*

- *(b) if $|G_{CT}| = 1$ and at least one next token $n \in \psi^{-1}(m)$ has at least one distinct token, then for the single common token $l \in G_{CT}$, $\dot{z}_{ml} < 0$.*

*Proof.* For any token $l$, we have:

$$\dot{z}_{ml} = \eta_Z \gamma \sum_{n \in \psi^{-1}(m)} f_{nl} \sum_{n' \neq n} \beta_{nn'} \left[ (\boldsymbol{f}_n^\top \boldsymbol{f}_{n'}) f_{nl} - f_{n'l} \right] \tag{97}$$

**Distinct token**. For a token $l$ distinct to $n$, by definition, for any $n' \neq n$, $\mathbb{P}(l|m, n') = 0$ and $f_{n'l}(t) \propto \mathbb{P}(l|m, n') \exp(z_{ml}) \equiv 0$. Therefore, we have:

$$\dot{z}_{ml} = \eta_Z \gamma f_{nl}^2 \sum_{n' \neq n} \beta_{nn'} \boldsymbol{f}_n^\top \boldsymbol{f}_{n'} = \eta_Z f_{nl}^2 \xi_n > 0 \tag{98}$$

Note that $\dot{z}_{ml} > 0$ is achieved by $\xi_n > 0$ from Theorem 1.

**Common token**. For any query token $m$, consider $n \in \psi^{-1}(m)$ and $n' \neq n$. if $n$ and $n'$ does not overlap then $\mathrm{diag}(\boldsymbol{f}_n)(\boldsymbol{f}_n \boldsymbol{f}_n^\top - I)\boldsymbol{f}_{n'} = -\mathrm{diag}(\boldsymbol{f}_n)\boldsymbol{f}_{n'} = 0$. When $n$ and $n'$ overlaps, let $G_{CT}(n, n') := \{l : \mathbb{P}(l|n)\mathbb{P}(l|n') > 0\}$ be the subset of common tokens shared between $n$ and $n'$, since $|G_{CT}| = 1$ and $\emptyset \neq G_{CT}(n, n') \subseteq G_{CT} := \bigcup_{n \neq n'} G_{CT}(n, n')$, we have $|G_{CT}(n, n')| = 1$ and $l \in G_{CT}(n, n')$, i.e., the common token $l$ is the unique overlap. Then we have:

$$f_{nl} \left[ (\boldsymbol{f}_n^\top \boldsymbol{f}_{n'}) f_{nl} - f_{n'l} \right] = (\boldsymbol{f}_n^\top \boldsymbol{f}_{n'}) f_{nl}^2 - \boldsymbol{f}_n^\top \boldsymbol{f}_{n'} = -(1 - f_{nl}^2)(\boldsymbol{f}_n^\top \boldsymbol{f}_{n'}) \tag{99}$$

So we have:

$$\dot{z}_{ml} = -\eta_Z \gamma \sum_{n \in \psi^{-1}(m)} (1 - f_{nl}^2) \sum_{n' \neq n} \beta_{nn'} \boldsymbol{f}_n^\top \boldsymbol{f}_{n'} = -\eta_Z \sum_{n \in \psi^{-1}(m)} (1 - f_{nl}^2) \xi_n \leq 0 \tag{100}$$

Since $\xi_n(t) > 0$, the only condition that will lead to $\dot{z}_{ml} = 0$ is $f_{nl}^2 = 1$. However, since at least one such $n$ has another distinct token $l'$, and thus $f_{nl'} > 0$, by normalization condition, $f_{nl} < 1$ and thus $\dot{z}_{ml} < 0$.

$\qquad\square$

Note that for multiple common tokens, things can be quite involved. Here we prove a case when the symmetric condition holds.

**Corollary 2** (Multiple CTs, symmetric case). *If Assumption 2 holds and assume*

- *(1) Single query token $m_0$. For any next token $n \in [K]$, $\psi(n) = m_0$.*

- *(2) Symmetry. For any two next tokens $n \neq n'$, there exists a one-to-one mapping $\phi$ that maps token $l \in G_{DT}(n)$ to $l' \in G_{DT}(n')$ so that $\mathbb{P}(l|n) = \mathbb{P}(\phi(l)|n')$;*

- *(3) Global common tokens with shared conditional probability: i.e., the global common token set $G_{CT}$ satisfies the following condition: for any $l \in G_{CT}$, $\mathbb{P}(l|n) = \rho_l$, which is independent of next token $n$;*

- *(4) The initial condition $Z(0) = 0$.*

*Then for any common token $l \in G_{CT}$, $\dot{z}_{m_0,l} < 0$.*

*Proof.* Since there is a global query token $m_0$, we omit the subscript $m_0$ and let $z_l := z_{m_0,l}$.

We want to prove the following ***induction hypothesis***: for any $t$ (a) $z_l(t) = z_{\phi_m(l)}(t)$ for $n$, $n'$ which are next tokens that the distinct token $l$ (and $l'$) belongs to, and (b) the normalization term $o_n^2(t) := \sum_l \tilde{c}_{l|n}^2(t) = o^2(t)$, i.e., it does not depend on $n$.

We prove by induction on infinitesimal steps $\delta t$. First when $t = 0$, both conditions hold due to the initial condition $Z(0) = 0$. Then we assume that both conditions hold for time $t$, then by symmetry, we know that for any $n_1$ and any distinct token $l_1 \in G_{DT}(n_1)$:

$$\dot{z}_{l_1}(t) = \eta_Z \gamma f_{n_1 l_1}^2 \sum_{n' \neq n_1} \beta_{n_1 n'} \boldsymbol{f}_{n_1}^\top \boldsymbol{f}_{n'} = \eta_Z \gamma f_{n_2 l_2}^2 \sum_{n' \neq n_2} \beta_{n_2 n'} \boldsymbol{f}_{n_2}^\top \boldsymbol{f}_{n'} = \dot{z}_{l_2}(t) \tag{101}$$

where $l_2 = \phi(l_1)$ is the image of the distinct token $l_1$. This is because (1) $\boldsymbol{f}_{n_1}^\top \boldsymbol{f}_{n'} = \sum_{l \in G_{CT}} \rho_l^2 \exp(2 z_l(t))/o^2(t)$ is independent of $n_1$ and $n'$ by inductive hypothesis, therefore, $\beta$ is also independent of its subscripts. And (2) $f_{n_1 l_1}^2 := \tilde{c}_{l_1|n_1}^2/o^2(t) = \tilde{c}_{l_2|n_2}^2/o^2(t) = f_{n_2 l_2}^2$.

Therefore, $\dot{z}_{l_1}(t) = \dot{z}_{l_2}(t)$, which means that $z_{l_1}(t') = z_{l_2}(t')$ for $t' = t + \delta t$.

Let $G_{CT}(n_1, n_2) := \{l : \mathbb{P}(l|n_1)\mathbb{P}(l|n_2) > 0\}$ be the subset of common tokens shared between $n_1$ and $n_2$, then for their associated $n_1$ and $n_2$, obviously $G_{CT}(n_1, n_2) \subseteq G_{CT}$ and we have:

$$o_{n_1}(t') = \sum_l \tilde{c}_{l|n_1}^2(t') = \sum_l \mathbb{P}^2(l|n_1) \exp(2 z_l(t')) \tag{102}$$

$$= \sum_{l_1 \in G_{DT}(n_1)} \mathbb{P}^2(l_1|n_1) \exp(2 z_{l_1}(t')) + \sum_{l \in G_{CT}(n_1, n_2)} \mathbb{P}^2(l|n_1) \exp(2 z_l(t')) \tag{103}$$

$$= \sum_{l_1 \in G_{DT}(n_1)} \mathbb{P}^2(\phi(l_1)|n_2) \exp(2 z_{\phi(l_1)}(t')) + \sum_{l \in G_{CT}(n_1, n_2)} \rho_l^2 \exp(2 z_l(t')) \tag{104}$$

$$= \sum_{l_2 \in G_{DT}(n_2)} \mathbb{P}^2(l_2|n_2) \exp(2 z_{l_2}(t')) + \sum_{l \in G_{CT}(n_1, n_2)} \mathbb{P}^2(l|n_2) \exp(2 z_l(t')) \tag{105}$$

$$= o_{n_2}(t') \tag{106}$$

So we prove the induction hypothesis holds for $t' = t + \delta t$. Let $\delta t \to 0$ and we prove it for all $t$.

Now we check the dynamics of common token $l \in G_{CT}$. First we have for any $n \neq n'$, $f_{nl}^2(t) = \tilde{c}_{l|n}^2(t)/o^2(t) = \rho_l^2 \exp(2 z_l(t))/o^2(t) = \tilde{c}_{l|n'}^2(t)/o^2(t) = f_{n'l}^2(t)$ and thus $f_{nl}(t) = f_{n'l}(t) := f_l(t) > 0$, therefore:

$$f_{nl}\left[(\boldsymbol{f}_n^\top \boldsymbol{f}_{n'})f_{nl} - f_{n'l}\right] = -f_l^2(1 - \boldsymbol{f}_n^\top \boldsymbol{f}_{n'}) < 0 \tag{107}$$

On the other hand, from the proof on induction hypothesis, we know all off-diagonal elements of $E$ are the same and are positive. Then all all the off-diagonal elements of $E'$ are also the same and are positive. Following Theorem 1, we know $\beta_{nn'} > 0$ and is independent of the subscripts. Therefore, $\dot{z}_l < 0$. $\square$

**Theorem 3** (Growth of distinct tokens). *For a next token $n$ and its two distinct tokens $l$ and $l'$, the dynamics of the **relative gain** $r_{l/l'|n}(t) := f_{nl}^2(t)/f_{nl'}^2(t) - 1 = \tilde{c}_{l|n}^2(t)/\tilde{c}_{l'|n}^2(t) - 1$ has the following analytic form (here the query token $m = \psi(n)$ and is uniquely determined by distinct token $l$):*

$$r_{l/l'|n}(t) = r_{l/l'|n}(0)e^{2(z_{ml}(t) - z_{ml}(0))} =: r_{l/l'|n}(0)\chi_l(t) \tag{11}$$

*where $\chi_l(t) := e^{2(z_{ml}(t) - z_{ml}(0))}$ is the **growth factor** of distinct token $l$. If there exist a dominant token $l_0$ such that the initial condition satisfies $r_{l_0/l|n}(0) > 0$ for all its distinct token $l \neq l_0$, and all of its common tokens $l$ satisfy $\dot{z}_{ml} < 0$. Then both $z_{ml_0}(t)$ and $f_{nl_0}(t)$ are monotonously increasing over $t$, and*

$$e^{2 f_{nl_0}^2(0) B_n(t)} \leq \chi_{l_0}(t) \leq e^{2 B_n(t)} \tag{12}$$

*here $B_n(t) := \eta_Z \int_0^t \xi_n(t') \mathrm{d}t'$. Intuitively, larger $B_n$ gives larger $r_{l_0/l|n}$ and sparser attention map.*

*Proof.* Let $m = \psi(n)$ be the query token associated with next token $n$. For brievity, we omit subscript $m$ in the proof and let $z_l := z_{ml}$.

First of all, for tokens $l$ and $l'$ that are both distinct for a specific next token $n$, from Eqn. 98, it is clear that

$$\frac{\dot{z}_l}{\dot{z}_{l'}} = r_{l/l'|n}(t) + 1 = (r_{l/l'|n}(0) + 1) \frac{e^{2(z_l(t) - z_l(0))}}{e^{2(z_{l'}(t) - z_{l'}(0))}} \tag{108}$$

This means that

$$e^{-2(z_l-z_l(0))}\dot{z}_l = (r_{l/l'|n}(0)+1)e^{-2(z_{l'}-z_{l'}(0))}\dot{z}_{l'} \tag{109}$$

Integrate both side over time $t$ and we get:

$$e^{-2(z_l(t)-z_l(0))} - 1 = (r_{l/l'|n}(0)+1)\left[e^{-2(z_{l'}(t)-z_{l'}(0))} - 1\right] \tag{110}$$

From this we can get the close-form relationship between $r_{l/l'|n}(t)$ and $z_l(t)$:

$$r_{l/l'|n}(t) = r_{l/l'|n}(0)e^{2(z_l(t)-z_l(0))} \tag{111}$$

Now let $l_0$ be the dominating distinct token that satisfies $r_{l_0/l|n}(0) > 0$ for any distinct token $l$, then

- we have $\dot{z}_{l_0} > 0$ due to Theorem 2.

- for any token $l' \neq l_0$ that is distinct to $n$, we have:

$$\dot{r}_{l_0/l'|n} = r_{l_0/l'|n}(0)e^{2(z_{l_0}(t)-z_{l_0}(0))}\dot{z}_{l_0} > 0 \tag{112}$$

- for any common token $l'$, since $\dot{z}_{l'} < 0$, we have

$$\dot{r}_{l_0/l'|n} = \frac{\mathrm{d}}{\mathrm{d}t}\left(\frac{\tilde{c}_{nl_0}^2}{\tilde{c}_{nl'}^2}\right) = \frac{\mathbb{P}^2(l_0|n)}{\mathbb{P}^2(l'|n)}e^{2(z_{l_0}-z_{l'})} \cdot 2(\dot{z}_{l_0} - \dot{z}_{l'}) > 0 \tag{113}$$

Therefore, we have:

$$\frac{\mathrm{d}}{\mathrm{d}t}(f_{nl_0}^2) = \frac{\mathrm{d}}{\mathrm{d}t}\left(\frac{1}{M + \sum_{l' \neq l_0} r_{l'/l_0|n}}\right) > 0 \tag{114}$$

Therefore, $f_{nl_0}^2(t)$ is monotonously increasing. Combined with the fact $f_{nl_0}^2(t) \leq 1$ due to normalization condition $\|\boldsymbol{f}_n\|_2 = 1$, we have:

$$\xi_n(t) \geq \frac{1}{\eta_Z}\dot{z}_{l_0} = f_{nl_0}^2(t)\xi_n(t) \geq f_{nl_0}^2(0)\xi_n(t) \tag{115}$$

Integrate over time and we have:

$$B(t) \geq \int_0^t \dot{z}_{l_0}(t')\mathrm{d}t' = z_{l_0}(t) - z_{l_0}(0) \geq f_{nl_0}^2(0)B(t) \tag{116}$$

where $B(t) := \eta_Z \int_0^t \xi_n(t')\mathrm{d}t'$. Plugging that into Eqn. 111, and we have:

$$e^{2f_{nl_0}^2(0)B(t)} \leq \chi_{l_0}(t) \leq e^{2B(t)} \tag{117}$$

$\square$

## F    Estimation in Sec. 6

**Theorem 4** (Phase Transition in Training). *If the dynamics of the single common token $z_{ml}$ satisfies $\dot{z}_{ml} = -C_0^{-1}\eta_Z\gamma(t)e^{4z_{ml}}$ and $\xi_n(t) = C_0^{-1}\gamma(t)e^{4z_{ml}}$, then we have:*

$$B_n(t) = \begin{cases} \frac{1}{4}\ln\left(C_0 + \frac{2(M-1)^2}{KM^2}\eta_Y\eta_Z t^2\right) & t < t_0' := \frac{K\ln M}{\eta_Y} \\ \frac{1}{4}\ln\left(C_0 + \frac{2K(M-1)^2}{M^2}\frac{\eta_Z}{\eta_Y}\ln^2(M\eta_Y t/K)\right) & t \geq t_0 := \frac{2(1+o(1))K\ln M}{\eta_Y} \end{cases} \tag{14}$$

*As a result, there exists a* phase transition *during training:*

- ***Attention scanning***. *At the beginning of the training, $\gamma(t) = O(\eta_Y t/K)$ and $B_n(t) \approx \frac{1}{4}\ln K^{-1}(\rho_0^4 + 2\eta_Y\eta_Z t^2) = O(\ln t)$. This means that the growth factor for dominant token $l_0$ is (sub-)linear: $\chi_{l_0}(t) \geq e^{2f_{nl_0}^2(0)B_n(t)} \approx [K^{-1}(\rho_0^4 + 2\eta_Y\eta_Z t^2)]^{0.5f_{nl_0}^2(0)}$, and the attention on less co-occurred token drops gradually.*

- **Attention snapping.** When $t \geq t_0 := 2(1 + \delta')K \ln M/\eta_Y$ with $\delta' = \Theta(\frac{\ln \ln M}{\ln M})$, $\gamma(t) = O\left(\frac{K \ln(\eta_Y t/K)}{\eta_Y t}\right)$ and $B_n(t) = O(\ln \ln t)$. Therefore, while $B_n(t)$ still grows to infinite, the growth factor $\chi_{l_0}(t) = O(\ln t)$ grows at a much slower *logarithmic rate*.

*Proof.* Since every next token $n$ shares the same query token $m$, we omit the subscript $m$ and let $z_l := z_{ml}$.

We start from the two following assumptions:

$$\dot{z}_l = -C_0^{-1}\eta_Z\gamma(t)\exp(4z_l) \tag{118}$$

$$\xi_n(t) = C_0^{-1}\gamma(t)\exp(4z_l) \tag{119}$$

Given that, we can derive the dynamics of $z_l(t)$ and $\xi_n(t)$:

$$\exp(-4z_l)\dot{z}_l = -C_0^{-1}\eta_Z\gamma(t) \tag{120}$$

$$\mathrm{d}\exp(-4z_l) = 4C_0^{-1}\eta_Z\gamma(t)\mathrm{d}t \tag{121}$$

$$\exp(-4z_l) = 4C_0^{-1}\eta_Z\int_0^t \gamma(t')\mathrm{d}t' + 1 \quad (\text{use } z_l(0) = 0) \tag{122}$$

Let $\Gamma(t) := \eta_Z \int_0^t \gamma(t')\mathrm{d}t'$, then $\Gamma(0) = 0$ and $\mathrm{d}\Gamma(t) = \eta_Z\gamma(t)\mathrm{d}t$. Therefore, we have

$$\xi_n(t) = C_0^{-1}\gamma(t)\exp(4z_l) = \frac{\gamma(t)}{C_0 + 4\Gamma(t)} \tag{123}$$

and thus $B_n(t) := \eta_Z \int_0^t \xi_n(t')\mathrm{d}t'$ can be integrated analytically, regardless of the specific form of $\gamma(t)$:

$$B_n(t) = \eta_Z \int_0^t \frac{\gamma(t')\mathrm{d}t'}{C_0 + 4\Gamma(t)} = \int_0^t \frac{\mathrm{d}\Gamma}{C_0 + 4\Gamma} = \frac{1}{4}\ln(C_0 + 4\Gamma(t)) \tag{124}$$

Recall that $\gamma(t) = \frac{(M-1)^2 h(t/K)}{M-1+\exp(Mh(t/K))}$ (Theorem 1). Note that $h$ (if treated in continuous time step) is strictly monotonically increasing and satisfies $h(0) = 0, \mathrm{d}h(t/K) = \eta_Y(M - 1 + \exp(Mh(t/K)))^{-1}\mathrm{d}t/K$ (Lemma 6 and Lemma 7), we can let $\gamma(h) := \frac{(M-1)^2 h}{M-1+\exp(Mh)}$ and get:

$$\Gamma(t) := \eta_Z \int_{t=0}^t \gamma(t')\mathrm{d}t' \tag{125}$$

$$= \eta_Z K \int_{h(0)}^{h(t/K)} \gamma(h') \cdot \frac{M - 1 + \exp(Mh')}{\eta_Y} \cdot \mathrm{d}h' \tag{126}$$

$$= \frac{\eta_Z}{\eta_Y}K(M-1)^2 \int_{h(0)}^{h(t/K)} h'\mathrm{d}h' \tag{127}$$

$$= \frac{\eta_Z}{\eta_Y} \cdot \frac{K(M-1)^2}{2}h^2(t/K) \tag{128}$$

Therefore, $B_n(t)$ has a close form with respect to $h$, regardless of the specific form of $h(t)$:

$$B_n(t) = \frac{1}{4}\ln\left(C_0 + 2\frac{\eta_Z}{\eta_Y}K(M-1)^2 h^2(t/K)\right) \tag{129}$$

**(1)** When $t < t_0' := K \ln(M)/\eta_Y$, from Lemma 9 we have $h(t/K) = (1 + o(1)) \cdot \eta_Y t/(MK)$. We neglect the $o(1)$ term and denote $\nu := \eta_Y/\eta_Z$, then we have when $t \leq t_0'$:

$$B_n(t) = \frac{1}{4}\ln\left(C_0 + \frac{2(M-1)^2}{\nu K M^2}\eta_Y^2 t^2\right) \tag{130}$$

And $B_n(t_0') = \frac{1}{4}\ln\left(C_0 + 2K(M-1)^2 M^{-2}\nu^{-1}\ln^2(M)\right)$.

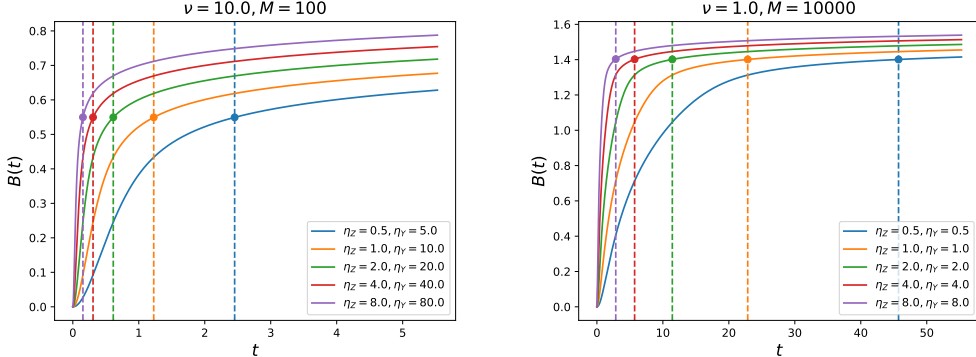

Figure 9: Numerical simulation of $B_n(t)$ with changing $\eta_Z$ and fixed $\nu = \eta_Z/\eta_Y$. The dotted line denotes the transition time $t_0$, and $B_n(t_0)$ marked with the solid dot is independent of $\eta_Z$.

**(2)** Similarly, when $t > t_0 := 2(1 + \omega_1)K \ln M/\eta_Y$ with $\omega_1 = \Theta(\ln \ln M/\ln M)$ is defined in Lemma 8, from Lemma 9 we have $h(t/K) = (1 + o(1))\ln(M\eta_Y t/K)/M$. We neglect the $o(1)$ term and get when $t > t_0$:

$$B_n(t) = \frac{1}{4} \ln \left( C_0 + \frac{2K(M-1)^2}{\nu M^2} \ln^2(M\eta_Y t/K) \right) \tag{131}$$

From this we know $B_n(t_0) = \frac{1}{4}\ln(C_0 + 2K(M - 1)^2 M^{-2}\nu^{-1} \ln^2(2(1 + \omega_1)M \ln M))$. It's interesting to find that $B_n(t_0)$ just depends on $K, M$ and $\nu$, and thus fixing $\nu$ and changing $\eta_Z$ will not influence the value of $B_n(t_0)$, which means that the main difference between $B_n$ is arises at the stage $t > t_0$. This matches the results in Fig. 9.

**(3)** Finally, we estimate $B_n(t)$ when $t$ is large. When $\nu$ is fixed and $t \gg (M\eta_Y)^{-1} \exp(1/\sqrt{2\nu})$, we have

$$B_n(t) = (1 + o(1)) \cdot \left[ \frac{1}{2} \ln \ln(M\eta_Y t/K) + \frac{1}{4} \ln(2K(M-1)^2 M^{-2}\nu^{-1}) \right] \tag{132}$$

$$= \Theta \left( \ln \ln(\frac{M\eta_Z \nu t}{K}) - \ln(\frac{\nu}{K}) \right) \tag{133}$$

Therefore, from Eqn. 133 we get:

**(a)** Fix $\nu$, larger $\eta_Z$ result in larger $B_n(t)$ and sparser attention map.

**(b)** Fix $\eta_Z$, larger $\nu$ (i.e., larger $\eta_Y$) result in smaller $B_n(t)$ and denser attention map since $\ln \ln(x)$ is much slower than $\ln(x)$.

These match our experimental results in the main paper (Fig. 6). $\qquad \square$

## G  Experiments

We use WikiText [25] dataset to verify our theoretical findings. This includes two datasets, Wiki-Text2 and WikiText103. We train both on 1-layer transformer with SGD optimizer. Instead of using reparameterization $Y$ and $Z$ (Sec. 3.1), we choose to keep the original parameterization with token embedding $U$ and train with a unified learning rate $\eta$ until convergence. Fig. 10 shows that the averaged entropy of the self-attention map evaluated in the validation set indeed drops with when the learning rate $\eta$ becomes larger.

Note that in the original parameterization, it is not clear how to set $\eta_Y$ and $\eta_Z$ properly and we leave it for future work.

Furthermore, we use the recall-threshold relationship to reshow that smaller $\eta_Y/\eta_Z$ and larger $\eta_Z$ will result in a sparser self-attention map as Fig. 11 and Fig. 12. Here we use some thresholds to retag every entry of the attention as a 0-1 value based on its softmax value for every query token, and then calculate the recall value by the average value of the proportion of the distinct tokens with new labels

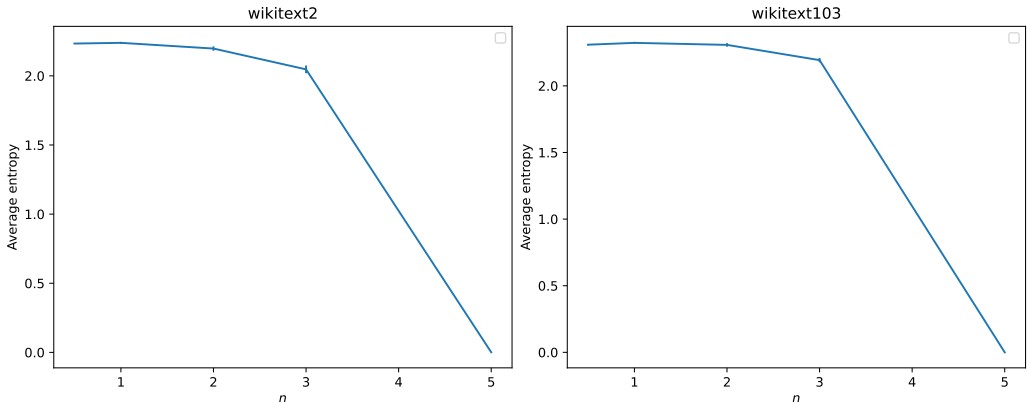

Figure 10: Average self-attention map entropy over the validation sets in 1-layer transformer after training, when the learning rate $\eta_Y$ and $\eta_Z$ changes. Note that higher learning rate $\eta$ leads to higher $B_n(t)$ and thus low entropy (i.e., more sparsity), which is consistent with our theoretical finding (Sec. 6). All the experiments are repeated in 5 random seeds. Error bar with 1-std is shown in the figure.

equal to 1 to the total number of distinct tokens. In the figures, the horizontal coordinates denote the threshold values around $1/M$ for different last/next tokens setting, and the vertical coordinates denote the recall rate. The dataset is **Syn-Medium** mentioned in Section 8, and every data point in the figures is the mean value over 10 seeds. It's obvious that a sparser attention map will result in a slower descent rate of the recall-threshold line since sparser attention corresponds to fewer distinct tokens with higher attention weights, and the results of Fig. 11 and Fig. 12 match that of Fig. 6.

## H   Technical Lemma

**Lemma 18.** *Let $\boldsymbol{h} = [h_1, h_2, \ldots, h_M]^\top \in \mathbb{R}^M$ is some $M$-dimensional vector, $\boldsymbol{h}_X := [h_{x_1}, \ldots, h_{x_{T-1}}]^\top \in \mathbb{R}^{T-1}$ is a vector selected by input sequence $X$, then given event $x_T = m, x_{T+1} = n$, there exists some $\boldsymbol{q}_{m,n} = [q_{1|m,n}, q_{2|m,n}, \ldots, q_{M|m,n}]^\top \in \mathbb{R}^M$ so that $\boldsymbol{q} \geq 0$ and*

$$\frac{1}{T-1}X^\top \boldsymbol{h}_X = \sum_{l=1}^{M} q_{l|m,n} h_l \boldsymbol{e}_l = \boldsymbol{q}_{m,n} \circ \boldsymbol{h} \tag{134}$$

$$\frac{1}{T-1}X^\top \mathrm{diag}(\boldsymbol{h}_X)X = \sum_{l=1}^{M} q_{l|m,n} h_l \boldsymbol{e}_l \boldsymbol{e}_l^\top = \mathrm{diag}(\boldsymbol{q}_{m,n} \circ \boldsymbol{h}) \tag{135}$$

*where $q_{l|m,n}$ satisfies $\sum_{l=1}^{M} q_{l|m,n} = 1$. And with probability at least $1 - \delta$ we have*

$$\max\left(0, \mathbb{P}(l|m,n) - \sqrt{\frac{\ln(2/\delta)}{2(T-1)}}\right) \leq q_{l|m,n} \leq \mathbb{P}(l|m,n) + \sqrt{\frac{\ln(2/\delta)}{2(T-1)}} \tag{136}$$

*And thus $q_{l|m,n} \to \mathbb{P}(l|m,n)$ when $T \to +\infty$.*

*Proof.* Given that $x_T = m$ and $x_{T+1} = n$, then we have

$$\frac{1}{T-1}X^\top \boldsymbol{h}_X = \frac{1}{T-1}\sum_{t=1}^{T-1} h_{x_t}\boldsymbol{x}_t = \sum_{l=1}^{M}\left(\frac{1}{T-1}\sum_{t=1}^{T-1}\mathbb{I}[x_t = l]\right) h_l\boldsymbol{e}_l =: \sum_{l=1}^{M} q_{l|m,n}h_l\boldsymbol{e}_l \tag{137}$$

And similar equations hold for $\frac{1}{T-1}X^\top \mathrm{diag}(\boldsymbol{h}_X)X$. Then we consider the case that the previous tokens are generated by conditional probability $\mathbb{P}(l|m,n)$ as the data generation part, so $\mathbb{I}[x_t = l], \forall t \in [T-1]$ are *i.i.d.* Bernoulli random variables with probability $\mathbb{P}(l|m,n)$ and $T q_{l|m,n}$ satisfies binomial distribution. By Hoeffding inequality, we get

$$\mathbb{P}(|q_{l|m,n} - \mathbb{P}(l|m,n)| \geq t) \leq 2\exp(-2(T-1)t^2) \tag{138}$$

Then we get the results by direct calculation. $\qquad\square$

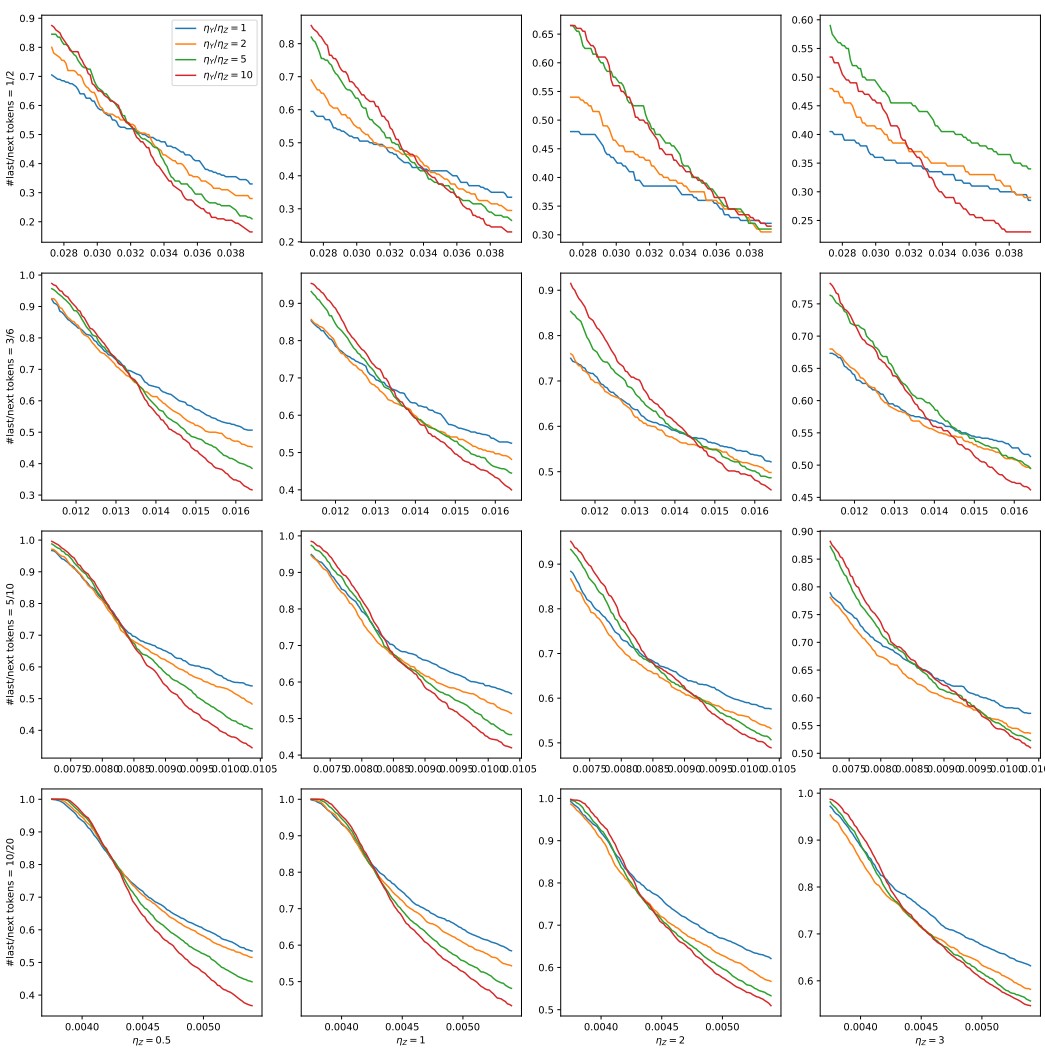

Figure 11: Recall value of attention on all distinct tokens versus threshold with changing learning rate ratio $\eta_Y/\eta_Z$. Smaller $\eta_Y/\eta_Z$ corresponds to a smaller descent rate and thus sparser attention.

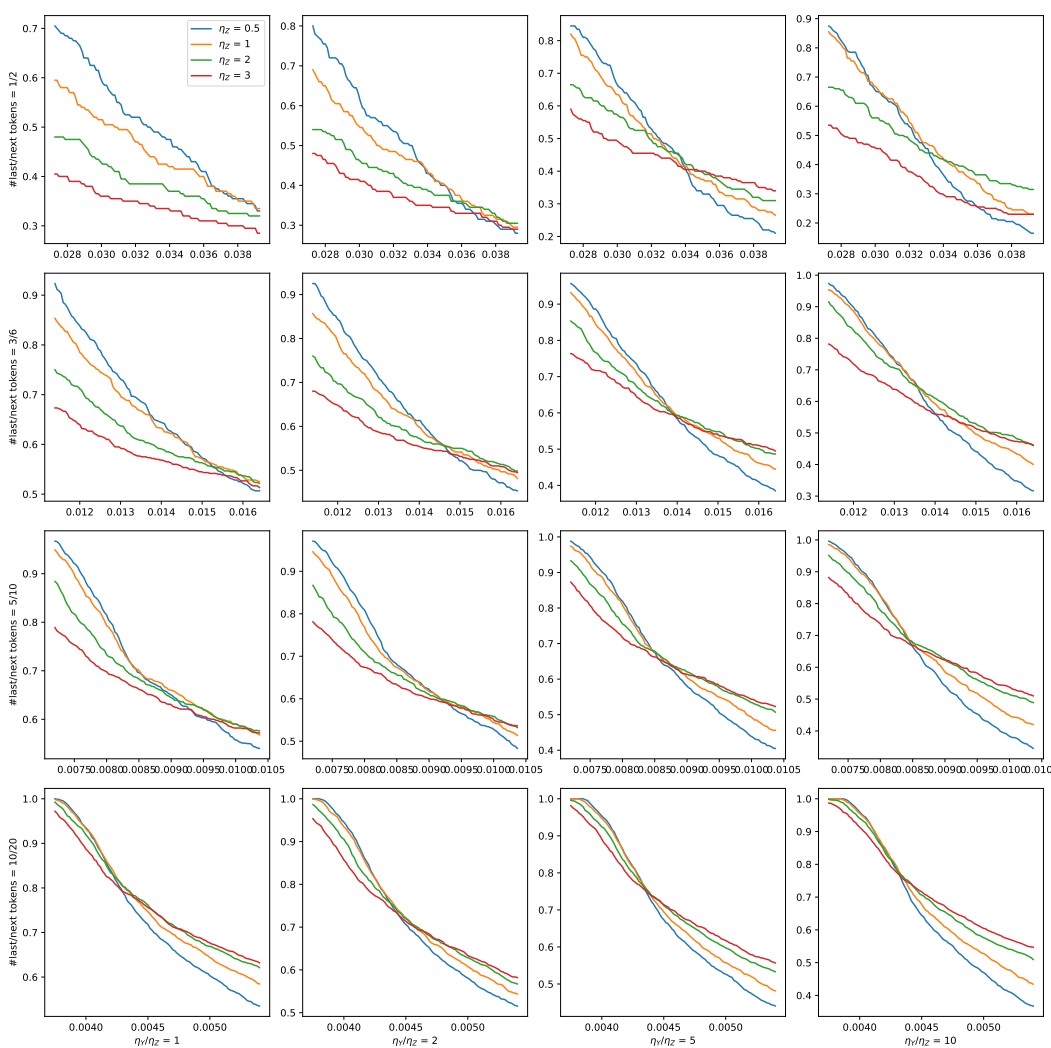

Figure 12: Recall value of attention on all distinct tokens versus threshold with changing learning rate $\eta_Z$. Larger $\eta_Z$ corresponds to a smaller descent rate and thus sparser attention.

