# Scan and Snap: Understanding Training Dynamics and Token Composition in 1-layer Transformer

## Abstract

Transformer architecture has shown impressive performance in multiple research domains and has become the backbone of many neural network models. However, there is limited understanding on how it works. In particular, with a simple predictive loss, how the representation emerges from the gradient *training dynamics* remains a mystery. In this paper, for 1-layer transformer with one self-attention layer plus one decoder layer, we analyze its SGD training dynamics

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

 in $\tilde{\boldsymbol{u}}_T$: $\tilde{\boldsymbol{u}}_T = U^\top \text{LN}(X^\top \boldsymbol{b}_T)$, where $\text{LN}(\boldsymbol{x}) := \boldsymbol{x}/\|\boldsymbol{x}\|_2$. NormFormer [58] also leverages this setting. Our analysis can also be extended to standard LayerNorm [6], which also subtracts the mean of $\boldsymbol{x}$. Empirically $\tilde{\boldsymbol{u}}_T$ or $W_V \tilde{\boldsymbol{u}}_T$ is normalized (instead of $X^\top \boldsymbol{b}_T$) and here we use an approximation to facilitate analysis.

**Objective**. We maximize the likelihood of predicted $(T + 1)$-th token using cross entropy loss:

$$\max J := \mathbb{E}_{\mathcal{D}} \left[ \boldsymbol{u}_{x_{T+1}}^\top W_V \tilde{\boldsymbol{u}}_T - \log \sum_l \exp(\boldsymbol{u}_l^\top W_V \tilde{\boldsymbol{u}}_T) \right] \tag{2}$$

We call $x_T = m$ as the **last token** of the sequence, and $x_{T+1} = n$ as the **next token** to be predicted. Other tokens $x_t$ ($1 \le t \le T - 1$) that are encoded in $X$ are called **contextual tokens**. Both the contextual and last tokens can take values from 1 to $M$ (i.e., $m \in [M]$) and next token takes the value from 1 to $K$ (i.e., $n \in [K]$) where $K \le M$.

## 3.1 Data Generation

Next we specify a data generation model, named *sequence class*, for our analysis.

**Sequence Class.** We regard the input data as a mixture of multiple *sequence classes*. Each sequence class is characterized by a triple $s_{m,n} := (\mathbb{P}(l|m, n), m, n)$. To generate a sequence instance from the class, we first set $x_T = m$ and $x_{T+1} = n$, and then generate the contextual tokens with conditional probability $\mathbb{P}(l|m, n)$. Let $\text{supp}(m, n)$ be the subset of token $l$ with $\mathbb{P}(l|m, n) > 0$.

In this work, we consider the case that given a next token $x_{T+1} = n$, the corresponding sequence always ends with a specific last token $x_T = m =: \psi(n)$. This means that we could index sequence class with next token $x_{T+1} = n$ alone: $s_n := (\mathbb{P}(l|\psi(n), n), \psi(n), n)$, $\mathbb{P}(l|m, n) = \mathbb{P}(l|n)$ and $\text{supp}(n) := \text{supp}(\psi(n), n)$.

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

209 we omit the subscript $m$ in $\boldsymbol{z}_m$ and use $z_l$ to represent $z_{ml}$.

210 **Theorem 2** (Fates of contextual tokens). *Let* $G_{CT}$ *be the set of common tokens (CT), and* $G_{DT}(n)$
211 *be the set of distinct tokens (DT) that belong to next token* $n$. *Then if Assumption 2 holds, under the*
212 *self-attention dynamics (Eqn. 10), we have:*

213      • *(a) for any distinct token* $l \in G_{DT}(n)$, $\dot{z}_l > 0$;

214      • *(b) if* $|G_{CT}| = 1$, *then for the single common token* $l \in G_{CT}$, $\dot{z}_l < 0$.

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

Under these approximations, its dynamics (Eqn. 10) can be written as follows:

$$\dot{z}_l = \eta_Z \gamma(t) \sum_{n \in \psi^{-1}(m)} f_{nl} \sum_{n' \neq n} \beta_{nn'}(f_{nl}^2 - 1)f_{nl'} \approx -K\rho_0^{-4}\eta_Z\gamma(t)e^{4z_l}, \quad \xi_n(t) \approx K\rho_0^{-4}\gamma(t)e^{4z_l}$$

(13)

Surprisingly, we now find a *phase transition* by combining the rate change of $\gamma(t)$ in Theorem 1:

**Theorem 4** (Phase Transition in Training)**.** *If the dynamics of the single common token $z_l$ satisfies $\dot{z}_l = -K\rho^{-4}\eta_Z\gamma(t)e^{4z_l}$ and $\xi_n(t) = K\rho^{-4}\gamma(t)e^{4z_l}$, then we have:*

$$B_n(t) = \begin{cases} \frac{1}{4}\ln\left(\rho_0^4/K + \frac{2(M-1)^2}{KM^2}\eta_Y\eta_Z t^2\right) & t < t'_0 := \frac{K\ln M}{\eta_Y} \\ \frac{1}{4}\ln\left(\rho_0^4/K + \frac{2K(M-1)^2}{M^2}\frac{\eta_Z}{\eta_Y}\ln^2(M\eta_Y t/K)\right) & t \geq t_0 := \frac{2(1+o(1))K\ln M}{\eta_Y} \end{cases} \quad (14)$$

*As a result, there exists a* phase transition *during training:*

- ***Attention scanning**. At the beginning of the training, $\gamma(t) = O(\eta_Y t/K)$ and $B_n(t) \approx \frac{1}{4}\ln K^{-1}(\rho_0^4 + 2\eta_Y\eta_Z t^2) = O(\ln t)$. This means that the growth factor for dominant token $l_0$ is (sub-)linear: $\chi_{l_0}(t) \geq e^{2f_{nl_0}^2(0)B_n(t)} \approx [K^{-1}(\rho_0^4 + 2\eta_Y\eta_Z t^2)]^{0.5f_{nl_0}^2(0)}$, and the attention on less co-occurred token drops gradually.*

- ***Attention snapping**. When $t \geq t_0 := 2(1+\delta')K\ln M/\eta_Y$ with $\delta' = \Theta(\frac{\ln\ln M}{\ln M})$, $\gamma(t) = O\left(\frac{K\ln(\eta_Y t/K)}{\eta_Y t}\right)$ and $B_n(t) = O(\ln\ln t)$. Therefore, while $B_n(t)$ still grows to infinite, the growth factor $\chi_{l_0}(t) = O(\ln t)$ grows at a much slower logarithmic rate.*

This gives a few insights about the training process: **(a)** larger learning rate $\eta_Y$ of the decoder $Y$ leads to shorter phase transition time $t_0 \approx 2K\ln M/\eta_Y$, **(b)** scaling up both learning rate ($\eta_Y$ and $\eta_Z$) leads to larger $B_n(t)$ when $t \to +\infty$, and thus sparser attention maps, and **(c)** given fixed $\eta_Z$, small learning rate $\eta_Y$ leads to larger $B_n(t)$ when $t \geq t_0$, and thus sparser attention map. Fig. 3 shows numerical simulation results of the growth rate $\chi_l(t)$. Here we set $K = 10$ and $M = 1000$, and we find smaller $\eta_Y$ given fixed $\eta_Z$ indeed leads to later transition and larger $B_n(t)$ (and $\chi_l(t)$).

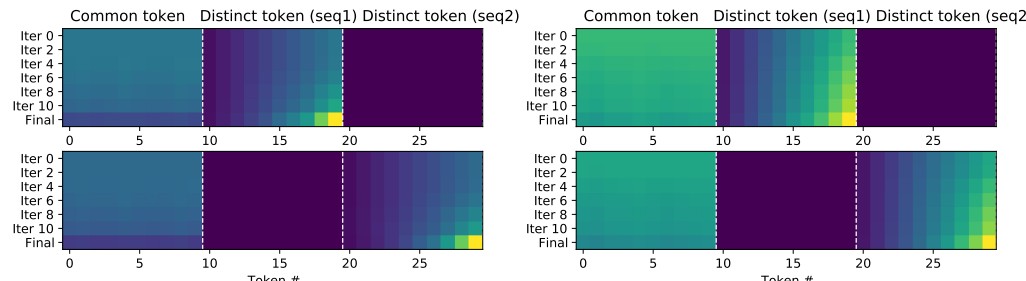

Figure 4: Visualization of $\boldsymbol{c}_n$ ($n = 1, 2$) in the training dynamics of 1-layer Transformer using SGD on `Syn-Small` setting. Top row for last token $n = 1$ and bottom row for last token $n = 2$. **Left:** SGD training with $\eta_Y = \eta_Z = 1$. Attention pattern $\boldsymbol{c}_n$ becomes sparse and concentrated on highest $\mathbb{P}(l|n)$ (rightmost) for each sequence class (Theorem 3). **Right:** SGD training with $\eta_Y = 10$ and $\eta_Z = 1$. With larger $\eta_Y$, convergence becomes faster but the final attention maps are less sparse (Sec. 6).

## 7    Discussion and Limitations

**Positional encoding**. While our main analysis does not touch positional encoding, it can be added easily following the relative encoding schemes that adds a linear bias when computing self attention (E.g., T5 [56], ALiBi [53], MusicTransformer [33]). More specifically, the added linear bias $\exp(z_{ml} + z_0) = \exp(z_{ml})\exp(z_0)$ corresponds to a prior of the contextual token to be learned in the self-attention layer.

**Residue connection**. Residue connection can be added in the formulation, i.e., $\hat{\boldsymbol{u}}_T = \text{LN}(\text{LN}(\tilde{\boldsymbol{u}}_T) + \boldsymbol{u}_{x_T})$, where $\tilde{\boldsymbol{u}}_T$ is defined in Eqn. 1, and $\hat{\boldsymbol{