# OpenReview forum: "Scan and Snap: Understanding Training Dynamics and Token Composition in 1-layer Transformer"
_NeurIPS.cc/2023/Conference — NeurIPS 2023 poster_

### Official Review · Reviewer_nH6i · 2023-06-10

**Soundness:** 4 excellent
**Presentation:** 4 excellent
**Contribution:** 4 excellent
**Rating:** 7
**Confidence:** 3

**Summary:**

The key contribution of this paper is that the training dynamics of a 1-layer Transformer model is demystified theoretically and empirically. The authors have found that the self attention operator performs a discriminative scanning algorithm on the input tokens, attending more on distinct tokens and focusing less on common tokens. They also demonstrate that there is a phase transition to an attention snapping step.

**Strengths:**

- The Transformer's self attention operator has been effective yet a black box in many applications. The authors' attempt in providing a theoretical explanation of the learning process is valuable.
- A simplified setting helps better understanding.
- Clear and straightforward analysis that aligns well with the paper's findings.

**Weaknesses:**

I do not see any critical weakness in this paper. However, more analysis might be needed on multi-layer Transformer settings. Although the authors have provided the attention patterns of a multi-layer model in Figure 7, it only covers a single layer in the model. Considering that most transformer-based models have multiple layers, it would be interesting to see if there are any different learning patterns in different layer depths.

**Questions:**

Extending the discussion on multiple-layer settings from the weakness section,
1. Will there be a specific pattern regarding the phase transition across different layers? e.g. slower phase transition in the last layer etc.
2. Is there a possibility that each layer may specialize on different roles? e.g. initial layers "scan" while latter layers "snap"
3. (optional) Will these findings be consistent across different domains that use Transformer models? e.g. computer vision models, graph transformers etc.

**Limitations:**

The authors have implicitly discussed the limitations. However, the broader impact was not discussed. Please add a broader impact section.

---

> ### Author Rebuttal · Authors · 2023-08-06
>
> We thank the reviewer for the insightful and encouraging comments! We appreciate the suggestions! Here are the answers to the questions.
>
> > Attention patterns in multi-layer model
>
> We haven’t systematically analyzed the attention patterns in multi-layer models yet. Initial experiments show that the attention score in the top-layer is not as sparse as the bottom-layer attention patterns. We will leave it for future work.
>
> > Will there be a specific pattern regarding the phase transition across different layers? e.g. slower phase transition in the last layer etc.
>
> We haven’t done multilayer analysis systematically yet. The intuition is that the sparsity attention patterns in the lower-layer will lead to combinations of tokens that co-occur a lot, which then are further combined in the higher layer. So it is possible that top layer attention will freeze after bottom layer attention (and slower phase transition), since they provide the building blocks for the top.
>
> > Is there a possibility that each layer may specialize in different roles? e.g. initial layers "scan" while latter layers "snap"
>
> Yeah it is possible that different layers are in different stages, as mentioned in the previous answers.
>
> > (optional) Will these findings be consistent across different domains that use Transformer models? e.g. computer vision models, graph transformers etc.
>
> It is possible since for different domains, the token combinations hierarchy are different. E.g., in vision, each patch may carry less amount of information, information tends to spread evenly across the image, and a deep and balanced hidden hierarchy is needed; while in language, a few tokens may carry the majority of information, and tokens may have nonlocal relationships, etc. Graphs from different domains may contain very different feature structures. Our goal is not to design specific models for specific domains, but to characterize how features are learned in (multi-layer) Transformer architecture, under different data distribution.
>
> > Please add a broader impact section
>
> Thanks for your suggestions. Here is one tentative version:
>
> Our work gives a framework to analyze the dynamics of a 1-layer transformer using the tools of nonlinear dynamics systems. Our finding that self-attention pays sparse attention to a subset of distinct tokens that co-occur a lot with the query can have many implications. First, it helps understand the inductive bias of self-attention, reduce spurious correlation (by removing tokens that are not likely to be attended to) and improve generalization. Second, it can also help understand the mechanism of hierarchical token combination, and potentially open the door of understanding multi-layer transformers. The two stage training process, namely “scan and snap”, can also be an important tool to understand how the multi-layer transformer works in practice.

---

> > ### Comment · Reviewer_nH6i · 2023-08-10
> > **Thank you for the rebuttal**
> >
> > It's a shame that the authors did not provide quantitative analyses to support their responses, but I understand that analyzing multi-layer settings would require more than a week. Other than that, the authors tried to provide nice intuitions for my questions. Thank you.
> > Also, while I do agree with the other reviewers that a 1-layer setting might not be enough to fully understand how Transformer works, this paper is definitely a good stepping stone for demystifying the Transformer model. I don't think I've seen a paper that does this, and I believe this paper deserves attention.

---

> > > ### Author Response · Authors · 2023-08-11
> > > **Thanks for your support!**
> > >
> > > We really appreciate your support! We hope this paper can shed some lights on the underlying mechanism of self-attention, in particular towards the quantitative and rigorous descriptions on where its sparsity comes from and how it is related to data distribution. Our work also contributes to the novel techniques used to analyze the behavior of self-attention and neural networks. It is a hard journey but we will keep trying.
> > >
> > > A systematic understanding of multi-layer settings would require a nontrivial amount of additional work and is out of the scope of this submission. Here we show our (ungrounded) intuitions to the best of our knowledge to quench the needs of the reviewers. We will try our best to show some initial empirical results during the discussion period (while it is not guaranteed).
> > >
> > > Thanks again!

---

### Official Review · Reviewer_ZSxf · 2023-06-26

**Soundness:** 2 fair
**Presentation:** 3 good
**Contribution:** 2 fair
**Rating:** 6
**Confidence:** 4

**Summary:**

This paper analyzes a simple architecture of 1-layer transformer’s SGD training dynamics for the task of next token prediction. The authors prove that self-attention acts as a discriminative scanning algorithm, with an inductive bias to favor unique key tokens that frequently co-occur with the query tokens.


**Strengths:**

This work aims at understanding how Transformers work, and tries to understand the training dynamics in a simplified setup with 1-layer transformer. The work conducted both theoretical analysis and empirical experiments.

**Weaknesses:**

1. While it is an important research question to understand how Transformers work, this work only offers some observations regarding how attention weights change over time on two different types of context tokens. The setup is too simplistic to generalize to real-world scenarios. The conclusions depend on a lot of assumptions made in the paper.

2. Several assumptions made in the theoretical analysis are not clearly justified:
a) The decoder layer learns much faster than the self-attention layer. The learning rates are hyper-parameters and in typical settings, they are the same for the decoder layer and the self-attention layer.
b) The weak correlation assumption in Assumption 2 is not explained at all under what condition it is valid.

3. The empirical experiments don’t share the same conditions as the theoretical analysis: e.g. batch size 1 assumption and much larger learning rate in decoder layer than self-attention.


**Questions:**

1. The assumptions used in the analysis need further justifications regarding the learning rate of Y and Z, and the weak correlation assumption in Assumption 2.
2. The Residual connections are such a key component of Transformers. Neglecting them in the analysis leaves the conclusions questionable in generalizing to the actual architectures used in Transformer experiments.

**Limitations:**

It is still unclear how the conclusions and observations from this paper helps to understand how Transformers work. While the research direction is generally good, I don’t find much surprise in the result that Transformers sparsify and focus attention on unique tokens.

---

> ### Author Rebuttal · Authors · 2023-08-06
>
> We thanks the reviewer to give insightful comments!
>
> > The assumptions used in the analysis need further justifications regarding the learning rate of Y and Z, and the weak correlation assumption in Assumption 2.
>
> We explain the intuition of the Assumption 2 (Weak correlation) as below:
>
> **Regarding upper bound**.  $|\lambda_{\max}(E)| < 1/K$ ensures that $f_i$ are almost orthogonal to each other since $E_{ij} = f_i^\top f_j < 1/K$ holds for all $i \neq j \in [K]$. And this means that for different sequence classes, the conditional probability of each contextual token given the query/last token is quite different from each other. This scenario is common in real cases, in particular when the vocabulary size $M$ becomes large, and each sequence class may only relate to a small portion of the tokens.
>
> **Regarding the lower bound**. $|\lambda_i(E)| \ge 6/\sqrt{M}$ illustrates that all tokens can be common tokens with small conditional probabilities, resembling a uniform distribution. This condition is only used in formula (93), which proves property (a) of Theorem 1. Specifically, $E_{nn}'$ and $u_{nk}$ in (93) are hard to be estimated, and we use this lower bound to prevent the worst case and prove that $\xi_n > 0$ always holds.
>
> Additionally, even if Assumption 2 doesn't hold, Theorem 5 in the Appendix shows that formula (9) and property (b) in Theorem 1 still hold. We just need Assumption 2 to ensure $\xi_n > 0$ and to maintain the favorable properties of distinct tokens as discussed in Sections 5 and 6.
>
> > The empirical experiments don’t share the same conditions as the theoretical analysis. e.g. batch size 1 assumption and much larger learning rate in decoder layer than self-attention.
>
> As mentioned in the reply to reviewer **sxfK**, Lemma 1 gives a gradient formula when batchsize = 1 for simplicity, and the gradient of batchsize > 1 can be easily obtained by summing over different samples, and our framework applies to large batch sizes as well. We indeed tested the case where the learning rate of the decoder is larger than self-attention (see Fig. 6) and the trend is consistent with our theoretical findings.
>
> > I don’t find much surprise in the result that Transformers sparsify and focus attention on unique tokens.
>
> Note that as mentioned in reviewer **TXYd**, previous and concurrent works [1][2], published in ICLR/ICML, gave theoretical justification on why Transformers gradually attend more to distinct key tokens, under SGD training of 1-layer transformer. In comparison, our work analyzes more rich phenomena in 1-layer transformers that can be categorized as frequency and discriminative bias, which has not been brought up before. For example, we analyze multi-class settings, we connect sparse attention patterns with the co-occurrence frequency of contextual token and query, characterize relative growth in details for all tokens that are relevant to the class label (rather than only compare relevant with irrelevant tokens [1][2]), characterize the complete training dynamics (rather than initial steps [2]) and summarize two stage behaviors of attention scores toward convergence, etc.
>
> We want to emphasize that we want to provide a theoretical framework and lay the foundation of a rigorous theoretical understanding of Transformer. Most of the time there should be no surprise and the theory should explain what happens in practice, but in a more fine-grained and quantitative manner. Such quantitative explanations lead to new discoveries that could go beyond empirical findings and intuitions.
>
> > Residual connections
>
> First, note that none of the previous/concurrent works [1][2][3] analyze residual connections in their training dynamics. Here [1][2] are accepted in ICLR'23/ICML'23.
>
> In addition, our framework can already incorporate residual connection as part of the input $f_n$ to the decoder layer $Y$ (Sec. 7). We could further characterize the scale of the coefficients $\beta_{nn’}$:  when $\psi(n) \neq \psi(n’)$, i.e., the last/query tokens of seq class $n$ and $n’$ are different, $\beta_{nn’}$ should be much smaller. In this case, conclusions about distinct and common tokens can be further refined. For simplicity, the main paper focuses on cases without residual connection.
>
> **Reference**
>
> [1] Li et al., "A Theoretical Understanding of shallow Vision Transformers: Learning, Generalization, and Sample Complexity", ICLR 2023.
>
> [2] Oymak et al., "On the Role of Attention in Prompt-tuning", ICML 2023.
>
> [3] Tarzanagh et al., "Max-Margin Token Selection in Attention Mechanism", https://arxiv.org/pdf/2306.13596.pdf

---

> > ### Comment · Reviewer_ZSxf · 2023-08-13
> >
> > I appreciate the effort from the authors to explain the intuitions behind the assumptions, and point out the fact that under empirical study, the conclusions still hold even outside of the assumptions made in theoretical analysis. I would suggest the authors add some of the intuitions and justifications to the paper to make the paper easier to digest for readers.
> >
> > However, I don't think it makes sense to justify this paper using the fact that other papers made similar assumptions were accepted earlier. Please focus on making this paper higher quality in terms of scientific merit and readability for the community.
> >
> > Upon second reading the entire paper, I am convinced that it is scientifically valuable to theoretically/empirically observe: 1) distinct tokens gain attention weights (again, not surprising to me from a statistical learning point of view), and 2) the phase transition from a winner-take-all for the most distinct token to a (nearly) frozen sparse attention pattern with a logarithmical growth rate. In particular, 2) is quite interesting in itself, and I hope it can motivate more work along this line of research.
> >
> > Therefore, I decide to revise my Rating to 6 and suggest the authors make some effort to clarify the intuitions behind the assumptions in the paper.

---

> > > ### Author Response · Authors · 2023-08-15
> > > **Thanks!**
> > >
> > > Thanks for your reply! We really appreciate that you think the work is scientifically valuable. We are happy to see that the score has been revised! We sincerely hope that this work can motivate more works along the line of research, which is precisely our motivation to start this project.
> > >
> > > Thanks for your suggestions. We will focus on make this work higher quality from the scientific point of view. Since reviewer **TXYd** brought up the relevant works, we reference them accordingly, analyze them in details and position our work among the proper context of literature. Thanks reviewer **TXYd** for bringing these references up, which we missed in our literature review.

---

### Official Review · Reviewer_TXYd · 2023-07-02

**Soundness:** 3 good
**Presentation:** 2 fair
**Contribution:** 3 good
**Rating:** 5
**Confidence:** 4

**Summary:**

This paper theoretically studies the SGD training dynamics of a 1-layer Transformer. Based on some assumptions, this paper introduces the frequency bias and discriminative bias of Transformers. To be more specific, this work shows that Transformers gradually attend more to distinct key tokens and less to common tokens. Later this procedure ends with a phase transition with a fixed token combination. Experiments are provided to justify the findings. I am going to update my score after the rebuttal if there are any revisions in the manuscript (if possible).

---------------------------------------------------------------------------------
After rebuttal, I increase my score from 4 to 5.

**Strengths:**

1. As far as I know, the proof technique and the framework are generally novel, at least for the Transformer architecture.
2. Some high-level insights are well presented and intuitively correct.


**Weaknesses:**

1. Some parts of this paper are not very clear. For example, I feel the decoder layer is not formally defined. In around line 92, only the self-attention layer is introduced. Does the decoder layer refer to $W_V$? I thought $W_V$ usually refers to the matrix for the value embeddings.

2. Although this paper considers a different setup, such as a different Transformer formulation, loss function, and different assumptions, the conclusion that Transformers gradually attend more to distinct key tokens is not a new one ([1] and [2] in the following) in the theoretical works. Especially the sentence "we are the first to analyze the attention dynamics and reveal its inductive bias on data input, " in line 320 is an overclaim. Here are several recent works that also study the training dynamics of Transformers.

[1] Li et al., "A Theoretical Understanding of shallow Vision Transformers: Learning, Generalization, and Sample Complexity", ICLR 2023.

[2] Oymak et al., "On the Role of Attention in Prompt-tuning", ICML 2023.

[3] Tarzanagh et al., "Max-Margin Token Selection in Attention Mechanism", https://arxiv.org/pdf/2306.13596.pdf

I know [2] and [3] are concurrent works, while [1] is an older one about training dynamics. I would like to see a comparison and discussion between the manuscript and these works.

3. I suggest presenting a table of notations somewhere in the paper since there are many notations. For example, Section 5 is difficult to follow, although I can understand the conclusion.

**Questions:**

1. Although discussing a different pair of $\eta_Y$ and $\eta_Z$ is interesting, how can it happen in practical SGD? Is it an assumption or an algorithm?

2. From my understanding, in this paper, the feed-forward layer after the self-attention layer of the Transformer is a linear layer. I am wondering whether the analysis can be extended to an MLP layer (even only one non-linear activation) with Relu or Gelu. I think feed-forward layers with non-linear activations are more common in practice.

3. I didn't fully check the proof. I found a possible typo in equation 15 in line 536. I think the last term should be $\log(\bf{1}^\top \exp(Y^\top LN(X^\top\bf{b}_T)))$.


**Limitations:**

There is no potential negative societal impact of their work.

---

> ### Author Rebuttal · Authors · 2023-08-06
>
> Thanks the reviewer for the insightful comments. Here are the answers.
>
> > I feel the decoder layer is not formally defined. In around line 92, only the self-attention layer is introduced.
>
> We define the decoder layer $Y$ and self-attention layer $Z$ after reparameterization $Y = UW_V^TU^T$ and $Z = UW_QW^T_K U^T / \sqrt{d}$. Then we study the dynamics of $Y$ and $Z$ directly.
>
> > Overclaimed sentence "we are the first to analyze the attention dynamics and reveal its inductive bias on data input"
>
> We acknowledge that previous and concurrent works[1][2][3] also show that self-attention attends to relevant tokens, and we will definitely reference the works and tune town our claim! In comparison, our work analyzes a lot more phenomena in 1-layer transformers that belong to frequency and discriminative bias, which has not been brought up before. For example, sparse attention patterns are connected with the co-occurrence frequency of contextual token and query, characterization of such connection over training including two stage behaviors of attention scores, etc. Please check the main rebuttal.
>
> >  A comparison and discussion between the manuscript and these works
>
> Please check the overall rebuttal for an overall comparison. See below for detailed comparison:
>
> **Comparison with [1]**
>
> **Setting, Assumptions and Conclusions**. [1] analyzes the SGD convergence of 1-layer ViT model (1 layer self-attention + 2 layer FFN with ReLU, with the top layer of FFN fixed as random, token embedding fixed). Under a specific binary data model in which the data label is determined by counting the number of tokens that belong to pos/neg pattern, [1] gives a generalization bound when the number of hidden nodes in FFN is large, and at the same time, shows that the self-attention attends to relevant tokens and becomes sparse (if #relevant tokens are small).
>
> In comparison, our work focuses on language models, assume broader data distribution (e.g., multiple classes, arbitrary conditional probability of token given class label) and incorporate layernorm naturally. We propose more detailed quantitative properties, e.g., attention sparsity even among relevant tokens, two-stage evolution of attention scores, with a much simpler analysis.
>
> **Techniques**. The techniques used in [1] are based on feature learning techniques applied to MLP [R4 etc]. It identifies lucky neurons if the number of hidden neurons is large enough. In comparison, our framework and analysis is much simpler by leveraging that certain nonlinear continuous dynamics systems can be integrated out analytically to yield clean solutions (e.g., Theorem 3 (Eqn. 11) and Theorem 4 (Eqn. 127)), avoiding complicated bounds in [1]. This allows us to characterize the converging behavior of self-attentions when $t->+\infty$. To our best knowledge, our framework and techniques are novel, which is also acknowledged by the reviewer.
>
> **Comparison with [2] (published on Jun. 6, after the submission deadline)**
>
> [2] focuses on 1-layer attention-based prompt-tuning, in which some parameters of the models are fixed (Wp, Wq). The analysis focuses on the initial (3x one-step) SGD trajectory, and constructs the dataset model containing specific context-relevant/context-irrelevant data, and the context-vector indicates the token relevance. As a result, [2] shows the attention becomes sparse (i.e., attending to context-relevant tokens) over time, which is consistent with ours, and shows that prompt-attention can find the relevant tokens and achieve high accuracy while self-attention/linear-attention can’t.
>
> In comparison, our work goes beyond the 2-classes model and further points out that the attention weight will be relevant to the conditional probability of the contextual tokens, which is more detailed than the sparse attention result in [2] that relies on the sparsity assumption of contextual tokens itself. We also focus on the pre-training stage (training from scratch, predicting the next token), characterize the entire trajectory under SGD for the self-attention layer, in particular its converging behavior.
>
> **Comparison with [3] (published on Jun. 23, after the submission deadline)**
>
> Compared to [2], [3] also analyzes the dynamics of the query-key matrix $W$ and the embedding of a single tunable token $p$ (often [cls] token). It makes connection between the binary classification problem with 1-layer transformer and max-margin SVM formulation, when the tokens are linearly separable. The dynamics is characterized completely, which is nice. Note here $p$ is not an attention since its norm can be shown to go to infinity over training.
>
> In comparison, our work does not learn the embedding of an individual token, but focuses on the dynamics of (all-pair) attention scores during training. We also work on multiple-class setup and do not explicitly assume the linear separability among classes.
>
> We thank the reviewer for the references and we will put the detailed comparison, as well as reference these papers in the next revision.
>
> > Nonlinearity after the decoder $Y$.
>
> Adding nonlinearity right before cross entropy loss will make our analysis a bit more complicated but not impossible. Specifically, the nonlinearity will modify the back-propagated gradient from cross entropy loss, and Theorem 1 will take a different (and maybe more complicated) form. For simplicity, we choose not to add the nonlinearity layer for this paper.
>
> > Possible typo in equation 15.
>
> We indeed missed the term $X^T$ within LN() and will fix.
>
> **Reference**
>
> [1] Li et al., "A Theoretical Understanding of Shallow Vision Transformers: Learning, Generalization, and Sample Complexity", ICLR 2023.
>
> [2] Oymak et al., "On the Role of Attention in Prompt-tuning", ICML'23 (Jun. 6)
>
> [3] Tarzanagh et al., "Max-Margin Token Selection in Attention Mechanism", arXiv’23 (Jun. 23)
>
> [R4] Z. Allen-Zhu et al, Learning and Generalization in Overparameterized Neural Networks, Going Beyond Two Layers, NeurIPS’19

---

> > ### Comment · Reviewer_TXYd · 2023-08-14
> > **Thank you for the responses. Further questions here**
> >
> > Thank the authors for the responses. Here are some further questions.
> >
> > I think the response to Weaknesses 3 about the presentation is missing. Actually, it is also one of my major concerns. I highly suggest a table to summarize the notations. Also, I suggest avoiding so many notations in the text and even in Lemmas and Theorems. It is better to show some informal lemmas first to make readers understand the conclusion being presented, especially for some lemmas. If the space is limited, the formal version of key lemmas does not have to appear in the main body. Can the authors show a table of notations in the response? I think there is no limit on the length of the response now. Also, I think this table can be put in the appendix of this paper.
> >
> > The remaining questions are about the extension and the assumption of this work.
> >
> > 1. It states that relative positional encoding can be considered in this work. Do you mean the formulation in this way: a trainable $b$ for $softmax(k_i q_l+b_i)$? What about an absolute positional encoding, such as $(v_i+b_i)softmax((k_i+b_i)\cdot (q_l+b_l))$? Or can you give a brief discussion on the formulation of absolute positional encoding you can handle?
> >
> > 2. For the positional encoding, since you assume the length of the sequence is infinite, how do you define or initialize the positional encoding? Is the positional encoding still meaningful? I think I cannot tell whether a token is far or close to the query if the length is infinite.
> >
> > 3. Why do you need the infinite length positional encoding assumption? From my understanding, by this assumption, you can study the transformer based on some stable properties. Is that right? I guess the usage of the analysis of nonlinear dynamic systems is also a reason, but I am not so familiar with that. Can you briefly discuss how to relax the analysis to a finite sequence length?
> >
> > 4. I think this work may have the potential to study some out-of-distribution generalization of the language models as future work. Can you provide a discussion?
> >
> > For other answers, I am generally satisfied, especially with the comparison with other works. A minor point is that I don't think [R4] is a feature-learning work. I think it is still NTK work. From my understanding, feature learning works have a strong assumption of the data features, based on which they can derive stronger conclusions. Also, the returned model may change much from the initialization.
> >
> > For questions raised by other reviewers, I just took a look very briefly. I don't think a one-layer Transformer is a big issue. The analysis of multi-layer fully-connected networks is already challenging, let alone Transformers. The theoretical work of Transformer just started one year. It is better for researchers first to figure out the mechanism of one-layer cases.

---

> > > ### Author Response · Authors · 2023-08-14
> > > **Notation table**
> > >
> > > We appreciate the reviewer for positive feedbacks, in particular acknowledging that our comparisons with existing works are satisfying and 1-layer setting for Transformer is an important direction to work on.
> > >
> > > We will answer the questions below.
> > >
> > > ## Notation table
> > > We apologize for the missing answers regarding to the notation table, due to limited space of the rebuttal. Please check below for the notation table, which will be put in the appendix. In our next revision, we will follow your advice to include an informal lemma/theorems to give the reader an overall picture in the main text. We are sorry for the possible confusions.
> > >
> > > | Notation | Description |
> > > |-----------|---------------|
> > > | $M$ | vocabulary size |
> > > | $T$ | sequence length |
> > > | $\mathbf{e}_k$ | One-hot vector (1 at component $k$) |
> > > | $X \in \mathbb{R}^{{T-1}\times M}$ | Input sequence (of length $T-1$) |
> > > | $\mathbf{b}_T \in \mathbb{R}^{T-1}$ | Vector of self-attention weights to predict token at time $T$ |
> > > | $\mathbf{x}_t \in \mathbb{R}^M $ | contextual token ($0 \le t \le T-2$) (one-hot) |
> > > | $\mathbf{x}_{T-1} \in \mathbb{R}^M$ | last/query token (one-hot) |
> > > | $\mathbf{x}_T \in \mathbb{R}^M $ | next token to be predicted (one-hot) |
> > > | $\mathbf{x}_t[i] \in \mathbb{R}^M$ | $i$-th training sample of token at location $t$ in the sequence |
> > > | $K$ | Number of possible choices the next token $\mathbf{x}_T$ could take |
> > > | $\boldsymbol{\alpha}(t)$ | Softmax score of the output layer |
> > > | **Learnable parameters** | |
> > > | $Y \in \mathbb{R}^{M\times M} $ | decoder layer parameters |
> > > | $Z \in \mathbb{R}^{M\times M} $ | self-attention logits |
> > > | $\mathbf{z}_m$ | $m$-th row of $Z$ (i.e., attention logits for a query/last token $m$) |
> > > | **Hyperparameters** | |
> > > | $\eta_Y$ | Learning rate of the decoder layer |
> > > | $\eta_Z$ | Learning rate of the self-attention layer |
> > > | **Token Types and Distribution** | |
> > > | $\psi(n)$ | Mapping from next token $\mathbf{x}_T = n$ to its unique last/query token |
> > > | $\psi^{-1}(m)$ | The subset of next tokens for last/query token $\mathbf{x}_{T-1}=m$ |
> > > | $\mathbb{P}(l\|m,n)$ | Conditional probability of contextual token $l$ given last token is $m$ and next token to be predicted as $n$. |
> > > | $G_{CT}$ | Subset of common tokens  |
> > > | $G_{DT}(n)$ | Subset of distinct tokens for $\mathbf{x}_T = n$ |
> > > |**Attention Score** | |
> > > | $\mathbf{\tilde c}_n \in \mathbb{R}^M$ | Unnormalized attention score given next token $\mathbf{x}_T = n$ |
> > > | $\mathbf{c}_n \in \mathbb{R}^M$ | $\ell_1$-normalized attention score given next token $\mathbf{x}_T = n$ |
> > > | $\mathbf{f}_n \in \mathbb{R}^M$ | $\ell_2$-normalized attention score given next token $\mathbf{x}_T = n$ |
> > > | $\mathbf{g} \in \mathbb{R}^M$ | Back-propagated gradient for $f_n$ |
> > > | $F$ | Input matrix of the decoder layer. Each column of $F$ is $f_n$ |
> > > |**Self-attention dynamics**| |
> > > | $r_{l/l'\|n}(t) $ | Relative gain between distinct token $l$ and $l'$ for next token $n$ |
> > > | $B_n(t)$ | Growth factor bound of the relative gain |
> > > | $\gamma(t)$ | Speed control coefficient |
> > >
> > > To make things easy to read, we will answer the remaining questions shortly in the next comment.

---

> > > > ### Author Response · Authors · 2023-08-14
> > > > **Answers to follow-up questions**
> > > >
> > > > > It states that relative positional encoding can be considered in this work. ... What about an absolute positional encoding? Or can you give a brief discussion on the formulation of absolute positional encoding you can handle?
> > > >
> > > > Yes the relative positional encoding can be considered, exactly as you mentioned (i.e., $\exp(\ldots + b(k))$), where $k$ is the distance between the contextual and the query token. Note that many modern LLMs (e.g., LLaMA v1/v2, T5, PaLM, MPT, Baichuan, etc) use relatively encoding.
> > > >
> > > > For absolute positional encoding, our current framework may not handle them out of box. The major difficulty is that the resulting $\mathbf{f}_n$, which is a (normalized) linear combination of input embeddings, also contains positional information for the decoder $Y$ to learn. Currently Theorem 1 tells us that after training $Y$ for a while with $K$ different inputs ($\mathbf{f}_1, \ldots, \mathbf{f}_K$), the back-propagated gradient can be represented as a linear combination of the $K$ inputs and combination coefficients have special properties (Theorem 1(a)). With positional encoding involved, since each $\mathbf{f}_n$ also has variance depending on the (infinitely possible) relative positions of the contextual tokens given the next token $n$, things can be a bit more complicated and may require a substantially different technique.
> > > >
> > > > Finally, recent work [R5] shows that additive positional encoding can be approximated as relative positional encoding empirically. This may suggest that relative positional encoding can be more useful.
> > > >
> > > > [R5] T-C. Chi et al, Dissecting Transformer Length Extrapolation via The Lens of Receptive Field Analysis, ACL'23.
> > > >
> > > > > Why infinite length positional encoding assumption is necessary?
> > > >
> > > > Using infinite length facilitates the analysis. Lemma 2 shows that when $T\rightarrow +\infty$, the sample-based terms (e.g., $X^\top \mathbf{b}_T$ converges to statistical terms ($\mathbf{c}_n$). This avoids additional variance due to sampling from the training set, and is much easier to deal with technically.
> > > >
> > > > On the other hand, we want to emphasize that the assumption does not impose fundamental limits. With finite sequence length, the analysis is still largely valid but would need more careful handling on the variance. E.g., one may needs to bound the nominator and the denominator of $X^\top \mathbf{b}_T$ separately when $T$ is finite, and bound their ratio. Since such a bound, which is expected to be quite complicated, may not fundamentally lead to new insights into the dynamics of self-attention, we choose a cleaner path.
> > > >
> > > > > How do you define/initialize the positional encoding in the limit of infinite length? Is the positional encoding still meaningful? I think I cannot tell whether a token is far or close to the query if the length is infinite.
> > > >
> > > > You can definitely define positional encoding when $T \rightarrow +\infty$! E.g, if you choose to use ALiBI style positional encoding, then the additional bias can be a decaying (and/or sinusoidal) function of length $b=b(k)$, even if $k$ can be infinite. This gives a measure on how far away a contextual token is from the query/last token.
> > > >
> > > > > I think this work may have the potential to study some out-of-distribution generalization of the language models as future work. Can you provide a discussion?
> > > >
> > > > We are not experts on OOD generalization and please take our opinions with a grain of salt. The term OOD may not be a well-defined term, since two samples that are far away from each other under one specific representation (and thus is treated OOD when predicting one given another), may turn out to be super close to each other under another representation (and is not treated as OOD). Our analysis shows that the self-attention dynamics may group the tokens that most co-occurs with the query together, and pay less attention to the contextual tokens that are related but not super-related, which could help battling against spurious correlation among tokens and lead to a better representation of the input. This better representation can help OOD performance.
> > > >
> > > > Let us know if you have more questions. Thanks.

---

> > > > > ### Comment · Reviewer_TXYd · 2023-08-16
> > > > >
> > > > > Thank you for the reply.
> > > > >
> > > > > I am satisfied with the plan of revisions on the presentation, including the table. For my new questions, I can feel that some of my new questions are a bit out of the range of this work, so overall, I am satisfied with the answers, although some answers are not exciting enough to me. Actually, one very interesting part of the analytical framework in this work is the construction of the conditional probability between tokens, while many existing works on Transformers formulate the data only based on features/patterns. So this is a new perspective.
> > > > >
> > > > > Before I make any changes to the rating, I would like to mention that I said this in my last comment,  **"A minor point is that I don't think [R4] is a feature-learning work. I think it is still NTK work. From my understanding, feature learning works have a strong assumption of the data features, based on which they can derive stronger conclusions. Also, the returned model may change much from the initialization."** Could you please clarify it for a correct discussion of related works?

---

> > > > > > ### Author Response · Authors · 2023-08-16
> > > > > > **Thanks again and clarification of [R4]**
> > > > > >
> > > > > > Dear reviewer, thanks for the encouraging comments. We are glad to know you are satisfied with our revision plan and answers. We agree that one of the comparative advantage of our work is to quantify the patterns with conditional probability. Under this framework, we will provide more detailed analysis with fewer assumptions in the future.
> > > > > >
> > > > > > We kindly disagree with the reviewer about [R4] and do not think it is NTK work. [R4] indeed has a structured assumption about the data distribution: it assumes that the data are generated from a teacher network (see Eqn. 3.1 for two layer and Eqn. 4.1 for three layer cases) and thus the data distribution is not arbitrary. The model indeed changes the weights under SGD gradient dynamics, i.e., the weights are initialized with Gaussian distribution with small standard derivation (Sec. 4.1, $m_1$ and $m_2$ are large) and their loss on dataset generated by the teacher is bounded compared to optimal after training (Theorem 1 and 2), when $m_1$ and $m_2$ are large. Unlike NTK setting that treats the learning process as a convex optimization, [R4] also consider the nonlinear and nonconvex interactions between layers in their proof (e.g., Lemma 6.5 about sign changes of ReLU layers during training).
> > > > > >
> > > > > > Let us know if you have more questions. Thanks.

---

> > > > > > > ### Comment · Reviewer_TXYd · 2023-08-17
> > > > > > >
> > > > > > > I see your point. It makes sense. It seems we may have different definitions of NTK and feature learning. I agree that [R4] applies non-convex optimization to the learning process, especially for the three-layer case. My understanding is that their proof of the two-layer case is still close to a convex optimization. One can check their Lemma B.4. Meanwhile, there is also evidence of NTK for their three-layer case. In their lemma 6.5 (1), they mention "sparse sign change," which means very few sign changes. Therefore, I think they still try to build a connection between the non-convex problem to the convex case. Note that if there are no sign changes, the network becomes linear, and the optimization becomes convex. That's why I feel it is an NTK work rather than a feature-learning work.
> > > > > > >
> > > > > > > For the feature-learning work, the authors can check "Technical novelty" part of [1], which is right before their Section 4. They include some feature-learning works. My understanding is that feature-learning frameworks usually consider a much stronger data assumption than NTK, such as orthogonal features of data. They can often derive that the returned model has a closed form with these orthogonal features included. Therefore, it means the learning process learns the features. [R4] shows their returned model can fit the target function, which is somehow relevant but still different from my understanding.
> > > > > > >
> > > > > > > Anyway, I think this part is a minor point, and we do not have to spend too much time on it. My suggestion is that when you say [1] is a feature learning work, it is better not to mention [R4] as a reference. I think [R4] is at least not a **typical** feature learning work. There are many other better choices, as mentioned above in [1].
> > > > > > >
> > > > > > > I do appreciate the effort of the authors during this rebuttal. I think my major concerns are addressed. I am looking forward to seeing an improved presentation in the next version. I have increased my score to 5.
> > > > > > >
> > > > > > > [1] Li et al., "A Theoretical Understanding of Shallow Vision Transformers: Learning, Generalization, and Sample Complexity", ICLR 2023.
> > > > > > >
> > > > > > > [R4] Z. Allen-Zhu et al, Learning and Generalization in Overparameterized Neural Networks, Going Beyond Two Layers, NeurIPS’19

---

> > > > > > > > ### Author Response · Authors · 2023-08-17
> > > > > > > > **Thanks for raising the score and interesting discussion!**
> > > > > > > >
> > > > > > > > Dear reviewer, thanks for your additional comments and we are glad to hear that your major concerns have been addressed and raised the score to 5.
> > > > > > > >
> > > > > > > > We agree that the assumption in [R4] is not typical for feature learning papers, in which data distribution is often assumed to be a generative model with clear interpretation (often with a clear signal term plus noise). From our point of view, as long as the paper analyzes the change of features (or lower-layer representation) during training, it is beyond NTK. Obviously, your definition on feature learning is different and that's totally fine.

---

### Official Review · Reviewer_zgiA · 2023-07-05

**Soundness:** 3 good
**Presentation:** 3 good
**Contribution:** 3 good
**Rating:** 6
**Confidence:** 1

**Summary:**

This paper presents a rigorous mathematical analysis of the training dynamics of a 1-layer Transformer architecture without positional encoding for the task of next token prediction. The authors demonstrate that the self-attention mechanism in the Transformer exhibits a discriminative scanning algorithm that gradually focuses on distinct key tokens while paying less attention to common key tokens. The paper also shows that the self-attention layer undergoes a phase transition controlled by the learning rate of the decoder layer, resulting in a stable token combination. The authors verify their findings on synthetic and real-world data (WikiText-103).

The reviewer didn't check the proof in mathematics for this manuscript.

**Strengths:**

1. The paper provides a formal and mathematically rigorous analysis of the training dynamics of 1-layer Transformer models, contributing to a better understanding of how these models work.

2. The paper demonstrates the impact of the learning rate on the phase transition in the self-attention layer and its influence on the final token combination.

3. The authors verify their findings on both synthetic and real-world data, which strengthens the validity of their conclusions.

**Weaknesses:**

1. The experimental part of the paper is limited in scope, focusing on 1-layer Transformer models without positional encoding and not addressing more complex architectures.

2. The paper's assumptions (no positional encoding, long input sequence, and faster learning in the decoder layer) may limit the generalizability of the findings to a broader range of Transformer models and tasks.

3. In the training and fine-tuning of the Transformer model, Adam/AdamW is used more often than the SGD analyzed in this paper.

**Questions:**

N/A

**Limitations:**

addressed

---

> ### Author Rebuttal · Authors · 2023-08-06
>
> We thank the reviewer for the insightful comments!
>
> > The experimental part of the paper is limited in scope, focusing on 1-layer Transformer models without positional encoding and not addressing more complex architectures.
>
> We emphasize that most of the experiments are focused on verification of the theory. Also we tried a 3-layer transformer on WikiText-2 and showed the behavior of attention sparsity predicted by the theory. Overall our work is mainly theoretical and we focus on checking whether our theoretical prediction is correct or not, so we mainly focus on 1-layer transformer.
>
> > Adam/AdamW are used more often in practice
>
> Note that previous and concurrent works [1][2], some works published in ICLR'23/ICML'23, also focus on the setting of SGD training in 1-layer transformer. While Adams are used in practice, SGD is a base case that we want to understand well first.
>
> Also, in this work, we empirically show the behavior of Adam optimizer in our synthetic setting (Fig. 5), which is indeed different from SGD and has very interesting behaviors. In Adam, the frequency bias seems to be controlled not only by the co-occurrence frequency of key and query tokens, but also controlled by Adam’s learning rate. This could explain why we need cosine learning rate for Adam, in order to sweep through possible co-occurrence frequency pairs. Note that this phenomenon is not mentioned by previous works, to our best knowledge, and can trigger interest in the community.
>
> We leave a rigorous analysis on Adam to future work.
>
> > Assumptions may limit the generalizability
>
> We want to emphasize that these assumptions are reasonable. People already show decoder only approaches also work reasonably well in no positional encoding [B] and currently both GPT4/Claude and the LLM community is now exploring much longer sequence input [C,D] (e.g., 32k, 65k or longer). We explained the third assumption ($\eta_Z \ll \eta_Y$), which is a technical assumption for the theorems, in the overall rebuttal.
>
> **Reference**
>
> [1] Li et al., "A Theoretical Understanding of shallow Vision Transformers: Learning, Generalization, and Sample Complexity", ICLR 2023.
>
> [2] Oymak et al., "On the Role of Attention in Prompt-tuning", ICML 2023.
>
> [3] Tarzanagh et al., "Max-Margin Token Selection in Attention Mechanism", https://arxiv.org/pdf/2306.13596.pdf
>
> [B] A. Kazemnelad et al, The Impact of Positional Encoding on Length Generalization in Transformers, arXiv'23
>
> [C] S. Chen et al, Extending Context Window of Large Language Models via Positional Interpolation, arXiv'23
>
> [D] J. Ding et al, LONGNET: Scaling Transformers to 1,000,000,000 Tokens, arXiv'23

---

> > ### Author Response · Authors · 2023-08-18
> > **Let us know if you have more questions.**
> >
> > Dear reviewer zgiA, the deadline of discussion period is approaching. Please let us know if you have any further concerns regarding to our work. Thanks!

---

> > ### Comment · Reviewer_zgiA · 2023-08-20
> > **Re: Rebuttal by Authors**
> >
> > Overall, the reviewer is satisfied with the author's response. The rating has been increased accordingly.

---

### Official Review · Reviewer_sxfK · 2023-07-21

**Soundness:** 3 good
**Presentation:** 2 fair
**Contribution:** 2 fair
**Rating:** 5
**Confidence:** 2

**Summary:**

The paper studies the training dynamics of single layer Transformers. They identify a certain scan and snap procedure of the Transformer that learns a winner-take-all solution given certain data statistics / training dynamics or the self-attention learns to combine tokens.
They accompany their theoretical results and analyses with empirical results.

**Strengths:**

I salute the effort of the authors to mathematically study the training dynamics of Transformers. This is very interesting and timely. I think that the authors put in a lot of effort into this study to work out quite interesting results.

**Weaknesses:**

The paper is very dense, hard to parse and therefore difficult to understand. Although I am aware of the difficulty to present theoretical results in a comprehensible manner, I urge the authors to invest time into the presentation of the work.
In general, there are various assumptions made along the paper which is fine if justified or discussed. See questions.
Note that I read the paper a couple of times and still have major difficulties to obtain an overview of the results.



**Questions:**

Again, in general I found the assumptions very strong and I felt that these shortcomings could be a bit better addressed in the experimental section.

1)  Lemma 1: gradient dynamics of batchsize 1 - you are assuming here training the Transformer on a single example only? This feels very restrictive.
2) Lemma 1: Y and Z are assumed to be independent but depend partially on the same parameter matrices. This feels again like a very strong assumption.
3) Can you elaborate on why you choose to integrate layernorm in your analysis? Could simplify your analyses if not important for your results, or did I miss their importance for your analyses?


**Limitations:**

I believe assumptions and limitations are discussed.

---

> ### Author Rebuttal · Authors · 2023-08-06
>
> We thank the reviewer for the comments! Here are the answers:
>
> > Lemma 1: gradient dynamics of batchsize 1 -> Transformer trained with single example only.
>
> First of all, in machine learning, training with batchsize = 1 means that in each gradient step, only one sample is used to update the gradient, it does not mean we train on a single example only, since each time we could use a different sample from the dataset.
>
> Furthermore, the fact that Lemma 1 takes the form of batchsize = 1 does not mean it cannot be applied to batchsize > 1: we can simply sum over sample index i (omitted in Eqn. 3) to get the gradient of Y and Z.
>
> > Lemma 1: Y and Z are assumed to be independent but depend partially on the same parameter matrices. This feels again like a very strong assumption.
>
> This is a common reparameterization technique used when analyzing Transformer and many previous theoretical works, including works [2] suggested by **TXYd**, leverage this technique. In [2], key/query matrices are merged to one (in their Eqn. 4), q is defined as W_kW_q^T P^T (in their Eqn. 4) and its dynamics is computed instead of the dynamics of prompts embedding P. In [3] the key-query weights are merged into one matrix W and its dynamics is studied. In [A], similar to ours, they use X to replace QK^T as the variable to be optimized and study the property of X in optimization instead.
>
> > Why choose to integrate layernorm in your analysis?
>
> LayerNorm plays an important role in our analysis. LayerNorm provides the additional projection operator $P^\perp_{X^T b_T}$ (as shown in the proof) that will cancel out one term in the derivative of softmax (line 539 in supp material), leading to much simpler analysis (e.g., Eqn.95 in supp material). We will make it clear in the revised version.
>
> **Reference**
>
> [1] Li et al., "A Theoretical Understanding of Shallow Vision Transformers: Learning, Generalization, and Sample Complexity", ICLR 2023.
>
> [2] Oymak et al., "On the Role of Attention in Prompt-tuning", ICML 2023 (Jun. 6)
>
> [3] Tarzanagh et al., "Max-Margin Token Selection in Attention Mechanism", arXiv’23 (Jun. 23)
>
> [A] S. Li et al, The Closeness of In-Context Learning and Weight Shifting for Softmax Regression, arXiv

---

> > ### Comment · Reviewer_sxfK · 2023-08-12
> > **Thank you**
> >
> > Thank you for your response and note that this was an emergency review only providing me very limited time to review. I apologize for the very shallow review. I will discuss with the other reviewers once the rebuttal phase end and for now only increase my score slightly.
> > Many thanks again

---

> > > ### Author Response · Authors · 2023-08-15
> > > **Thanks!**
> > >
> > > Thanks for your valuable time! We understand that the review process can be time-limited and stressful. Any comments and feedbacks are welcome and we will address them to the best of our knowledge.
> > >
> > > Let us know if you have more questions and we would greatly appreciate if you are confident in further increasing the score. Thanks again.

---

### Author Rebuttal · Authors · 2023-08-06

We thank all reviewers for their insightful feedback.

We are glad to hear that reviewers agree that a rigorous framework/analysis of the training dynamics of Transformer is valuable, interesting, novel and timely[**sxfK**, **TXYd**, **nH6i**], with clear high-level intuitions [**TXYd**, **nH6i**] and experiments on both synthetic and real-world datasets [**zgiA**]. We will address the common questions below and will reply to each reviewer about answers to their detailed questions.

**[ZSxf, zgiA, nH6i] Regarding the setting of 1-layer transformers**. We observe a mixed point of view around this aspect. Some reviewers think it is too simplistic (**ZSxf**, **zgiA**) while others (nH6i) think it makes the overall picture clear and helps understanding.

From our point of view, analyzing a 1-layer transformer is a necessary step. As shown in our paper, there are many interesting and non-trivial behaviors even in this apparently simple case. To see this, let’s check previous and concurrent works [1][2][3] listed by reviewer **TXYd** asking for comparison. [1][2][3] are important works, [1][2] published in ICLR'23/ICML'23, and all focus on 1-layer transformers ( = 1 layer self-attention + 1 FFN). A common theme is that they show self-attention in1-layer attends to relevant tokens during training under various settings and data models (e.g., when the model is fixed and we fine-tune on soft-prompts), therefore, even with 1-layer, many phenomena are nontrivial and deserve dedicated work published in top-tier conferences.

**Our contributions**. Compared to [1][2][3], our work is novel in the following ways:

+ Among relevant/distinct tokens of a sequence class, we characterize their relative growth quantitatively, showing that self-attention attends to *a subset of distinct tokens* that co-occur a lot with the query, leading to attention sparsity even among relevant tokens. In comparison, [1][2] only show high attention scores for relevant tokens and thus attention sparsity relies on number of relevant tokens in the assumption.

+ We characterize complete training dynamics for multi-class settings, by *integrating nonlinear dynamics analytically*. We also summarize two stage behaviors of attention scores (scan & snap) toward convergence. In comparison, [1] relies on a much more complicated technique and [2] focuses on initial gradient steps and [3] assumes linear separability of tokens generated from two classes and uses max-margin SVM framework.

We list the detailed comparison in the our rebuttal to reviewer **TXYd**.

Note that even with 1-layer, our work is already criticized as very dense (**sxfK**). In fact, it can be **unrealistic** to expect a rigorous and detailed formulation of a multi-layer transformer in a single 9-page conference paper, and at the same time, with a rich connection with real-world applications. It'd better to focus on simplified settings to make the take-home message clear. Therefore we focus on 1-layer Transformer and leave its multi-layer extension as the future work. We really appreciate reviewer **nH6i** for the understanding!

**[zgiA, nH6i] Future directions to address multi-layer cases**. For multi layer, an important component is how the input tokens are combined together to form high-level concepts during training. Our work shows that the training leads to sparse attention even among relevant tokens, and demonstrates that there is a priority in token combinations for 1-layer attention based on their co-occurrence: even if 10 contextual tokens are relevant to the query, the self-attention may pick 1-2 token to combine first due to attention sparsity. This can be regarded as a starting point to study how tokens are composed hierarchically. In comparison, showing that attention attends to all relevant tokens [1][2][3] may not suggest a hierarchical / multi-layer architecture, which is used in practice.

**[ZSxf, sxfK, zgiA] Concerns on the strength of assumptions.** Understanding transformers in a mathematically rigorous manner is a highly nontrivial problem, and the assumptions we made are comparable with or even weaker than previous works. For example, [R1] analyzes positional attention with symmetric initialization, without considering input data. [1] models the data distribution as discriminative/non-discriminative patterns, similar to ours, assume hinge loss (rather than cross entropy loss), and perform SGD training. [2] also models the data distribution as relevant/irrelevant. [R3] models transformer dynamics near the initialization and freeze many parameters at random initialization. In comparison, we characterize the entire training dynamics with a few well-defined assumptions. We will list the detailed comparison in the reply to reviewer **TXYd**.

**[ZSxf, TXYd] Fast training of the decoder Y than the self-attention layer Z ($\eta_Z \ll \eta_Y$)**: This is a technical assumption in order to obtain Theorem 1, which establishes the relationship between the input signal $f_n$ of the decoder Y and the back-propagated gradient $g$, once the decoder $Y$ has been sufficiently trained. We indeed see works that use different learning rates at different layers in the empirical studies, e.g., [R2].

As a future work, we are actively looking for better approaches that can get rid of Assumption 2 and separate learning rate in the analysis.

**References**

[R1] S. Jelassi et al. Vision transformers provably learn spatial structure. NeurIPS’22.

[R2] E. Dinan et al. Effective Theory of Transformers at Initialization

[R3] A. Bietti et al, Birth of a Transformer: A Memory Viewpoint

[1] Li et al., "A Theoretical Understanding of Shallow Vision Transformers: Learning, Generalization, and Sample Complexity", ICLR 2023.

[2] Oymak et al., "On the Role of Attention in Prompt-tuning", ICML 2023 (Jun. 6)

[3] Tarzanagh et al., "Max-Margin Token Selection in Attention Mechanism", arXiv’23 (Jun. 23)

---

### Decision · Program_Chairs · 2023-09-21

**Decision:**

Accept (poster)

**Comment:**

All reviewers agree that this paper studies the important problem of analyzing training dynamics of Transformer models, but under very strong assumptions, such as, 1 layer transformer model without positional encoding, different convergence rates for different parts of the model, no non-linearity in the MLP layers and the results are in the limit of infinite sequence length.

While these are pretty strong assumptions, given the popularity of Transformer models in practice, the paper makes small but important progress towards better understanding of the Transformer dynamics.

Reviewers also raised some concerns about comparison to some of the recent papers analysing Transformer training dynamics. Authors provided a detailed response on the difference with their work and promised to update the paper with detailed comparison.

Overall the paper is on borderline and I suggest acceptance given the importance of the problem studied and the results in the paper. I suggest authors to update the paper to improve presentation following review suggestions. Further the paper (including abstract) needs to state the assumptions correctly as the results are in the limit of infinite sequence length not just long sequence length.